# Global land and carbon consequences of mass timber products

Kai Lan ●[1,2,5], Alice Favero ●[3,5], Yuan Yao ●[1] ✉, Robert O. Mendelsohn ●[4] ✉ & Hannah Szu-Han Wang[1]

Mass timber products can reduce greenhouse gas emissions by replacing steel and cement. However, the increase in wood demand raises wood prices, and the environmental consequences of these market changes are unclear. Here we investigate the global carbon and land use impacts of adopting mass timber products, focusing on cross-laminated timber as a case study. Our results show that higher wood prices reduce the production of traditional wood products but expand productive forestland by 30.7–36.5 million hectares from 2020 to 2100 and lead to more intensive forest management. If the cumulative global cross-laminated timber production reaches 3.6 to 9.6 billion m³ by 2100, long-term carbon storage can increase by 20.3–25.2 GtCO₂e, primarily in forests (16.1–17.7 GtCO₂e) and in cross-laminated timber panels (4.1–8.1 GtCO₂e). Including emission reductions from steel, cement, and traditional wood products, the net reduction of life-cycle greenhouse gas emissions will be 25.6–39.0 GtCO₂e.

Limiting the global average temperature to 1.5 °C or 2 °C by 2100 requires large-scale decarbonization of industrial systems and the built environment[1–4]. In 2020, the building construction industry caused 10% (3.2 GtCO₂e) of global greenhouse gas (GHG) emissions, and it is expected to increase its share of emissions as the global population and urbanization increase[5]. Decarbonizing the construction industry by utilizing low-carbon intensity and renewable materials is a promising solution[5]. Mass timber products like cross-laminated timber (CLT) are emerging renewable alternatives to traditional building materials (e.g., reinforced concrete and steel), especially for mid- and high-rise buildings[6]. Previous studies have shown the potential carbon benefits of adopting mass timber products over traditional construction materials[6–8]. Life Cycle Assessment (LCA) has been employed at the product level (e.g., 1 m³ wood product)[9,10] and the stand level (e.g., 1-ha forestland)[11] to quantify the environmental performance of mass timber products. Although these studies advanced the knowledge of CLT, most of them only include the direct impacts associated with wood product life cycles (e.g., forest growth, harvesting, wood production, product use, and end-of-life). There is a lack of understanding regarding the effects of increasing wood demand on the life-cycle GHG emissions of both traditional wood products (e.g., lumber, particle boards, pulp and paper products) and mass timber products, and the long-run implications to land use and forest management[10].

Several studies have investigated the global forest impact of mass timber adoption[6,12,13], recognizing the value of using CLT as part of a potential global climate mitigation program. However, few studies have addressed how various forest types, forest management activities, and the timber market will evolve in response to the new demand for mass timber. Moreover, the carbon consequences throughout the life cycles of both CLT and traditional wood products, from forest to end-of-life, have not been fully examined. There is also limited exploration of the high cost of converting marginal land into plantations and growing timber in non-forest ecosystems. Finally, these land use models may not adequately measure the age classes of forests, which are necessary to understand when to plant and how quickly the timber market can respond to changes in demand.

In this work, we aim to address these gaps in the literature by developing an interdisciplinary modeling framework that integrates economics, ecology, and LCA to quantify the interactions among demand, supply, forest management, and land use at both regional

[1]Center for Industrial Ecology, Yale School of the Environment, Yale University, New Haven, CT, USA. [2]Department of Forest Biomaterials, College of Natural Resources, North Carolina State University, Raleigh, NC, USA. [3]RTI International, Research Triangle Park, NC, USA. [4]Yale School of the Environment, Yale University, New Haven, CT, USA. [5]These authors contributed equally: Kai Lan, Alice Favero. ✉e-mail: y.yao@yale.edu; robert.mendelsohn@yale.edu

and global levels. This framework integrates the Global Timber Model (GTM)[14–17], with process-based LCA models to quantify the direct and indirect land and carbon implications of emerging wood product demand from 2020 to 2100. This framework is applied specifically to CLT, a common mass timber product[6,11,18]. The LCA quantifies the life-cycle GHG emissions associated with CLT, traditional wood products, and substituted products (e.g., concrete, steel, electricity). This study compares a baseline scenario without future CLT demand with three alternative CLT demand scenarios to examine the effects of CLT demand on the timber market.

We utilize GTM to capture how adding the demand for mass timber products will likely lead to higher wood prices, which will potentially reduce the global supply for traditional wood products. The higher wood prices give landowners an incentive to plant more forests and increase forest management intensity by, for instance, increasing fertilization or changing forest rotation. GTM captures how global forest markets anticipate these future changes in demand and supply. The changes in supply, in turn, impact carbon storage. GTM also captures the ability of forest ecosystems to supply timber by modeling stands by age in global forests. The model does not include non-forest ecosystems because it is very costly to grow timber in these ecosystems. For example, one must drain wetlands and irrigate parklands and drylands to make them suitable for growing timber. The price of wood is unlikely to get high enough to justify these expenses.

The integrated modeling framework allows for a holistic assessment of the net climate mitigation effect of adopting CLT at global and regional scales. This modeling framework can support policymaking and decision-making about the environmental consequences of large-scale use of CLT. Moreover, this framework can be adjusted using other assumptions about CLT demand, alternative emerging wood products, or recycling. The framework is also useful for addressing policy tradeoffs concerning how to use forests for potentially competing goals such as carbon storage, carbon removal, and conservation. The results shed light on the future impacts of increased wood use on global forestlands, carbon, and the timber market. It informs better design and operations of coupled forest-CLT-building systems as a nature-based solution for climate change. Specifically, the CLT analysis reveals that increasing the demand for wood, at least up to a maximum sustainable level, will increase carbon storage in forests and

wood products and reduce GHG emissions from steel, cement, and traditional wood products.

## Results

### Overviews of methods and scenarios

Figure 1 shows how GTM and LCA are integrated to assess the effects of CLT demand on different factors (e.g., timber markets, forest area, and GHG emissions), and their implications on the carbon balances. GTM captures how increased future CLT demand raises wood prices and increases managed forestland and plantations by reducing natural forestland and farmland, and tracks incidental changes in forest carbon storage. At the same time, GTM quantifies how forest management intensity increases the growth rates of trees. These changes increase future wood (i.e., sawtimber and pulpwood) supply to meet future demand. Note that, in this study, GTM does not experiment with the demand and supply of fuelwood in response to the emerging CLT demand. The LCA captures the net biogenic carbon uptake by forest and life-cycle GHG emissions by forest operations, by CLT production and end-of-life, by potentially replacing steel and concrete, and by production and end-of-life of traditional wood products. All these complex interactions are captured in this internally consistent framework to measure the carbon and land consequences due to CLT adoption.

Economic growth and policy can lead to a range of future outcomes, which we capture with three plausible pathways of CLT demand and adoption scenarios versus a baseline scenario without CLT demand through 2100. The future baseline global demand for traditional wood products from sawtimber and pulpwood is based on the population and economic growth predicted in the Shared Socioeconomic Pathway 2 (SSP2) scenario[19]. In the baseline scenario, total wood supply increases from 2122 million m³ in 2020 to 3627 million m³ in 2100 (see Supplementary Table 1) which drives a price increase in sawtimber (59.4%) and pulpwood (22.8%) (see Section Effects on traditional wood supply and price) and a corresponding increase in plantations of 39.3 Mha (see Supplementary Table 2).

We then use the socioeconomic and biophysical assumptions of the baseline to examine three CLT scenarios. We focus on how each CLT scenario changes the baseline outcomes over time. We examine a medium CLT demand scenario from 2020 to 2100 that reaches 30% of

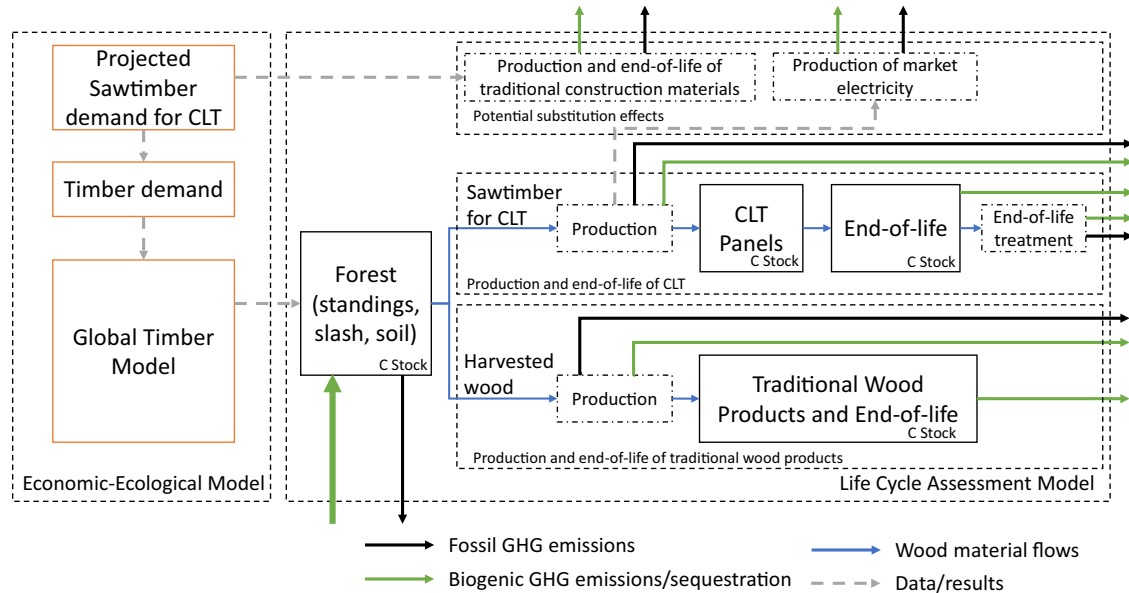

**Fig. 1 | The system boundary and carbon flows of the modeling framework.** The system boundary of the Life Cycle Assessment is cradle-to-grave. The green arrow to the forest box stands for CO₂ sequestration from the atmosphere. The

summation of the arrows entering and leaving the system boundary equals the net greenhouse gas (GHG) balance. CLT stands for cross-laminated timber.

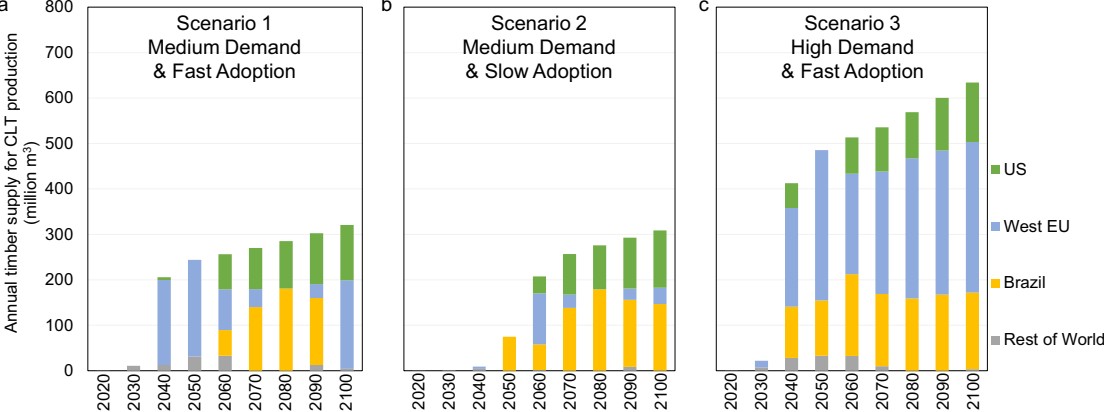

**Fig. 2 | Supply of sawtimber for CLT production from 2020 to 2100. a** Scenario 1 with medium demand and fast adoption; **b** Scenario 2 with medium demand and slow adoption; **c** Scenario 3 with high demand and fast adoption. The global sawtimber supply for cross-laminated timber in the three scenarios is shown for various major regions, including the US (United States of America), West EU (Western Europe), and Brazil, along with the rest of the world. Source data are provided as a Source Data file.

global new urban construction floor area, with either fast (Scenario 1, reaching 30% by 2050) or slow (Scenario 2, reaching 30% by 2080) adoption. We also look at fast adoption with a high demand scenario (Scenario 3, reaching 60% by 2050) (see "Methods" and Supplementary Fig. 1). These CLT demand scenarios are based on predicted future new construction of urban commercial and residential building floor areas which we assume depend on Gross Domestic Product (GDP) per capita and urban population (see "Methods")[20,21].

## Supply of sawtimber for mass timber products

Figure 2 shows the global sawtimber supply for CLT in the three scenarios. The annual global wood supply for CLT in Scenario 1 and Scenario 2 reaches about 300 million m³ (around 9% of total wood supply for traditional wood products, see Supplementary Table 1) by 2100, whereas the high demand scenario reaches 634 million m³ (19% of total wood supply for traditional wood products, see Supplementary Table 1). Because CLT likely will come from softwoods, GTM predicts CLT will likely be made from wood that comes from three regions: Western Europe, the United States, and Brazil (Fig. 2). These regions together produce more than 60% of the traditional global wood supply in the baseline scenario. It is cost-effective for these regions to switch a portion of their production from traditional wood products to CLT. The United States and Brazil are also predicted to plant trees for CLT. When there is a rapid, immediate increase in demand, the early wood supply for CLT comes from existing trees in Western Europe and the Rest of the World. For example, in Scenario 3 with high CLT demand and fast adoption, Western Europe supplies 43.0%–66.2% of wood for CLT from 2020 to 2100. But in Scenario 2, the wood supply for CLT from Western Europe decreases from 54.0% in 2060 to 11.6% in 2100.

## Effects on the traditional wood supply and price

When the global demand for CLT is added, the prices of sawtimber and pulpwood rise relative to the baseline scenario as shown in Fig. 3a, b (see Daigneault and Favero[22] for detailed socioeconomic assumptions under the baseline scenario in GTM). Compared to the baseline scenario, the faster CLT demand grows, the earlier prices of sawtimber and pulpwood increase. Higher overall CLT demand leads to higher overall future sawtimber and pulpwood prices. The higher future prices stimulate an increase in the future total wood supply (including sawtimber for traditional wood products, pulpwood, and sawtimber for CLT) (see Supplementary Table 1). The change in CLT is large enough to shift the aggregate demand for wood, but even in the most aggressive scenario, global aggregate wood supply only increases by

10.7%. Some of this extra wood supply comes from increasing forest management intensity, which increases the growth rate of managed forestland and leads to more plantations (see Fig. 4). Supply also increases by moving natural forestland and low-valued farmland into managed forestland (see Fig. 4). Previous literature[12] predicted that all future supply could come from plantations and "other land" categories (non-forest ecosystems), but plantations require highly productive land. It is not clear that the previous literature distinguished high-productivity versus marginal productive lands within each ecosystem[12]. Non-forest ecosystems (savannah, parkland, and mangroves) require expensive inputs to make them suitable for growing timber. Dryland has to be irrigated, and wetlands have to be drained. The price of wood in these CLT scenarios is not high enough to pay for these inputs. Finally, it is important to recognize that higher future wood prices will reduce the quantity of wood going to sawtimber for traditional wood products and pulpwood, so there is more wood available for CLT.

## Forest area change

Figure 4 shows the change in total forest area by 2100 relative to the baseline (see Supplementary Data 1 for the region definitions in the GTM). There are three forest types: plantations, managed forests, and natural forests. GTM also captures marginal farmland for livestock and crops in the analysis. At the global level, CLT demand drives a small increase in plantations, a large increase in managed forests, and a moderate decrease in natural forests and marginal farmland relative to the baseline (see the baseline forest area in Supplementary Table 2). Overall, global forestland increases with the introduction of CLT at the expense of low-valued farmland relative to the baseline. Because the baseline predicts a reduction in forestland, the CLT scenarios effectively protect more existing forestland from being converted into future farmland. Under Scenario 1 by 2100, plantations increase by 11.2 million hectares (Mha), managed forestland increases by 32.6 Mha, and natural forestland declines by 10.6 Mha. The net increase of 33.2 Mha in forestland comes from preventing future forestland from being converted to farmland. Under Scenario 2 by 2100, plantations increase by 11.0 Mha, managed forestland increases by 27.9 Mha, and natural forestland declines by 8.1 Mha for a net increase of 30.7 Mha. Under Scenario 3 by 2100, plantations increase by 14.7 Mha, managed forestland increases by 40.8 Mha, and natural forestland declines by 18.9 Mha for a net increase of 36.5 Mha of forestland. In general, the plantation land as a fraction of harvested forestland (8.2%) increases very slightly as CLT increases.

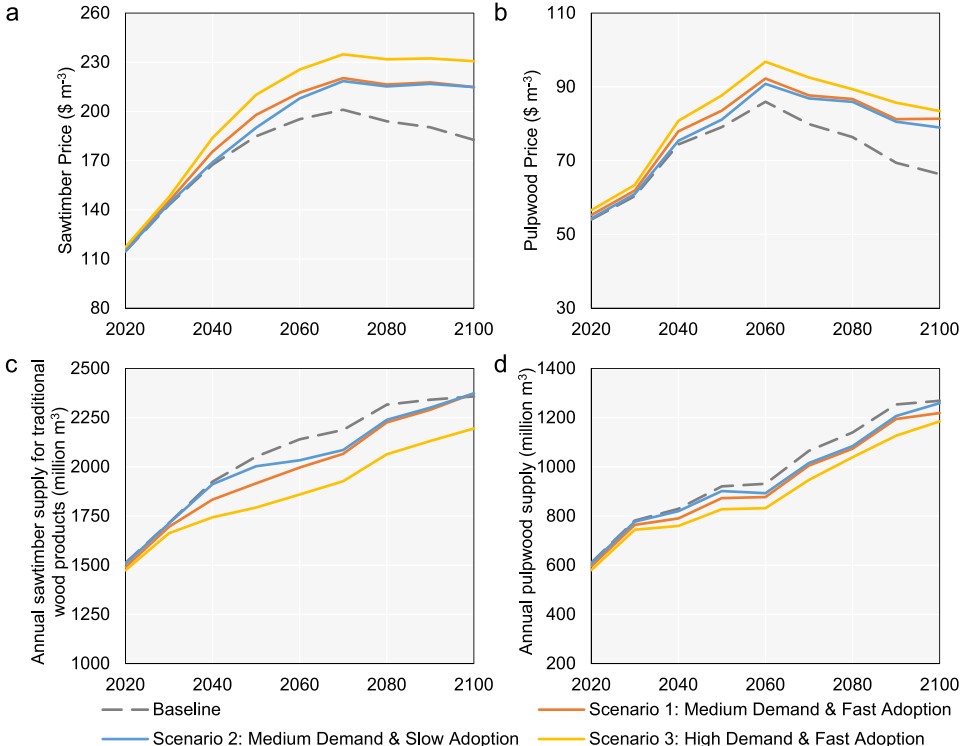

**Fig. 3 | Global price and supply of sawtimber and pulpwood from 2020 to 2100.**
**a** Sawtimber price. **b** Pulpwood price. **c** Globally annual sawtimber supply.
**d** Globally annual pulpwood supply. Notes: The demand and prices of sawtimber
for traditional wood products and pulpwood under the baseline scenario are driven
by socioeconomic assumptions included in the Global Timber Model. Source data
are provided as a Source Data file.

The largest net increase in plantations and managed forestland occurs in the temperate forests of Western Europe and the United States, where most CLT will likely be produced. There are also large increases in managed forestland in China and Canada, as these regions supply more traditional wood products. Sub-Saharan Africa and South America (excluding Brazil) are predicted to replace natural forests with managed forests. Compared to the baseline scenario, Brazil converts natural forest into plantations to supply CLT. Southeast Asia is the only region predicted to lose both managed and natural forestland under the CLT scenarios.

## Global carbon stock changes

Figure 5 depicts the change in three different carbon stocks on a $CO_2$-equivalent ($CO_2$e) basis: (1) forest carbon pools, including aboveground biomass, slash, and soil; (2) carbon stored in CLT panels (light blue bars) and landfill sites (gray bars); and (3) carbon stored in traditional wood products (orange bars). Since CLT demand reduces the production of traditional wood products (see Fig. 3), there will be less carbon stored in traditional wood products.

As shown in Fig. 5, CLT demand is expected to increase the global carbon stock on land across all the CLT scenarios by 20.3–25.2 $GtCO_2$e by 2100. Forest aboveground biomass captures the bulk of this added carbon with 14.6–16.2 $GtCO_2$e by 2100. The other two forest carbon pools, forest soil and forest slash, add 1.5–2.1 $GtCO_2$e by 2100. Moreover, additional carbon stock in forests increases under the slow CLT adoption rate because it gives the market more time to grow new trees. On average, CLT could increase forest carbon stock (including forest aboveground biomass, forest soil, and forest slash) by 16.1 $GtCO_2$e in the fast adoption scenario (Scenario 1) and 17.7 $GtCO_2$e in the slow adoption scenario (Scenario 2) between 2020 and 2100. Compared to Scenario 1, more CLT demand in Scenario 3 increases the forest carbon stock by 10.1%. There is a limit, however, to how much more carbon can be stored in managed forests. The cost of converting more land to

forestland and intensifying forest management increases with scale because there is only so much productive land on the planet, and we use this valuable land for development, especially agricultural purposes.

The carbon stored in CLT panels also increases with the size of the CLT market. The CLT panels are the second largest driver of the increase in carbon stock by the end of the century (4.1 $GtCO_2$e in Scenario 1, 3.2 $GtCO_2$e in Scenario 2, 8.1 $GtCO_2$e in Scenario 3). On the contrary, the CLT demand decreases the carbon stock of traditional wood products by 0.7–2.9 $GtCO_2$e across the scenarios. To explore the impacts of varied CLT end-of-life cases on the results, two additional conceptual end-of-life cases of CLT panels are conducted: (1) material recycling case with 50% closed-loop recycling and 50% landfilling; (2) energy recovery case with 100% CLT panels combusted for power generation (see "Methods"). The impacts from the different end-of-life cases of CLT panels are minor (see results in Supplementary Tables 3 and 4) due to the small quantity of CLT that reached their end-of-life (only those adopted in 2020–2040 in Fig. 2, given a life span of 60 years).

## Global change in greenhouse gas emissions

Figure 6 shows the cumulative changes in life-cycle GHG emissions under the CLT demand scenarios relative to the baseline scenario at the global level. All changes are measured in $CO_2$e, including changes in biogenic $CO_2$ uptake and biogenic and fossil-based GHG emissions (including $CO_2$, $CH_4$, and $N_2O$) throughout the life-cycle stages. The net result is marked by triangles in Fig. 6. The cumulative GHG emission results in 2100 are summarized in Supplementary Table 5.

Through 2100, cumulative global CLT production is 4.8, 3.6, and 9.6 billion m³ in Scenarios 1, 2, and 3, respectively (see Supplementary Table 5), delivering a reduction in net GHG emissions of 27.0, 25.6, and 39.0 $GtCO_2$e, respectively. On average, by 2100, producing 1 m³ CLT is estimated to deliver a net GHG mitigation of 5.6 $tCO_2$e in Scenario 1, 7.2 $tCO_2$e in Scenario 2, and 4.0 $tCO_2$e in Scenario 3. Comparing

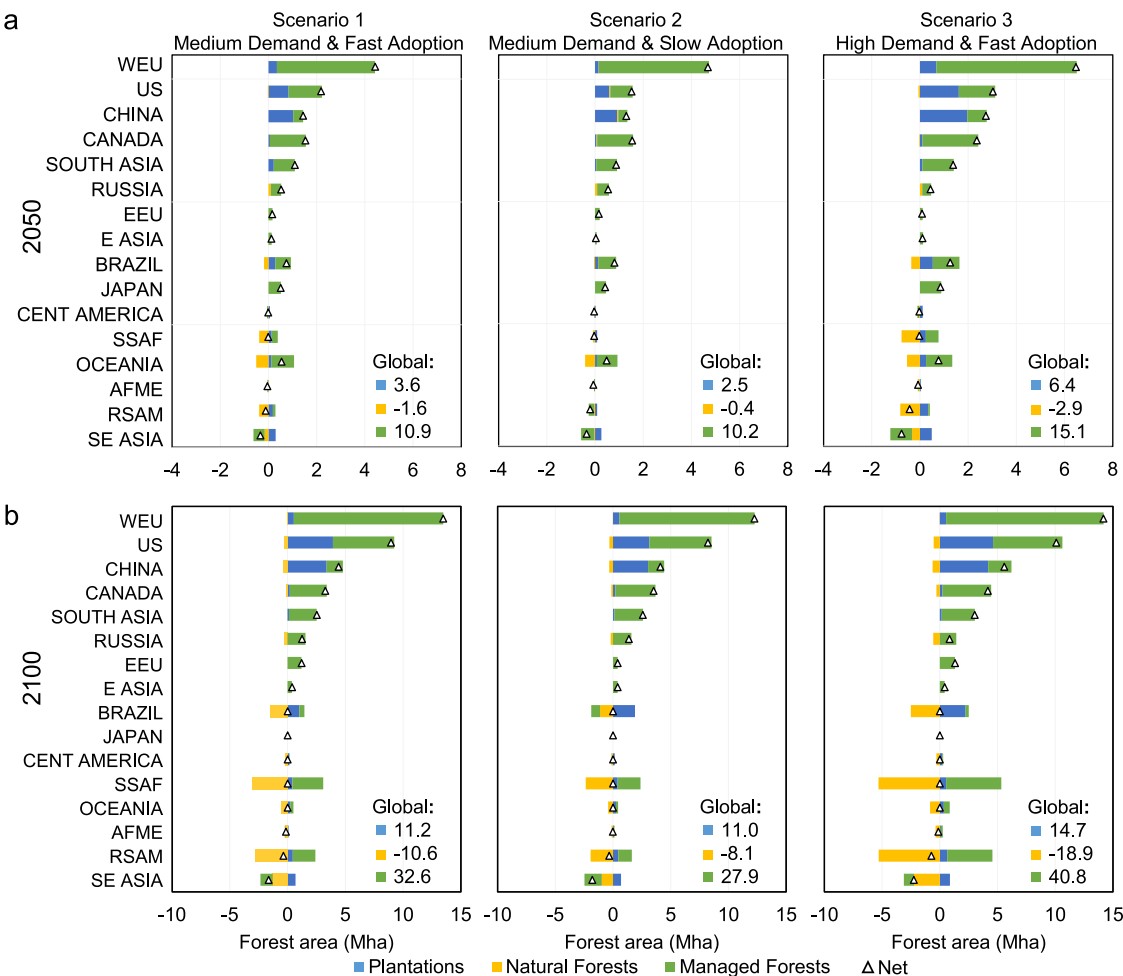

**Fig. 4 | Forest area changes in three scenarios compared to the baseline.**
**a** Forest area changes by 2050; **b** forest area changes by 2100. There are 16 regions: WEU (Western Europe), US (United States of America), CHINA, CANADA, SOUTH ASIA, RUSSIA, EEU (Eastern Europe), E ASIA (East Asia), BRAZIL, JAPAN, CENT AMERICA (Central America), SSAF (Sub-Saharan Africa), OCEANIA, AFME (North Africa and Middle East), RSAM (Rest of South America), and SE ASIA (Southeast Asia). In the Global Timber Model (GTM), plantations are defined as monoculture forests, even-aged systems that are intensively managed for pulpwood and sawtimber production. Examples of plantation systems in the United States include loblolly, slash, and long-leaf pine, and Douglas fir. Managed forests, referring to managed naturally generated forests, originally were naturally regenerated, but most of them are now replanted forests. The key distinction between plantations and managed forests is that plantations are on highly productive land and involve much more costly and intense management (as in a farm). Natural forests are defined as inaccessible, unmanaged, and naturally regenerating forests. Source data are provided as a Source Data file.

Scenarios 1 and 3, the average GHG benefit per $1\,m^3$ CLT declines as the scale of the CLT market increases because market and land competition increase. Comparing Scenario 1 and 2, a slower adoption rate leads to a larger GHG benefit per $1\,m^3$ CLT basis because the market has more time to grow the additional needed trees and increase the forest carbon stock per $1\,m^3$ CLT basis.

Because biogenic carbon uptake comes from $CO_2$ in the atmosphere, we assume the uptake is a reduction in atmospheric $CO_2$. Hence, in Fig. 6, the negative values mean the increase in biogenic carbon uptake. Three factors explain the total net cumulative increase in the biogenic carbon uptake of 22.9–29.3 $GtCO_2e$ (green bars in Fig. 6) from the future CLT demand. The largest contributor is the increased biogenic carbon uptake that is finally stored in forests (16.1–17.7 $GtCO_2e$). Biogenic carbon uptake for CLT production (10.6–28.6 $GtCO_2e$) is the next largest factor. Finally, there is a reduction of biogenic carbon uptake (−4.5 to −17.0 $GtCO_2e$) from the reduction of wood output for traditional wood products because of the high price of wood. Relative to the baseline without CLT, future CLT encourages less future low-valued cropland and more managed forestland. Higher wood prices also increase forest management investments (increasing plantations and managed forests, as shown in

Fig. 4), raising forest growth rates. Despite that, CLT increases future harvesting, there will be more future forest carbon and a net movement of $CO_2$ from the atmosphere into forests. The net wood carbon stored in forests (see Figs. 5 and 6) represents 59.6% of the final net GHG reduction benefit in Scenario 1, 69.1% in Scenario 2, and 45.5% in Scenario 3.

The changes in GHG emissions from CLT production and end-of-life CLT (light blue bars) and traditional wood products (orange bars) sum to a net increase of 1.6–2.4 $GtCO_2e$ (see Supplementary Table 5). However, when the potential substitution benefits (from market electricity and concrete and steel) are included, the net changes in all the remaining GHG emissions are −4.1 $GtCO_2e$ in Scenario 1, −1.8 $GtCO_2e$ in Scenario 2, and −9.6 $GtCO_2e$ in Scenario 3. These net changes in GHG emissions of production and end-of-life make up between 7.1% and 24.7% of the total net reduction in GHG emissions in each scenario (see Supplementary Table 5).

Previous studies estimated the carbon benefits (including avoided GHG emissions of traditional building materials and carbon storage in CLT panels) of $1\,m^3$ CLT to be 1.1–1.5 $tCO_2e$ in the U.S.[7,8]. The results in this study are higher because we account for two factors: one is the increased carbon storage in forests, and the other is the decrease in

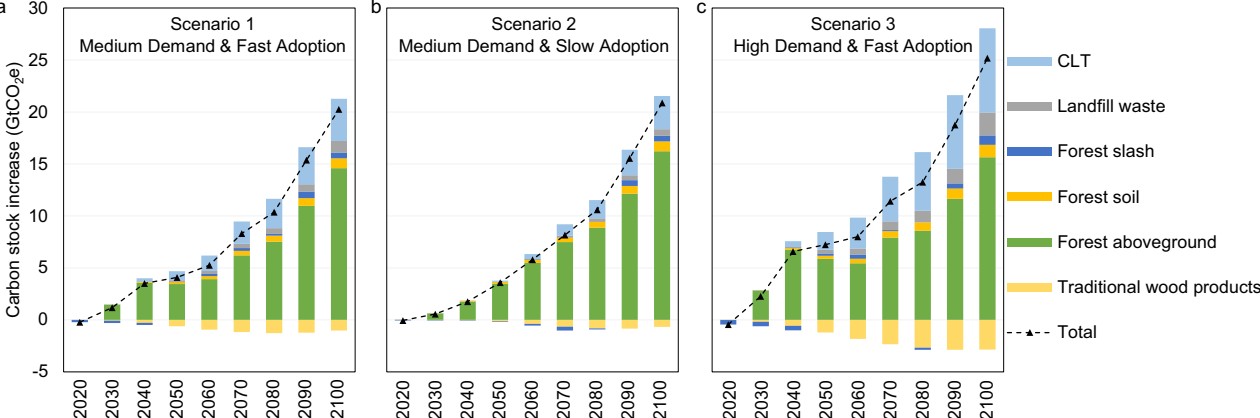

**Fig. 5 | Change in global carbon stock from CLT adoption relative to the baseline between 2020 and 2100 across three scenarios. a** Scenario 1 with medium demand and fast adoption; **b** Scenario 2 with medium demand and slow adoption; **c** Scenario 3 with high demand and fast adoption. All the numbers are reported on the $CO_2$e basis. The positive values imply an increase in carbon storage and, therefore, more $CO_2$ taken out of the atmosphere, whereas the negative values imply a loss of carbon storage and therefore more $CO_2$ in the atmosphere. CLT stands for cross-laminated timber. Source data are provided as a Source Data file.

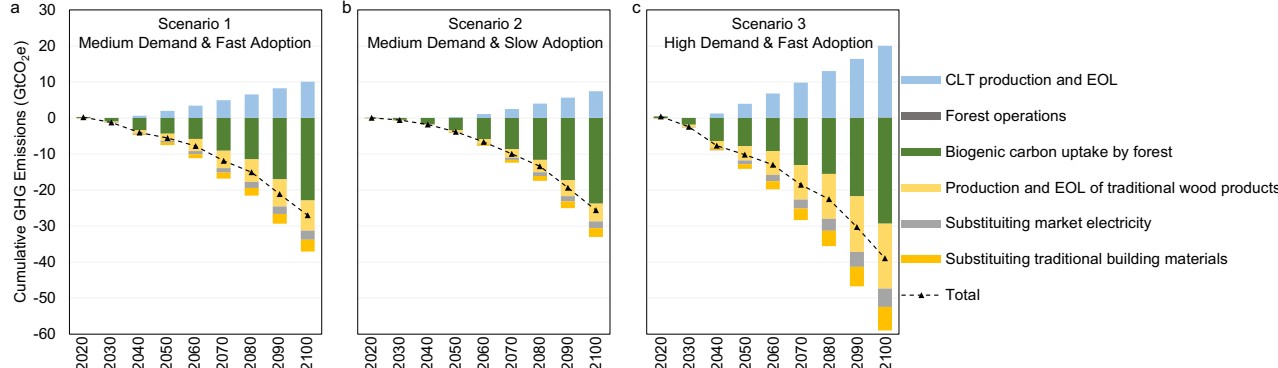

**Fig. 6 | Global cumulative change in greenhouse gas emissions relative to the baseline from three cross-laminated timber scenarios from 2020 to 2100. a** Scenario 1 with medium demand and fast adoption; **b** Scenario 2 with medium demand and slow adoption; **c** Scenario 3 with high demand and fast adoption. Positive values imply increased greenhouse gas (GHG) emissions, and negative values imply reductions in GHG emissions or increases in $CO_2$ sequestration relative to the baseline. CLT stands for cross-laminated timber; EOL stands for end-of-life. Source data are provided as a Source Data file.

life-cycle GHG emissions due to the reduction in traditional wood products. Without these factors, our estimated global average GHG benefit per 1 m³ CLT by 2100 is 2.1 t$CO_2$e, which is comparable to the previous studies. The remaining differences likely arise from the different time frames, locations, building structures, and end-of-life assumptions.

This study also conducts a sensitivity analysis to investigate the impacts of variations in input parameters or assumptions on the results (see Supplementary Fig. 2 for detailed results). Most parameter variations have minor effects. The most sensitive parameters in the analysis are the GHG emission factors of electricity and concrete, which determine the potential substitution benefits. If the global electricity system is decarbonized from 0.70 to 0.33 kg$CO_2$e per kWh, the total GHG emission benefit in Scenario 3 will be 6.1% smaller. If the GHG emissions of producing concrete are reduced by 40.0%, the total cumulative change of GHG emissions will be 2.4% higher. These changes are small relative to the overall effects of CLT explored in Scenarios 1–3.

## Discussion

The framework in this study integrates ecology, economics, and LCA to understand how dynamic CLT projections affect carbon stock, GHG

emissions, and land uses at the regional and global levels. The LCA component captures the life-cycle GHG emissions of CLT and traditional wood products, along with the substitution effects of replacing traditional building materials and market electricity. The key insight offered by this integrated framework is the understanding of how the timber market and specifically forest landowners will respond to the higher wood prices from the new demand for CLT. The framework reveals that the adoption rate affects this response since it takes time to grow an additional wood supply. More importantly, the framework reveals that landowners will act in anticipation of a new demand that has not been recognized in previous studies. Forest owners will start to plant and invest in new forests as soon as they know a new demand for wood is coming. The model pays careful attention to ecological constraints by limiting forestland to ecosystems that can support forests. The model also acknowledges that there are costs to increasing land for managed forests and even higher costs for attracting high-productivity land into plantations. Finally, the GTM model recognizes that it takes a long time to grow timber by carefully tracking acreage by age class. This allows the model to predict when to plant new trees and how quickly the supply will respond. Another insight of the economic framework is that CLT will actively compete with

traditional wood products for wood[13]. As wood prices increase, quantities of traditional wood products will fall. Consequently, there will be a smaller net increase in overall wood demand than previous studies projected[6,12].

The net effect that future CLT will have on direct GHG emissions traditionally captured by LCA will be relatively small. For the production and end-of-life stages, the combination of the increased GHG emissions by CLT and the reduced GHG emissions by traditional wood products leads to a cumulative increase of 1.6–2.4 $GtCO_2e$ by 2100. When also considering the substitution of steel and concrete and the use of wood waste to generate electricity, the net production and end-of-life GHG emissions become a reduction of 1.9–9.7 $GtCO_2e$. This is 7.1%–24.7% of the total net cumulative GHG emissions, depending on the scenarios.

The largest carbon consequence of CLT adoption happens in the forest. Higher wood prices encourage landowners to grow more forests and invest in growing trees faster. By 2100, global plantations increase by 11.0–14.7 Mha; managed forestland increases by 27.9–40.8 Mha; but natural forestland decreases by 8.1–18.9 Mha. The combination of these changes in the forest leads to an overall increase in forestland and forest carbon. Global forest carbon stock increases by 16.1–17.7 $GtCO_2e$. The carbon in wood products and end-of-life sites increases by 3.2–7.4 $GtCO_2e$. By 2100, total carbon stock increases by 20.3 $GtCO_2e$ in the medium demand-fast adoption scenario, by 20.9 $GtCO_2e$ in the medium demand-slow adoption scenario, and by 25.2 $GtCO_2e$ in the high demand-fast adoption scenario. The total cumulative net life-cycle GHG emission reduction is 27.0 $GtCO_2e$, 25.6 $GtCO_2e$, and 39.0 $GtCO_2e$ for the fast, slow, and high scenarios, respectively. The net increase in carbon storage, including in forests (standing trees, soil, slash), in CLT, in traditional wood products, and in landfills, is 75%, 82%, and 65% of the total net GHG emission reduction for Scenarios 1, 2, and 3, respectively.

The cumulative GHG emission reductions are more modest than the projections of an earlier study that suggested 106 $GtCO_2e$ by 2100[12]. There are three major reasons for this difference. First, we simulate lower CLT demand scenarios with a CLT adoption rate of 60% of urban construction in our highest estimate, compared to 90% of construction being in wood buildings[12]. Second, we exclude many non-forestland types (e.g., savannah and parkland) because they are unsuitable for growing timber. Third, GTM recognizes that converting prime cropland into plantations is very costly. If plantations require high-productivity cropland, vast plantations will substantially reduce the world's ability to grow food. This would likely be expensive and increase the incentive to convert natural forestlands to farmland.

The results of this study agree with the literature that adopting CLT will lead to net beneficial carbon consequences[7,8,12]. Using CLT instead of steel and cement will increase wood demand and encourage the market to protect more forestland and intensify forest management, which ultimately increases forest carbon stock. Our study suggests there will be a large net increase of carbon stock in the forest and in CLT panels, although some carbon stock will be lost because of the declining production of traditional wood products.

The results suggest that it would be reasonable to subsidize the demand for CLT as a GHG mitigation strategy. By increasing the market value of timber, the subsidy would provide an additional market incentive to maintain the world's existing forests. Given the difficulties of regulating global land use, this subsidy would be a practical tool to increase carbon storage in forests. There are also small gains from reduced GHG emissions and storage of carbon in long-lasting wood products. Encouraging CLT adoption will increase forestland by 30.7–36.5 Mha, especially in the long run, and reduce carbon in the atmosphere.

Despite the large overall increase in forestland, GTM predicts natural forestland will fall by 8.1–18.9 Mha. There is a tradeoff between conservation and carbon goals in these scenarios. Widespread expansion of CLT will likely reduce natural habitat. These land use effects will vary across the planet, with substantially increased forestland in the United States and Western Europe, but large reductions in natural forestland in the tropics. Stricter protections of natural lands will reduce this side effect on conservation, but the regulations will also increase the cost of CLT and reduce the carbon stored. There is, consequently, a limit to how far one should expand CLT and timber production in general. The CLT scenarios we present in this paper are sustainable, but increasing subsidies further may not be. Even with the scenarios we present, moving from medium demand to high demand has additional benefits. Future research should look at these tradeoffs carefully and make clear what the consequences of alternative policies will be. There are tradeoffs between conservation and mitigation policies that affect land use. It is critical that policymakers be aware of those consequences.

From now through 2100, alternative assumptions about the end-of-life cases of CLT panels have only minor impacts on the results. The long life span of CLT panels implies that few panels will reach end-of-life in this century. With future improved circularity of mass timber products beyond 2100[23], this could potentially reduce the timber demand for CLT, and further impact the forest carbon stock and timber supply. Future research could investigate how alternative circularities of CLT affect the global carbon consequences of adopting CLT in the long term. At the same time, alternative recycling rates of CLT could also impact the product-level life-cycle GHG emissions of CLT panels by affecting the upstream GHG emissions. But this depends highly on the allocation methods.

GTM focuses on capturing the long-term forest sector adjustments that evolve over time in response to long-term demand or supply stimulus. This study adopted a unitary demand price elasticity of −1.0. The literature reports a range of demand price elasticities for IWP, with the unitary elasticity on the high end[24–26]. A more price inelastic demand for traditional wood products would lead to larger wood price increases over time and therefore a larger supply response, including more forestland and higher management intensity. This would lead to more carbon storage in forests. However, if traditional wood products have a lower price elasticity, increases in CLT would cause less substitution with traditional wood products. Traditional wood products would shrink much less than we estimated in the high CLT scenarios. The overall result is that CLT would still lead to substantial carbon benefits, but an even larger fraction of those savings would come from the forest itself. Additionally, if the CLT demand is even higher than the high demand scenario with the same fast adoption and elasticity, it can be anticipated that the net carbon benefits are likely to grow with higher forest and CLT carbon stocks, lower production and EOL GHG emissions of traditional wood products, and higher potential substitution benefits. Moreover, some market dynamics are not included in this study that might affect the competitiveness of CLT relative to other materials. For instance, we show that as CLT competes with other timber products, it will drive up average global prices of timber. If CLT becomes too expensive, it may face more difficulty replacing steel and cement (assuming no carbon prices will be applied to them). It would be attractive to explore this more carefully using an economy-wide economic model to assess these dynamic tradeoffs.

Finally, in this study, only CLT is considered because it is the most recognized mass timber product. Other emerging wood products are not yet included. However, as these products evolve, the framework and methodology presented by this study can be adapted to include them. Based on this study, future research can further investigate possible improvements in background processes (e.g., electricity generation, steel recycling rate, and concrete production) and foreground processes (e.g., CLT production) in each region. Moreover, future research efforts can investigate the effects of other drivers (economic growth and future climate change scenarios).

## Methods

### General methods

We develop an integrated framework that combines the GTM (an economic forest model) and LCA models to assess the carbon consequences of adopting emerging mass timber products from 2020 to 2100. We use this framework to examine three CLT adoption scenarios and compare the results with a baseline scenario without CLT. The middle of the road socioeconomic projections of future economic, population, and urbanization outcomes, SSP2[19], are the inputs for projecting CLT demand and GTM. The analysis examines three projections of CLT demand from 2020 to 2100: a medium-slow, a medium-fast, and a high-fast projection. GTM estimates how farmland, forestland, and forest management change and forest carbon stock change in response to the added CLT projection. GTM also calculates how total wood output is split across CLT and traditional wood products. The LCA calculates the carbon stored in all the wood products throughout their life cycle. The LCA also estimates both the fossil-based and the biogenic GHG emissions of the production of these wood products based on the process-based models, as well as the potential substitution benefits of CLT replacing steel and cement based on the average data from the whole building LCA literature[27]. The integrated analysis consequently captures the relevant changes in carbon flows from carbon storage in forests, products, and end-of-use pools as well as the direct emissions and avoided emissions caused by CLT.

### Global Timber Model

This analysis uses an economic-ecological forest sector model, the GTM[28]. GTM determines the level of management intensity (e.g., thinning, planting, intensive thinning, clear cutting, natural regeneration, fertilization) together with harvesting rates for each period to maximize the present value of timber market surplus over the next 100 years, given future expected demand for timber products, including CLT. The model evaluates the global land in forest ecosystems. This includes boreal forest, temperate deciduous forest, temperate coniferous forest, tropical moist forest, and tropical dry forest. The model does not include tropical grassland, tropical savannah, wetlands, temperate grassland, desert, or tundra. These alternative ecosystems do have trees[29], but the biomass density of the trees is no more than a third of the biomass of forests[30]. These non-forest ecosystems are not considered because they cannot support timber production without expensive inputs such as irrigation in dryland.

GTM is a dynamic model that solves how to maximize the value of timber production over time, given the future demand function for timber and the existing stands of trees of each age class in each forest ecosystem. There are three forest types in the model: natural forest, plantations, and managed forest. The forest model has ~350 wood supply regions across the world that reflect the different forests in each region. Information about where forests can grow, natural forest productivity, and natural average biomass comes from an ecological model, BIOME/LPX-Bern[31]. The managed forest and plantations are divided into age classes. The acreage by age class comes from historic harvest patterns since 1900[32]. The model determines the level of management intensity, planting, and harvesting that maximizes the present value of wood harvests over the next 200 years, given future demand. Because GTM tracks forest inventory very carefully, it has been able to reproduce the historic levels of forest carbon over the last century[33]. In this paper, we do not project the impact of climate change on future forests, although the model has been used to examine the impact of climate change in the past. This is a topic that can be addressed in future research. Recent analysis with GTM indicates that future market and land use projections are robust to parametric uncertainty related to forest growth and land supply parameters[34].

The model's optimization problem is formally written as in Eq. (1):

$$
max \sum_0^\infty \rho^t \left\{ \begin{array}{l} \int_0^{Q_t^{tot}} \left\{ D\left(Q_t^{ind}, Z_t\right) + D\left(Q_t^{CLT}\right) - C_H\left(Q_t^{tot}\right) \right\} dQ_t^{tot} - \\ \sum_i C_G^i\left(m_t^i, G_t^i\right) - \sum_i C_N^i\left(m_t^i, N_t^i\right) - \sum_i R_t^i\left(\sum_a X_{a,t}^i\right) \end{array} \right\}
$$
(1)

where $\rho^t$ is the discount factor, $Q_t^{tot}$ is total wood harvest, $Q_t^{ind}$ is wood for traditional wood production (pulp plus sawtimber), $Q_t^{CLT}$ is the wood needed for CLT production, $D(Q_t^{ind}, Z_t)$ is the global demand function for traditional wood products $D(Q_t^{ind}, Z_t)$, $Z_t$ is global GDP per capita[16], $C_H^i$ is the cost of harvesting and transporting wood to the mill, $C_G^i$ is the cost of planting to management intensity, $m$, of $G_t$ hectares of forest type $i$ (e.g., plantation, managed, natural), $C_{PL}^i$ is the cost of planting new forests, $C_N^i(m_t^i, N_t^i)$ is the cost of converting additional natural forestland, $N_t^i$, to new managed forestland, and $R_t^i(\sum_a X_{a,t}^i)$ is the opportunity cost of farmland. The opportunity cost of farmland is a function of the total area of forestland, $\sum_a X_{a,t}^i$. As more farmland is devoted to forestland, the opportunity cost rises to reflect the underlying inelastic price of food. The notations used in this study and corresponding descriptions are available in Supplementary Table 6.

Long-run demand has the following functional form: $Q_t^{ind} = A_t(Z_t)^\theta P^\omega$ where $A_t$ is a constant, $\theta$ is the income elasticity (0.87), and $\omega$ is the price elasticity (−1.00)[25,26,35]. Total industrial wood demand incorporates separate demand functions for sawtimber and pulpwood.

Equation (2) shows that the total quantity of wood harvested depends upon the area of land harvested in the timber types $i$ for each age $a$ and time $t$ ($H_{a,t}^i$) and the yield ($V_{a,t}^i$) which is a function of age, ecological forest productivity $\theta_t^i$ and management intensity $m_{t0}^i$.

$$
Q_t^{tot} = Q_t^{ind} + Q_t^{CLT} = \sum_i \left( \sum_a H_{a,t}^i V_{a,t}^i \left( \theta_t^i, m_{t0}^i \right) \right)
$$
(2)

The amount of land in each forest type that adjusts over time is calculated according to Eq. (3):

$$
X_{a,t}^i = X_{a-1,t-1}^i - H_{a-1,t-1}^i + G_{a=0,t-1}^i + N_{a=0,t-1}^i
$$
(3)

The initial stocks of land $X_t^i$ are based on FAOSTAT data[36], and all choice variables are constrained to be greater than or equal to zero, and the area of wood harvested $H_{a,t}^i$ does not exceed the total timber area. $G_t^i$ is the area of forestland regenerated planting, and $N_t^i$ is new forestland.

GTM assumes there is an international market for timber that leads to a global market-clearing price. As the price of wood for bioenergy rises to compete with industrial timber, both timber and bioenergy are traded internationally[37]. Competition for supply equilibrates their prices.

GTM is programmed into GAMS and solved in decadal time increments using the MINOS solver. Terminal conditions are imposed on the system after 200 years, far enough into the future so as not to affect the study results over the period of interest (2020–2100).

In GTM, the forest carbon stock is measured as the sum of carbon stock in three different carbon pools: aboveground, soil, and slash carbon. Aboveground carbon $C_{a,t}^i$ accounts for the carbon in all components of the aboveground living tree, as well as carbon in the forest understory and the forest floor, but does not include dead organic matter in slash, which is contained in a separate pool. For this analysis, we assume that carbon is proportional to total biomass, such

that carbon in any forest of any age class is given in Eq. (4) as follows:

$$C_{a,t}^i = \sigma^i V_{a,t}^i (m_{t0}^i) \tag{4}$$

where $\sigma^i$ is a species-dependent coefficient that converts biomass to carbon. Given this, the total forest aboveground biomass carbon pool $TFCP_t^i$ for each timber type is calculated based on Eq. (5):

$$TFCP_t^i = \sum_a C_{a,t}^i X_{a,t}^i \tag{5}$$

Soil carbon includes carbon stored in mineral and organic soils (including peat). GTM models changes in soil carbon storage from forestland use change, but does not capture nuanced soil carbon dynamics associated with forest operations. Soil carbon $SOLC_t^i$ is measured as the stock of carbon in forest soils of type $i$ in time $t$. The value of $\bar{K}$, the steady state level of carbon in forest soils, is unique to each region and timber type. The parameter $\mu^i$ is the growth rate for soil carbon. In this analysis, we capture the marginal change in carbon value associated with management or land use changes. When land use change occurs, we track net carbon gains or losses over time as shown in Eq. (6):

$$SOLC_{t+1}^i = SOLC_t^i + SOLC_t^i (\mu^i) \left[ \frac{\bar{K} - SOLC_t^i}{SOLC_t^i} \right] \tag{6}$$

Finally, slash carbon $AS_t^i$ measures carbon stored in slash that remains on site, resulting from wood harvesting operations, as shown in Eq. (7).

$$AS_t^i = \sum_a \left( C_{a,t}^i H_{a,t}^i - \kappa^i V_{a,t}^i H_{a,t}^i \right) \tag{7}$$

Over time, the stock of slash $SP_t^i$ builds up through annual additions, and decomposes as displayed in Eq. (8):

$$SP_{t+1}^i = AS_t^i + \left( 1 - \vartheta^i SP_t^i \right) \tag{8}$$

Decomposition rates $\vartheta^i$ differ, depending on whether the forest lies in the tropics (3% year$^{-1}$), temperate (5% year$^{-1}$), or boreal zone (7% year$^{-1}$)[16].

Total forest carbon stock in each region $n$ (see Supplementary Data 1 for the region definitions in the GTM) at time $t$ is calculated by Eq. (9):

$$C\_GTM_{t,n} = \sum_i \left( TFCP_{t,n}^i + SOLC_{t,n}^i + SP_{t,n}^i \right) \tag{9}$$

Such that if $CGTM_{t,n} > CGTM_{t+1,n}$ forests in region $n$ are releasing emissions at time $t+1$ because forest carbon stock is declining, while if $CGTM_{t,n} < CGTM_{t+1,n}$ more sequestration is occurring.

## Life Cycle Assessment

In this study, a cradle-to-grave dynamic consequential LCA is developed to quantify the life-cycle carbon flows of forest and product systems due to the adoption of CLT (see Fig. 1 for system boundary and summarized carbon flows). The carbon consequences consider the global and regional carbon flow changes compared to the baseline without CLT adoption. The carbon consequences consist of two aspects, namely direct carbon flows and indirect carbon flows. Equations (10) and (11) show the total carbon stock change compared to the baseline in year $t$ and region $n$, $\Delta Total\_C_{t,n}$, and the total GHG emission change compared to the baseline in year $t$ and region $n$, $GHG_{Totalt,n}$, respectively. For Eq. (10), the direct carbon stock changes

include the changes in carbon stored in CLT products ($\Delta CLT\_C_{t,n}$) and their end-of-life sites ($\Delta CLT\_EOL\_C_{t,n}$). The indirect carbon stock changes include two components, forest carbon stock changes ($\Delta CGTM_{t,n}$) (given by GTM) and carbon stock changes in traditional wood products ($\Delta TWP\_C_{t,n}$). Forest carbon pools, as mentioned above, include aboveground biomass, slash, and soil carbon pools. Traditional wood products in this study include sawtimber products (e.g., lumber, particle boards) and pulpwood products (e.g., pulp and paper products), but exclude mass timber products. For Eq. (11), $Forest\_Seq_{i,t}$ is the total net forest biogenic carbon flow from the atmosphere compared to the baseline in year $t$ and region $n$, and evaluated by Eq. (12). $Forest\_Seq_{i,t}$ equals -44/12 multiplied by the sum of forest carbon stock changes ($\Delta CGTM_{t,n}$), forest output carbon for CLT ($\Delta TimberC_{t,n}^{CLT}$), forest output carbon of sawtimber ($\Delta TimberC_{t,n}^{Sawtimber}$), and forest output carbon of pulpwood ($\Delta TimberC_{t,n}^{Pulpwood}$). Given the mass balance, input carbon equals the stock change and output. $GHG_{Forest t,n}$ is the corresponding life-cycle GHG emissions by forest operations (e.g., harvesting, planting). $GHG_{CLT t,n}$ describes the GHG emissions by producing and using CLT across the life-cycle stages of the CLT system, including CLT production, end-of-life, and corresponding material substitutions. The carbon flows of CLT production are evaluated based on the process models (see the sections below). The upstream burdens of producing fuels, chemicals, and electricity, and the combustion of fuels are also included in this study. $GHG_{TWP t,n}$ describes the life-cycle GHG emissions of the production and end-of-life of traditional wood products. The Intergovernmental Panel on Climate Change (IPCC) AR6 GWP-100 factors are used to convert GHG emissions (including $CO_2$, $CH_4$, $N_2O$) to $CO_2$-equivalent basis[1].

$$\Delta Total\_C_{t,n} = (\Delta CLT\_C_{t,n} + \Delta CLT\_EOL\_C_{t,n}) + (\Delta C\_GTM_{t,n}) + (\Delta TWP\_C_{t,n}) \tag{10}$$

$$GHG_{Total t,n} = Forest\_Seq_{t,n} + GHG_{Forest t,n} + GHG_{CLT t,n} + GHG_{TWP t,n} \tag{11}$$

$$Forest\_Seq_{t,n} = -\frac{44}{12} \times (\Delta C\_GTM_{t,n} + \Delta TimberC_{t,n}^{CLT} + \Delta TimberC_{t,n}^{Sawtimber} + \Delta TimberC_{t,n}^{Pulpwood}) \tag{12}$$

## GHG emissions of CLT production and end-of-life

CLT production from sawtimber includes two major steps: (1) lumber production from sawtimber; and (2) CLT production from lumber. To produce lumber, sawtimber is transported to lumber mills after harvesting. Producing lumber includes sawing, kiln drying, planing, and energy generation for the dry kiln[38,39]. The upstream GHG emissions were derived from the Ecoinvent 3.9 cut-off database[40] (see Supplementary Table 7) and GREET 2022[41]. The mill residues (bark, sawdust, shavings, and chips) are combusted to produce energy for the kiln. If excessive, the rest is recovered for power generation. The byproducts of lumber mills, slabs, and chips from sawing are also recovered for power generation. More details are available in Supplementary Note 1 and Supplementary Table 8. Then lumber is made into CLT panels. The first unit operation, lumber preparation, ensures the lumber quality for CLT production, including grading, grouping, and moisture detecting[9,42]. Then the lumber is longitudinally end-jointed to make long continuous lumber, and layered and glued for face bonding[11]. With pressing, the layered and glued lumber layers form the CLT panels. Then the CLT panels are planned to remove uneven surfaces. The final step is end cutting to output CLT panels in customized shapes. The waste from CLT production is sent to landfill sites. More details are available in Supplementary Note 2. The LCI data of CLT production are collected from the literature and shown in Supplementary Table 8. After the life span of CLT, assumed 60 years[43], the

CLT panels are demolished and the discarded CLT waste is sent to the landfill site as end-of-life. The GHG emissions from landfills are estimated based on the IPCC First Order Decay (FOD) method for landfill[11,44]. Since landfill gas contains a high-volume fraction of $CH_4$, the energy recovery from landfill gas is considered. The details of landfilling are shown in Supplementary Note 3 and Supplementary Table 9. Besides the landfill case, two additional end-of-life cases of CLT panels are conducted: (1) material recycling case with 50% closed-loop recycling and 50% landfilling; (2) energy recovery case with 100% CLT panels combusted for power generation. These robustness checks explore how these alternative assumptions change the results. More details are available in Supplementary Note 2.

Since CLT as a mass timber product can substitute the traditional reinforced concrete and steel in buildings (life span assumed as 60 years[45,46]), this study calculates the potential substitution benefits of CLT to replace the conventional structural materials by providing the same floor area[27]. The details related to the material usage of the CLT and traditional building structures are available in Supplementary Note 2 and Supplementary Table 10. Hence, the GHG emissions directly caused by producing and using CLT across the life-cycle stages ($GHG_{CLT}$ in Eq. (11)) are quantified by Eq. (13).

$$GHG_{CLT\,t,n} = GHG_{CLT\_production\_EOL\,t,n} + GHG_{Sub\_electricity\,t,n} + GHG_{Sub\_building\,t,n} \quad (13)$$

$GHG_{CLT\_production\_EOL\,t,n}$ are the GHG emissions (including both fossil and biogenic) related to CLT production and end-of-life in year $t$ and region $n$, including timber production, CLT production, landfilling, and landfill gas recovery; $GHG_{Sub\_electricity\,t,n}$ the potential substitution benefits of recovering the mill byproduct for power generation; $GHG_{Sub\_building\,t,n}$ the potential substitution benefits of replacing traditional building materials.

### GHG emissions from the production and end-of-life of traditional wood products

To quantify the carbon stock changes in traditional wood products converted from the timber produced at the end of year $t$ and region $n$ compared to the baseline ($\Delta TWP\_C_{t,n}$), and the corresponding manufacturing GHG emissions of traditional wood products ($GHG_{TWPt,n}$), this study combines the manufacturing GHG emission data for traditional wood products collected from the literature and database[40] and the FOD method for traditional wood products used in the report, 2019 Refinement to the 2006 IPCC Guidelines for National Greenhouse Gas Inventories, by the IPCC[47,48]. As shown in Eq. (14)[47], $\Delta TWP\_C_{t,n}^l$ is the traditional wood products converted from the timber class $l$ ($l$ belongs to {sawtimber, pulpwood}) produced in year $t$ and region $n$ compared to the baseline; $k$ is s a decay parameter calculated based on average half-life of timber class $l$[48] (Eq. (16)); $\Delta TWP\_In_{t,n}^l$ is the carbon input of wood products made from timber class $l$ produced in year $t$ and region $n$ compared to the baseline. $\Delta TWP\_In_{t,n}^l$ is evaluated by Eq. (15) where $\Delta TimberC_{t,n}^l$ is the carbon input of wood class $l$ produced in year $t$ and region $n$, compared to the baseline (namely $\Delta TimberC_{t,n}^{Sawtimber}$ or $\Delta TimberC_{t,n}^{Pulp}$ in Eq. (12)), and $f_m^l$ is the factor describing how much biogenic carbon left in wood products after manufacturing. $\Delta TimberC_{t,n}^l$ is given by GTM as mentioned above. The value of the parameters and more details are available in Supplementary Note 4 and Supplementary Table 11.

$$\Delta TWP\_C_{t+1,n}^l = e^{-k_l}\Delta TWP\_C_{t,n}^l + \left[\frac{(1-e^{k_l})}{k_l}\right]\Delta TWP\_In_{t+1,n}^l \quad (14)$$

$$\Delta TWP\_In_{t,n}^l = \Delta TimberC_{t,n}^l \times f_m^l \quad (15)$$

$$k_l = \frac{\ln(2)}{HL^l} \quad (16)$$

GHG emissions from traditional wood products include two components, as shown in Eq. (17). The first component is the biogenic and fossil GHG emissions related to manufacturing. The term $\frac{44}{12} \times \Delta TimberC_{t,n}^l \times (1 - f_m^l)$ describes the manufacturing biogenic carbon release of producing these wood products; the term $EF^l \times \Delta TimberC_{t,n}^l$ describes the manufacturing fossil GHG emissions of producing these wood products by using the average emission factor $EF^l$. $EF^l$ is estimated based on the fossil GHG emissions for different wood products from literature and averaged by the global wood products share[40,48]. The value of the parameters and more details are available in Supplementary Note 4 and Supplementary Table 11. The second component $\frac{44}{12} \times ((\Delta TWP\_C_{t,n}^l + \Delta TWP\_In_{t,n}^l) - \Delta TWP\_C_{t+1,n}^l)$ describes the total biogenic carbon pool loss of the wood products made from timber class $l$ produced in year $t$ and region $n$ compared to the baseline.

$$GHG_{TWPt,n}^l = \frac{44}{12} \times \left((\Delta TWPC_{t-1,n}^l + \Delta TWPIn_{t,n}^l) - \Delta TWPC_{t,n}^l\right) + \frac{44}{12} \times \Delta TimberC_{t,n}^l \times (1 - f_m^l) + EF^l \times \Delta TimberC_{t,n}^l \quad (17)$$

### Projections and scenarios of cross-laminated timber demand

The sawtimber demand caused by adopting CLT ($\Delta TimberDemand_{t,n}^{CLT}$) is projected from 2020 to 2100, as shown in Eq. (18). In Eq. (18), $NewA_{i,t}$ is the annual newly constructed urban commercial and residential building floor areas that adopt steel and concrete structures in year $t$ and region $n$ and can be potentially replaced by CLT; $r_t$ is the adoption rate (%) of CLT (or say the percentage of newly constructed areas that are built with CLT); $f_{CLT}$ is the CLT usage factor ($m^3$ CLT per $m^2$ building area, see Supplementary Table 10)[27]; $c_{CLT}$ is the conversion factor for CLT from timber ($m^3$ CLT per $m^3$ wet timber) that is evaluated based on the process model mentioned above. $NewA_{i,t}$ from 2020 to 2100 in this study, is derived from the total building floor areas of urban commercial and residential buildings based on the multi-variable regression models for each region that assumes total building floor areas depend on GDP per capita and urban population[20,49]. The historical data of urban residential and commercial building floor areas from 1970 to 2010 are derived from the study by Deetman et al. for training the regression model[49]. The projections of GDP per capita and urban population follow the SSP2 scenario[19]. The details of the regression model and method are shown in Supplementary Note 5 and Supplementary Table 12. In this study, $r_t$ is assumed to follow the technology diffusion curve by using a logistic model (see Supplementary Note 5)[50,51]. Three different adoption scenarios are established to explore the impacts on the carbon consequences[27]: (1) medium and fast adoption: 30% adoption reached by 2050; (2) medium and slow adoption: 30% adoption reached by 2080; (3) high and fast adoption: 60% adoption reached by 2050. More details are shown in Supplementary Note 5 and Supplementary Data 1.

$$\Delta TimberDemand_{t,n}^{CLT} = (NewA_{t,n} \times r_t \times f_{CLT})/c_{CLT} \quad (18)$$

### Data availability

The data that support the findings of this study are all available from the main text, supplementary information, and supplementary data. Source data are provided with this paper.

### Code availability

All the code used in this study is available at Zenodo: https://doi.org/10.5281/zenodo.13334682.

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

## Acknowledgements

The authors thank the funding support from Yale University, Yale Planetary Solutions, and Yale Center for Natural Carbon Capture. K.L. thanks the funding support from North Carolina State University.

## Author contributions

Y.Y. and R.M. designed the idea. Y.Y. and R.M. supervised this study. A.F. generated the results of the Global Timber Model. K.L. and H.S.W. projected the wood demand and conducted the LCA. K.L. and H.S.W. visualized the results. K.L., A.F., R.M., Y.Y., and H.S.W. contributed to writing and finalizing the manuscript. Y.Y. and R.M. contributed equally to this work.

## Competing interests

The authors declare no competing interests.
