## [Transparent Peer Review file · Nature Communications]

Global Land and Carbon Consequences of Mass Timber Products

Corresponding Author: Professor Yuan Yao

Version 0:

Reviewer comments:

Reviewer #1

(Remarks to the Author)

The manuscript presents an integrated framework that combines a global timber model and a life cycle assessment model to evaluate the carbon and land impacts of adopting cross-laminated timber (CLT). The main findings indicate that CLT adoption can increase carbon storage in forests and wood products, reduce carbon emissions by substituting traditional building materials and electricity.

The manuscript is well-written, concise, and the results are interesting, potentially relevant for decision-making. However, the manuscript has some weaknesses in its methods and analysis that need addressing. These are discussed as follows:

1. Existing CLT buildings have not yet reached their mid-service life, and a potential future development could involve increased volumes of post-use CLT. In the End-of-Life (EOL) stage, the manuscript considers only landfilling. Given that massive wood-based products like CLT have a high heating value, the EOL stage, with its benefits and burdens, significantly impacts the overall greenhouse gas (GHG) performance of such building product and system from a life cycle perspective. Therefore, it is crucial to analyze the implications of more progressive EOL management options, such as reuse, recycling, cascading, and energy recovery, in addition to the landfilling. This can have significant impact on estimated carbon flows.
2. Among the different datasets and databases in Ecoinvent 3.7, the cut-off database was used to calculate carbon emissions. However, is this the most appropriate database for a consequential life cycle assessment (LCA) as aimed for in the paper? For a consequential LCA, the Ecoinvent 3.7 database for a consequential system model (long-term) may be more suitable. It is critical that appropriate dataset is used in showing the carbon consequences of the scenarios.
3. The authors should provide more details on how they projected the CLT demand scenarios, especially the assumptions regarding market penetration rates, building floor area, and wood species mix. How sensitive are the results to these assumptions?
4. The paper does not explicitly address uncertainties in data, CLT production technology, and their quantitative impacts on the calculated results. The authors should discuss the limitations and uncertainties of their modeling framework, which include the simplification of soil carbon dynamics and the variability in LCA, particularly since CLT technology is relatively young. For example, ongoing research is investigating open issues to further improve CLT technology in areas such as material efficiency and the development of climate-friendly alternatives to adhesives. These ongoing developments can significantly impact the assumptions made in the study and the overall findings. Therefore, the paper should discuss uncertainties, and the results should highlight sensitivities to variations in different data inputs, methodological choices, etc.

Reviewer #2

(Remarks to the Author)

Authors have used a combined forest economics model (Global Timber Model i.e., GTM) and an LCA model to explore the potential environmental impacts of using CLT in future building construction scenarios. Their findings suggest that

widespread CLT adoption could lead to a significant increase in carbon storage (up to 7 GtC) in standing forests and harvested wood products (including CLTs). Additionally, they estimate a reduction in greenhouse gas emissions of approximately 40 GtCO_{2e} due to improved biogenic carbon uptake and lower emissions associated with HWP production and disposal.

This paper was an excellent read, offering a comprehensive analysis of the environmental consequences of leveraging future timber buildings using CLTs for GHG mitigation. The authors innovatively combined a well-known forest economics model (GTM) with a comprehensive LCA analysis, a unique approach I have not encountered in any other existing study. This undertaking deserves high praise, especially considering how it highlights critical research gaps in current literature.

While I acknowledge the expertise of the co-authors and my own limited experience with an older version of GTM, I believe that open-sourcing both the GTM code and the employed LCA model would significantly enhance the transparency and replicability of this study. This would allow other researchers to verify the results, build upon them, and address potential limitations in future studies.

The research presented here is innovative, but I have significant concerns regarding both the chosen tools and the authors' justification for the study. Notably, the authors provide minimal justification for selecting GTM as a suitable model for addressing their research question. GTM is inherently not a land-use model and only simulates specific forest land pools. Additionally, GTM lacks the ability to model competition for land between agriculture, forestry, and other land uses. This significantly weakens the land-use arguments presented in the paper. Furthermore, the authors rely heavily on citing papers written among themselves, necessitating constant referencing back and forth between multiple publications. While they may have extensive background knowledge informing this study, the manuscript lacks sufficient context for neutral readers to fully grasp the results.

Most of the results are presented in relation to the baseline but the manuscript lacks explicit baseline results (excluding Figure 2, where it appears unnecessary, and Figure 3 does have baseline results). This omission hinders our ability to assess the magnitude of differences across the three scenarios compared to the baseline scenario itself. Consequently, we cannot evaluate the performance of the presented modeling framework on the baseline scenario. This manuscript's writing style primarily reflects a modeling study approach, potentially limiting its accessibility to the broader audience of Nature Communications. Given the lack of broader appeal and the presence of methodological concerns, I would not be able to recommend this manuscript for publication in Nature Communications.

I have attached my detailed and minor comments in the file attached which hopefully are of use to the authors. Detailed comments start from page 2 of the attached document.

Reviewer #3

(Remarks to the Author)

Review

The manuscript presents an interesting study of potential future carbon sequestration in construction by using cross-laminated timber for mid- and high-rise buildings. It investigates the carbon storage in forests and buildings and considers partial-equilibrium timber market effects as well as life-cycle carbon impacts.

The manuscript is well written, and the results are clearly communicated. However, some aspects are not explained.

1. The largest carbon savings are from carbon storage in land converted to forestry given the increased wood prices. However, it appears as if the land is treated as having no carbon before, or in an alternative scenario (Fig.5). This appears problematic. A short description of land-use related carbon storage before and after afforestation would be required.
2. The demand for CLT is driven by the demand for floor area. However, it is not explained how this demand is derived. Is it based on a cohort-based stock-flow model for buildings, as they become increasingly common in industrial ecology, or just on a correlation with per-capita GDP?
3. The supply of timber is somewhat surprising (Fig 4). There is no common pattern across the scenarios. It seems like initially, most timber comes from Europe in Scenario 1 and none in Scenario 2, while the situation for Brazil is vice versa. There is no explanation for why this pattern persists. It appears that the only difference between the scenarios is in the demand for CLT, or did I miss something? The text explains that there is international trade but does not explain whether there are any transport costs assumed (transport would also be source of emissions). It would be nice to know where the demand occurs and these CLT buildings will be constructed. With respect to demand, I see it as strange that the per-capita floor area for urban commercial buildings is assumed to remain the same while the GDP rises substantially. Also, the assumption is that few buildings today are with steel and concrete. I have no statistics, but at least for East Africa and Nigeria, almost all urban buildings are of concrete. The forest area coverage changes in Fig.4 do not reflect the patterns of harvest in Fig.2, which is also peculiar.

Minor comments:

- Fig. 1: End-of-life for HWP considered?
- Fig. 1: How is the carbon balance of the forests considered? Here, the uptake of the forest is only balanced by wood material flows. How about biomass which degrades in the forest?
- I recommend setting up a table where variables are defined, as part of the method section. The equations are hard to read as it is.
- The Excel table should include information on sources, where appropriate. I would also recommend including tables for the information displayed in figures.

Version 1:

Reviewer comments:

Reviewer #1

(Remarks to the Author)

The paper has undergone significant improvements from its previous version. However, the authors still need to elaborate on and address critical issues regarding the life cycle analysis. These points are discussed as follows:

- Some underlying assumptions in the paper appear to bias the results in favor of CLT construction and warrant re-examination. Specifically, the recycling rate for steel is assumed to be 35% based on ref.11, with the remaining steel being landfilled. This information contradicts what is reported in the cited study (refer to page 8, paragraph 4). The literature cited actually indicates recycling rates for steel construction materials ranging from 65% to 98%, significantly higher than the 35% used in the present analysis. The 35% recycling rate is notably low for steel, as evidenced by current data from the steel industry. The steel industry's information (e.g. <https://buildsteel.org/why-steel/sustainability/recycles-steel-scrap/>; <https://cssbi.ca/mid-rise-construction/sustainable-steel>; https://www.bir.org/images/BIR-pdf/Ferrous_report_2017-2021_Ir.pdf) points to higher recycling rates for steel. It is essential to consider the implications of a higher recycling rate for steel, especially in contexts where materials recovery is expected to become more prevalent in the future.
- The authors should provide clarification on how they modeled the recycling of CLT in a scenario involving 50% recycling and 50% landfilling. They need to explain what CLT is recycled into or used for and detail the type of data or Ecoinvent unit processes used to assess the benefits and burdens of CLT recycling.

Reviewer #2

(Remarks to the Author)

I appreciate the authors' efforts in providing additional background details for their manuscript. However, I still have reservations about the suitability of the chosen modeling framework for addressing the research question at hand. While GTM is a robust model, it may not be the most appropriate tool for this specific research question. Despite the authors' detailed responses to my previous review, some of my concerns remain unaddressed.

Regrettably, based on the current state of the manuscript, I am unable to recommend it for publication in Nature Communications.

Attached, I offer further comments based on the authors' responses.

Reviewer #3

(Remarks to the Author)

The authors have addressed the review comments well. They have proactively conducted additional analysis, expanded the supporting information, and published the code and data. This is very commendable. I realize that demand scenarios for buildings have not been a focus of this work. Rather, the authors have taken a prominently published scenario as a starting point. While this is a completely legitimate approach, I am not sure about that scenario and would have liked to see other scenarios explored, in particular ones that address the substantial unmet housing demand in developing countries. But their scenario is legitimate, and it is the authors' choice.

I see this paper has a substantial interest. The central message is that the increase of wood-based construction will lead to a substantial increase in the carbon stock of growing trees which surpasses the carbon stored in the timber. It is an insight that is novel, and it is robust to remaining issues raised by me or the other reviewers. It is important for policy, including policy processes that I am involved in, and so I would welcome the speedy publication of this paper.

Reviewer #4

(Remarks to the Author)

1. This paper was well written overall with deep studies on global CLT supply and demand. There are some comments that require authors' attention.
2. Page 8 Line 156, there is grammar error.
3. The demand increase on CLT would definitely lead to more plantation for saw lumber. However, demand of other wood products (wood studs, composite woods) will still be required. Is this possible that the global market could not provide enough stock to growing CLT demand (assumed in this paper under different scenarios)? Some countries are regulating to limit the harvesting of forest woods.
4. In the building industry, the AEC professionals are working to improve the circularity of buildings by reusing the materials. As per the assumption period in this paper (80 years from 2020-2100), the building designs might push to allow all the CLT materials to be reused, which would lead to less CLT requirements in the future scenarios. This aspect should definitely be considered in the "Discussion" section.
5. Right now, applying mass timber (including CLT) into building design would count the carbon sequestration as biogenic

- carbon. However, if the mass timber materials could be designed to be reused in a few life cycles, then the biogenic carbon might be treated as embodied carbon reductions. Is this a potential scenario to be considered and discussed?
6. The CLT environmental impacts might vary in different countries. Does the author ensure the accuracy of the impact values? Are the harvesting techniques variances and impacts to the soil considered?
 7. In Figure 3, the global prices increase in the next 40 years, how are these trends would help mass timber competing with concrete and steel from a developer's perspective, assuming concrete and steel price stay stable? The policy support factors and implementation strategies should be emphasized here for the decision makers.
 8. The title should be CLT not mass timber products as it does not cover all the other mass timbers.

Version 2:

Reviewer comments:

Reviewer #2

(Remarks to the Author)

Reviewer #4

(Remarks to the Author)

I reviewed this article last time.

The reviewer revised the article appropriately, and I am happy to recommend it for publication.

Reviewer #5

(Remarks to the Author)

While I find this manuscript useful, I see a need for further improvement in three main areas:

1. I am disappointed to see the authors' lack of comprehensive review on the related recent literature. One of the research gaps highlighted in the paper is that prior study didn't evaluate the market impacts of increased mass timber demand. This is not true. Nepal et al. (2021) evaluated global wood products market impacts of increased mass timber demand. You might want to emphasize how your evaluated impacts support or contradict with prior studies instead of claiming that no prior studies have evaluated the market impacts of increased mass timber demand. I believe that your evaluation of mass timber-induced increases on planted forests and intensified management activities (via timber price increases) can be considered a newer contribution, which you can emphasize in explaining how your studies complement prior studies in this topic area. However, the effects of on added planted forests and increased forest management activities, given a modest projected increase in timber price (up to 59% for sawtimber and up to 22.8% for pulpwood by 2100) seems substantial. It is important that the authors review prior studies evaluating the magnitude of increases in planted forests and management intensities due to price increases and clarify whether the magnitude of the large projected effects in this paper is supported by the prior studies. These papers do not necessarily need to be mass timber-induced price increases.

2. Based on my understanding of the several past studies on the related topic (which authors neglected to review), I feel that your estimated total carbon benefit of increased mass timber demand (4 tCO₂e to 7.2 CO₂e per m³ of CLT) is much higher. Apparently, this is because of your large projected forest carbon contribution (45% to 65%) to total estimated carbon benefits that you showed due to increases in planted forests and managed forests. To give you a perspective, for example, Taylor et al. (2023) estimated carbon benefits (carbon stored in CLT plus avoided GHG emissions) of 1.5 tCO₂/m³ of CLT, using data of actual U.S. specific mass timber building projects. Similarly, in another U.S. specific study, without considering such land use effect, Nepal et al. (2024) finds a carbon benefits of 1.1 t CO₂e per m³ of mass timber adopted in new mid to high-rise buildings in the United States. These numbers are comparable to yours without the land use and forest management effects and provide reference points to compare/contrast your findings. Because your substantial estimated carbon benefits of mass timber adoption stem from large increases in planted forests and forest management activities, it is important that authors review and compare what prior studies have found regarding price effects on forest area/forest management intensities (these do not necessarily need to be mass timber specific studies). Your results that forest carbon shares 45% to 69% of total carbon benefits of mass timber adoption implies almost one to one relationship of price increase with planted forest and forest management activities (and hence forest stock/forest carbon), given your projection of up to 59% increases in timber price by 2100. Please corroborate your findings with prior studies on the effects of timber price increases on forest land use/management.

3. A whole building LCA (WBLCA) is an appropriate approach to compare GHG emissions of wood vs alternate buildings. It is not clear if the process-based LCA that you are using, combined with prior literature (e.g. D'Amico et al. 2021) uses WBLCA data. If you are not using WBLCA, then please provide a justification for not using WBLCA approach and how your results would have changed had you chosen to use WBLCA approach. If you use WBLCA, clarify it in the methods section.

Specific comments:

1. Scenario development: Many countries are already producing and consuming CLTs today and they will continue to do so in future. Provide an estimate of the current global level CLT/mass timber production for a context, and also provide a justification of why baseline scenario does not consider any current demand in CLT.

2. 'Introduction' heading is missing.

3.Line 44. This statement is not true. Nepal et al. (2021) considers the effects of increasing wood demand on markets.

4.Line 46. More than these two studies have investigated the global forest impacts of mass timber adoption. See for example, Nepal et al. (2021).

5.Lines 48 to 51. This statement is not true. Nepal et al. (2021) reports that global increases in mass timber demand would not only increase wood prices and therefore reduce quantities of traditional industrial wood products (e.g., lumber, particle boards, pulp and paper products).

6.Line 50 and throughout the manuscript. CLT is also an industrial wood product (IWP). Consider replacing the term industrial wood products (IWP) with traditional wood products.

7.Line 66. What do you mean by reduce market? Market is a broad term; be specific and clarify whether you mean reduced supply, reduced demand, or reduced trade of these traditional products. Also specify whether this result is true at the global or regional level? Some countries may reduce production while other could increase production depending on their comparative advantage.

8.Line 68. Clarify what is meant by increased forest management intensity? What kind and level of silvicultural activity? (e.g., is it thinning only or thinning and fertilization etc.).

9.Line 76. What do you mean by wood going market to IWP? Correct/clarify.

10.Line 77. Future changes in supply does not necessarily increase carbon. Future changes in supply may also decrease carbon, for example, when we supply more wood than we grow from forest. Correct/clarify the sentence.

11.Line 96 to 114. The text and figure here all describe methods. There is nothing about results. These should go to the methods section.

12.Line 114. Correct typo (two periods).

13.Line 131. Correct typo. A space needed before the number "estiamte34.2%".

14.Line 311. Add missing word between the words 'new' and 'will'.

15.Lines 150 to 151. Figure 2. Where are fig a and b? How is Fig 2 title different from its fig 2b title?

16.Lines 171 to 172. This is contrary to what you said above in lines 157-159: "The higher future prices stimulate an increase in the future total wood supply (including sawtimber for IWP, pulpwood, and sawtimber for CLT) (see Supplementary Table 1)".

17.Lines 216-217. The distinction between plantations and managed forests is not clear. It reads that managed forests are also plantations but planted on less productive land. Is it correct? If so clarify.

18.Line 282. How does CLT reduce/halt the extent of future deforestation? By adding new plantation? Clarify.

19.What kind of forest management investments? Specify.

20.Line 326-328. Nepal et al. (2021) also reported market competition for wood between CLT and traditional wood products. They showed how increased demand for wood for CLT can divert use of wood away from traditional wood products.

21.Line 346. Clarify that carbon storage means the sum of carbon sequestered in forests, carbon stored in wood products carbon in use, and carbon stored in wood products in landfills.

22.Line 358. Is comparing your results with only one study enough to assert that your finding agrees with the literature? How does your results compare with other studies (e.g., Churkina et al. 2020, Taylor et al. 2023, Nepal et al. 2024?). I know that these studies do not use the same scenario or same methods as yours but they provide you some reference points to compare and contrast your results.

23.Lines 433. See general comment #3.

24.Line 435. "future baseline scenario". How is a future baseline scenario different from a current baseline scenario? Rephrase for better clarity.

25.Line 436. "input to the modeling". I believe that the ssp2 forecasts of future economic, population, and urbanization outcomes are input to the baseline scenario modeling and the estimated CLT demand are input to the CLT scenario modeling. Rephrase/Clarify.

26.Line 438. Why not consider a low adoption scenario. Why only medium slow, medium fast and high scenarios? See specific comment 1.

27.Line 440. CLT is also IWP. Consider labeling CLT and traditional wood products instead of CLT and IWP. See specific comment 6.

28.Line 444. GHG's or GHGs. Correct.

29.Line 449, 472 and throughout the manuscript. Clarify what is management intensity? What kinds and levels of silvicultural activities you are referring to?

30.Line 476. Add the phrase 'in the past' at the end of the sentence for better clarity. although the model has been used to examine the impact of climate change in the past.

31.Line 485-486. This phrase to describe CG 'the cost planting to management intensity' is not clear. How it is different from the cost of planting CPL?

32.Line 499. The term "stock of land" is unclear I believe you are referring to forest growing stock or forest inventory. If the land is the farmland, then what does the 'stock of land' mean?

33.Line 543. What is consequential LCA and why use it? Why can't the regular (attributional) LCA be used?

34.A brief and clear description of how substitution benefits of replacing traditional building materials is needed in the main text. Details can go to the Supplementary materials. The current supplementary material indicates substitution benefits of replacing traditional building materials were calculated based on D'Amato. Need a bit more details (e.g., boundary of LCA used, the estimated substitution factor (e.g., xx kg/m³ etc.), whether WBLCA was used to derive the substitution benefits etc.).

35.Lines 592. Clarify that the assumed life span of CLT buildings are 60 years, after which buildings are demolished and CLT panels from demolished buildings are sent to landfills or recycled or both. Also clarify the life span of steel and concrete buildings. These are important pieces of information and should be included in the main text (currently missing from supplementary material or the methods section in the main text).

References

D'Amico, B., Pomponi, F. & Hart, J. Global potential for material substitution in building construction: The case of cross laminated timber. *J. Clean. Prod.* 279, 123487 (2021).

Churkina, G. et al. Buildings as a global carbon sink. *Nat. Sustain.* 3, 269–276 (2020).

Nepal, P., Johnston, C. M. T. & Ganguly, I. Effects on Global Forests and Wood Product Markets of Increased Demand for Mass Timber. *Sustainability.* 13(24): 13943 (2021). <https://doi.org/10.3390/su132413943>.

Nepal, P. et al. The potential use of mass timber in mid-to high-rise construction and the associated carbon benefits in the United States. *PLOS ONE.* 19(3): e0298379. (2024) <https://doi.org/10.1371/journal.pone.0298379>

Taylor, A.; Gu, H., Nepal, P., Bergman, R.. Carbon Credits for Mass Timber Construction. *BioProducts Business.* 8, 12 (2023). <https://doi.org/10.22382/bpb-2023-001>.

Reviewer #6

(Remarks to the Author)

The main novelty of the paper is that despite additional substantial industrial wood harvest due to wood construction scenarios, forest growing stock is not depleted, but due to intensified forest management forest carbon stock is expected to increase substantially. The latter is caused by increased sawtimber prices due to rising lumber demand, and forest owners are intensifying management in anticipation of the future high lumber demand. From one side, GTM model is unique due to its dynamic optimization structure, which is able to capture forest management change in reaction to future lumber demand growth. On the other hand, no one has perfect foresight of the future. Nevertheless, dynamic optimization model is a valid approach to study possible future scenarios and optimal transitions towards the future. In addition, GTM model captures global wood markets interactions, including interactions between new mass timber products such as CLT and old type of wood-based products (IWP). Higher demand for CLT (driven by assumed CLT adoption rate in the new urban construction) drives additional sawtimber price increase, which in turn reduce demand for IWP products.

However, according to provided GTM method description, long run demand function for wood products is determined by GDP per capita development, price, income and price elasticity. Price elasticity is assumed as -1 based on provided references. At least two of the provided references do not have any information regarding industrial roundwood demand price elasticity. Morland et al., 2018 does provide industrial roundwood supply price elasticity, which isn't the same. Both Morland et al. 2018 and Buongiorno 2015 provide data on traditional wood-based products such as sawnwood, wood-based panels and various paper and paperboards grades. Based on these references price elasticity for the demand of sawnwood and wood-based panels are in the range of -0.1 to -0.5 (-0.3 on average). Price elasticity of -0.3 for traditional wood products imply relatively inelastic demand, while assuming -1 price elasticity for IWP imply rather elastic demand. With the assumed IWP price elasticity of -1, each percent of price increase reduces the demand by the same percentage. While each 1% of price increase with the price elasticity of -0.3 is going reduce demand by 0.3% only. Therefore, assuming elastic demand for IWP products leads to a greater IWP demand reduction due to sawtimber price increases.

One of the main findings of the study is that despite of the high demand CLT scenario the resulting additional sawtimber demand is more moderate due to substantial reductions of IWP products demand. However, based on the more conservative price elasticity estimates (taken from the same references), IWP products demand reductions are likely to be 2-3 times lower, which would result in the higher additional sawtimber demand (in the range of additional of 150 – 200 million m3 on top of high CLT demand scenario).

I am not asking the authors to make new model sensitivity runs with alternative price elasticity for IWP, since the paper has gone through several revisions already. However, it would be good to acknowledge in the discussion part the uncertainty regarding the extent of possible IWP products demand reduction, which will lead to higher sawtimber demands.

Consequently, based on higher sawtimber demand, what is the likely change for the overall carbon balance in the forest and forest products plus substitution of other non-wood construction materials. Is even higher demand for CLT going to improve the final carbon balance or other way around?

Besides the problem related to overestimation of IWP demand reduction, the paper in the current state is well written and present very interesting and novel results worth publishing.

Version 3:

Reviewer comments:

Reviewer #5

(Remarks to the Author)

The authors have addressed my comments satisfactorily. I recommend accepting the manuscript for publication.

Reviewer #6

(Remarks to the Author)

The main novelty of the paper is that despite additional substantial industrial wood harvest due to wood construction

scenarios, forest growing stock is not depleted, but due to intensified forest management forest carbon stock is expected to increase substantially. The latter is caused by increased sawtimber prices due to rising lumber demand, and forest owners are intensifying management in anticipation of the future high lumber demand. The GTM model is unique due to its dynamic optimization structure, which is able to capture forest management change in reaction to future lumber demand growth. Dynamic optimization model is a valid approach to study possible future scenarios and optimal transitions towards the future. In addition, GTM model captures global wood markets interactions, including interactions between new mass timber products such as CLT and old type of wood-based products (IWP).

Higher demand for CLT (driven by assumed CLT adoption rate in the new urban construction) drives additional sawtimber price increase, which in turn reduce demand for IWP products. One of the main findings of the study is that despite of the high demand CLT scenario the resulting additional sawtimber demand is more moderate due to substantial reductions of IWP products demand. However, based on the more conservative price elasticity estimates (taken from the same references), IWP products demand reductions are likely to be 2-3 times lower, which would result in the higher additional sawtimber demand. This issue regarding the uncertainty of the extent of possible IWP products demand reduction, leading to higher sawtimber demands is acknowledged and discussed in the discussion part.

The paper in the current state is well written and present very interesting and novel results worth publishing.

Reviewer #1

1.1. Comment: The manuscript presents an integrated framework that combines a global timber model and a life cycle assessment model to evaluate the carbon and land impacts of adopting cross-laminated timber (CLT). The main findings indicate that CLT adoption can increase carbon storage in forests and wood products, reduce carbon emissions by substituting traditional building materials and electricity.

The manuscript is well-written, concise, and the results are interesting, potentially relevant for decision-making. However, the manuscript has some weaknesses in its methods and analysis that need addressing. These are discussed as follows.

Response: We appreciate the positive feedback from the reviewer on our study. We are grateful for the reviewer's helpful comments. We have carefully addressed each comment and suggestion from the reviewer on a point-by-point basis and recorded them below and in the manuscript.

1.2. Comment: Existing CLT buildings have not yet reached their mid-service life, and a potential future development could involve increased volumes of post-use CLT. In the End-of-Life (EOL) stage, the manuscript considers only landfilling. Given that massive wood-based products like CLT have a high heating value, the EOL stage, with its benefits and burdens, significantly impacts the overall greenhouse gas (GHG) performance of such building product and system from a life cycle perspective. Therefore, it is crucial to analyze the implications of more progressive EOL management options, such as reuse, recycling, cascading, and energy recovery, in addition to the landfilling. This can have significant impact on estimated carbon flows.

Response: We thank the reviewer for recommendations on EOL management options. Since our study looks at 2020-2100 interval with 60-year CLT building life-time, only the CLT panels produced during 2020-2040 will be involved with EOL stage. Given the CLT production volume at the initial stage is relatively small, the impacts of different EOL cases may have limited impacts on the total results. Following the recommendations from the reviewer, two more conceptual EOL cases are added to test the variations from EOL on the total carbon stock change and GHG emissions. One is 50% recycling with 50% landfilling; the other one is 100% energy recovery for power generation. We have recorded the results in Supplementary Tables 3 and 4. The impacts of varied EOL cases of CLT panels are very minor. To better clarify this point, we

have added discussions in the manuscript and two tables in SI. Due to the long length of the tables, we kindly ask the reviewer to refer to the SI for details.

In Page 12:

“To explore the impacts of varied CLT end-of-life cases on the results, two additional conceptual end-of-life cases of CLT panels are conducted: 1) material recycling case with 50% recycling and 50% landfilling; 2) energy recovery case with 100% CLT panels combusted for power generation. The impacts from the different end-of-life cases of CLT panels are minor (see results in Supplementary Tables 3 and 4) due to the small quantity of CLT that reached their end-of-life (only those adopted in 2020-2040 in Fig.2, given a life span of 60 years).”

1.3. Comment: Among the different datasets and databases in Ecoinvent 3.7, the cut-off database was used to calculate carbon emissions. However, is this the most appropriate database for a consequential life cycle assessment (LCA) as aimed for in the paper? For a consequential LCA, the Ecoinvent 3.7 database for a consequential system model (long-term) may be more suitable. It is critical that appropriate dataset is used in showing the carbon consequences of the scenarios.

Response: We thank the reviewer for this question. We did not use the “substitution, consequential, long-term” version of Ecoinvent 3.7 for two reasons. The first reason is to avoid double-counting. The consequential system model (long term) “applies substitution to credit processes with the avoided burdens from supply chains that are replaced by the by-products generated within them” (Ecoinvent, 2024). In other words, the consequential system model of ecoinvent has default assumptions of substitution and their credits for all byproducts. But in our study, we need to separately quantify the consequences of future CLT production and adoption as well as to include the substitution effects of CLT materials and their byproducts based on our process-specific LCA models and economic model GTM. Using Ecoinvent 3.7 database for a consequential system model (long-term) may result in potential double-counting or inconsistent modeling of substitution effects.

The Ecoinvent consequential system model also has default assumptions of marginal suppliers, technology levels, market constraints and supply elasticity (Ecoinvent, 2024). In our study,

market constraints and elasticity were modeled by GTM. Using Ecoinvent consequential system model may result in inconsistent market assumptions. Hence, in this study, we use the cut-off version and calculate the consequences through our LCA- GTM integrated models. Since Ecoinvent has released the new version of Ecoinvent 3.9, we have updated our models based on Ecoinvent 3.9 but the result changes are very minor (less than 3%).

Reference:

Ecoinvent. Substitution, consequential, long-term. Substitution, consequential, long-term.
<https://support.ecoinvent.org/system-models>

1.4. Comment: The authors should provide more details on how they projected the CLT demand scenarios, especially the assumptions regarding market penetration rates, building floor area, and wood species mix. How sensitive are the results to these assumptions?

Response: We thank the reviewer for this question. In this study, we project the newly constructed urban building areas dependent on GDP per capita and urban population and then the CLT usage by considering the technology diffusion curve using a logistic model. Timber demand depends on CLT adoption in the building sectors; therefore, we designed three scenarios of CLT adoption, 1) medium and fast adoption: 30% adoption reached by 2050; 2) medium and slow adoption: 30% adoption reached by 2080; 3) high and fast adoption: 60% adoption reached by 2050. Following the suggestion by the reviewer, we added more discussions to the main text and the SI.

In Page 5:

“We examine a medium CLT demand scenario from 2020 to 2100 that reaches 30% of global new urban construction floor area, with either fast (Scenario 1, reaching 30% by 2050) or slow (Scenario 2, reaching 30% by 2080) adoption. We also look at fast adoption with a high demand scenario (Scenario 3) reaching 60% by 2050 (see Methods and Supplementary Figure 1).”

In Page 28:

“The sawtimber demand caused by adopting CLT ($\Delta TimberDemand_{t,n}^{CLT}$) is projected from 2020 to 2100, as shown in equation (18). In equation (18), $NewA_{i,t}$ is the annual newly constructed

urban commercial and residential building floor areas *that adopt steel and concrete structures* in year t and region n that can be potentially replaced by CLT; f_{CLT} is the CLT usage factor (m^3 CLT per m^2 building area, see Supplementary Table 10)³⁶; c_{CLT} is the conversion factor for CLT from timber (m^3 CLT per m^3 wet timber) that is evaluated based on the process model mentioned above; r_t the adoption rate (%) of CLT (or say the percentage of newly constructed areas that are built with CLT). $NewA_{i,t}$ from 2020 to 2100 in this study is derived from the total building floor areas of urban commercial and residential building based the multi-variable regression models for each region that assumes total building floor areas depend on GDP per capita and urban population^{17,39}. *The historical data of urban residential and commercial building floor areas from 1970 to 2010 are derived from the study by Deetman et al. for training the regression model*³⁹. The projections of GDP per capita and urban population follow the SSP2 scenario¹⁶. The details of the regression model and method are shown in Supplementary Note 5 and Supplementary Table 12. In this study, r_t is assumed to follow the technology diffusion curve by using a logistic model (see Supplementary Note 5 for the equation)^{40,41}. Three different adoption scenarios are established to explore the impacts on the carbon consequences³⁶: 1) medium and fast adoption: 30% adoption reached by 2050; 2) medium and slow adoption: 30% adoption reached by 2100; 3) high and fast adoption: 60% adoption reached by 2050. More details are shown in Supplementary Note 5 *and Supplementary Data*.

$$\Delta TimberDemand_{t,n}^{CLT} = (NewA_{t,n} \times f_{CLT}) / c_{CLT} \times r_t \quad (18)''$$

1.5. Comment: The paper does not explicitly address uncertainties in data, CLT production technology, and their quantitative impacts on the calculated results. The authors should discuss the limitations and uncertainties of their modeling framework, which include the simplification of soil carbon dynamics and the variability in LCA, particularly since CLT technology is relatively young. For example, ongoing research is investigating open issues to further improve CLT technology in areas such as material efficiency and the development of climate-friendly alternatives to adhesives. These ongoing developments can significantly impact the assumptions made in the study and the overall findings. Therefore, the paper should discuss uncertainties, and the results should highlight sensitivities to variations in different data inputs, methodological choices, etc.

Response: We thank the reviewer for these helpful suggestions. Following the suggestion from the reviewer, we added the sensitivity analysis that considers the variabilities in LCA modeling and potential substitution. Combining the previous recommendations from the reviewer, we added the different end-of-life cases of CLT panels to explore the effects of material efficiency and recycling. We also added the discussion related to the limitations of this study in this part.

In Page 12:

“To explore the impacts of varied CLT end-of-life cases on the results, two additional conceptual end-of-life cases of CLT panels are conducted: 1) material recycling case with 50% recycling and 50% landfilling; 2) energy recovery case with 100% CLT panels combusted for power generation. The impacts from the different end-of-life cases of CLT panels are minor (see results in Supplementary Tables 3 and 4) due to the small quantity of CLT that reached their end-of-life (only those adopted in 2020-2040 in Fig.2, given a life span of 60 years).”

In Page 14:

“This study conducts sensitivity analysis to investigate the impacts of variations in input parameters or assumptions on the results (see Supplementary Fig. 2 for detailed results). Most parameter variations have minor effects. The most sensitive parameters in the analysis are the GHG emission factors of electricity and concrete which determine the potential substitution benefits. If the global electricity system is decarbonized by more than 50% (i.e., from 0.70 to 0.33 kgCO₂e per kWh), the total cumulative change of GHG emissions from 2020 to 2100 in Scenario 3 will increase by 6.0% because the benefit of replacing market electricity will be much smaller. If the GHG emissions of producing concrete are reduced by 40.0%, the total cumulative change of GHG emissions from 2020 to 2100 in Scenario 3 rise by 2.4%. These impacts are much smaller than the effects of CLT demand and adoption rate explored in Scenarios 1–3.”

In Page 18:

“Based on this study, future research can further investigate possible improvements in background processes (e.g., electricity generation and concrete production) and foreground processes (e.g., CLT production) in each region. Moreover, future research efforts can investigate the effects of other drivers (economic growth and future climate change scenarios).”

SI Supplementary Figure 2:

Supplementary Figure 2. Sensitivity analysis of global cumulative change in greenhouse gas emissions relative to the baseline from 2020 to 2100 in Scenario 3. The parameters and variables with less than 1% impact on the results are not presented given the minor impacts.

Reviewer #2

2.1. Comment: Authors have used a combined forest economics model (Global Timber Model i.e., GTM) and an LCA model to explore the potential environmental impacts of using CLT in future building construction scenarios. Their findings suggest that widespread CLT adoption could lead to a significant increase in carbon storage (up to 7 GtC) in standing forests and harvested wood products (including CLTs). Additionally, they estimate a reduction in greenhouse gas emissions of approximately 40 GtCO_{2e} due to improved biogenic carbon uptake and lower emissions associated with HWP production and disposal.

Response: The original manuscript explained the results both in terms of Carbon (C) stored in trees and CLT and Carbon Dioxide (CO₂) removed from the environment. We can see that this has been confusing for readers. The revised manuscript consequently just uses GtCO_{2e} to measure all changes. The total effect in the high CLT scenario is the removal of 39.4 GtCO_{2e}. This is mostly due to a 7 GtC increase in carbon stocks (i.e., forest, CLT, landfill site, IWP) which is equivalent to removing 25.2 GtCO_{2e} in the atmosphere.

This paper was an excellent read, offering a comprehensive analysis of the environmental consequences of leveraging future timber buildings using CLTs for GHG mitigation. The authors innovatively combined a well-known forest economics model (GTM) with a comprehensive LCA analysis, a unique approach I have not encountered in any other existing study. This undertaking deserves high praise, especially considering how it highlights critical research gaps in current literature.

Response: Thank you for acknowledging the uniqueness of this study and recognizing our work as an “excellent read”.

While I acknowledge the expertise of the co-authors and my own limited experience with an older version of GTM, I believe that open-sourcing both the GTM code and the employed LCA model would significantly enhance the transparency and replicability of this study. This would allow other researchers to verify the results, build upon them, and address potential limitations in future studies.

Response: We agree with the referee that providing open-source coding would be a contribution to the literature. All the code related to GTM and future projections is available at online data

repository Zenodo: <https://doi.org/10.5281/zenodo.10562736>. For LCA, the modeling part is performed in OpenLCA software. All the processes we used are recorded in Supplementary Table 7. All data collected from literature and used in this study have been recorded in Supplementary Data and SI. Following the recommendation from the reviewer, we have added the code availability section to the main text.

“Code availability

All the code used in this study is available at Zenodo: <https://doi.org/10.5281/zenodo.10562736>.”

The research presented here is innovative, but I have significant concerns regarding both the chosen tools and the authors justification for the study. Notably, the authors provide minimal justification for selecting GTM as a suitable model for addressing their research question. GTM is inherently not a land-use model and only simulates specific forest land pools. Additionally, GTM lacks the ability to model competition for land between agriculture, forestry, and other land uses. This significantly weakens the land-use arguments presented in the paper. Furthermore, the authors rely heavily on citing papers written among themselves, necessitating constant referencing back and forth between multiple publications. While they may have extensive background knowledge informing this study, the manuscript lacks sufficient context for neutral readers to fully grasp the results.

Response: The referee is correct that land use models are important tools for understanding forces that shift land from one use to another. The specific issue at hand with CLT involves two land uses: agriculture and especially forestry. GTM was chosen because it includes a lot more detail about the forestry sector than any other land use model. It models stocking (the number of trees and acreage in each age class), site productivity, and growth through time. It recognizes that market forces will push forest management to anticipate future demand not just react to current demand. It also carefully models natural forestland, recognizing that this forestland contains an average amount of biomass in each type of forest. Although GTM does not carefully model the details of agriculture, it does carefully model the interaction between agriculture and managed forest through a supply function for farmland. The more land that the forest wants from farmland, the more expensive the land gets. This recognizes the price inelasticity of farmland (the underlying demand function for food).

The authors do cite previous GTM papers so that the reader understands what is known already and what is new. But we have tried to include enough information about GTM in the Methods and SI so that readers can understand how it works. We have also added more explanation in the text explaining why the model behaves as it does.

Most of the results are presented in relation to the baseline but the manuscript lacks explicit baseline results (excluding Figure 2, where it appears unnecessary, and Figure 3 does have baseline results). This omission hinders our ability to assess the magnitude of differences across the three scenarios compared to the baseline scenario itself. Consequently, we cannot evaluate the performance of the presented modeling framework on the baseline scenario.

Response: This is a good point. The paper focuses on how things change because of CLT. But we agree with the referee that we should be clearer about the baseline. We now make clear in the text that CLT causes only a small change in the global demand for wood. Even with the most aggressive CLT scenario, the global demand for wood increases by only 11%. This important perspective is now in the text on Page 6.

To follow the recommendations by the reviewer, we have added the data of the baseline scenario to SI (Supplementary Tables 1 and 2) and references to the main text.

In Page 6:

“The annual global wood supply for CLT in Scenario 1 and Scenario 2 reach about 300 million m³ (around 9% of total wood supply for IWP, see Supplementary Table 1) by 2100, whereas the high demand scenario reaches 634 million m³ (19% of total wood supply for IWP, see Supplementary Table 1).”

In Page 9:

“At the global level, CLT demand drives a small increase in plantations, a large increase in managed forest, and a moderate decrease in natural forest and marginal farmland relative to the baseline (see the baseline forest area in Supplementary Table 2).”

This manuscript's writing style primarily reflects a modeling study approach, potentially limiting its accessibility to the broader audience of Nature Communications. Given the lack of broader appeal and the presence of methodological concerns, I would not be able to recommend this manuscript for publication in Nature Communications.

Response: We have revised several parts of the manuscript to be sure that readers understand the reasons why the model generates specific results. For example, we explain that the increase in storage of carbon in the forest is a result of the renewable nature of modern forestry which plants forests in anticipation of future wood demand. We have kept the formal modeling language for the Supplementary Information. Detailed revisions and modifications are highlighted in the revised manuscript.

I have attached my detailed and minor comments in the file attached which hopefully are of use to the authors. Detailed comments start from page 2 of the attached document.

Response: We appreciate these detailed comments and have changed the manuscript to respond to them.

2.2. Comment: L44-46: This is an incorrect representation of Citation 10. The land conversion dynamics in the model used by Mishra et al. already take account of the cost of converting land – especially marginal land. Additionally, the assumption made by Mishra et al. is that the mass timber demand is counted as an additional demand on top of business as usually roundwood demand.

Response: We agree with the referee about Mishra et al. Mishra et al account for replanting cost with respect to marginal land. But they do not distinguish whether or not the marginal land can support a forest. Forests generally require a minimum amount of precipitation which is not available with “other land”. Mishra et al added the quantity of mass timber demand on top of the quantity of roundwood demand. However, summing quantities is not the same as summing demand functions. Mishra et al do not capture how the different uses of wood compete with one another for the available supply. The rise of CLT timber leads to a substantial reduction of wood going to sawtimber and pulp. We have revised our description of Mishra et al. to make this clearer.

In Page 2:

“Specifically, they did not take account of the indirect effects on *industrial* wood products (*IWP*) (e.g., *lumber, particle boards, pulp and paper* products) that will compete with mass timber demand and the *very high cost* of converting *marginal* land into plantations *as land availability declines*.”

2.3. Comment: L55: How exactly can secondary forest increase? These are naturally regenerating forests. The statement would make sense if secondary forests were newly established/reclassified. Unclear how this dynamic works in the model presented here.

Response: The language in the previous version was not accurate. “Secondary forest” should have been called “managed forest”. Early forest management was often a cut-and-run operation. The cutover land was either converted to agriculture or became a secondary forest (it regrew naturally). What GTM is modeling is the conversion of secondary forest into managed forest. Almost all of modern forest management is now a plant and cut renewable operation. The secondary forest from before is now harvested and then planted and converted into managed forest immediately after harvest. The secondary forest is being replaced by a managed forest. Planting leads to rapid full stocking which in turn leads to faster growth and more carbon being stored at each age. Plantations involve even more management including site preparation (fertilization) and thinning. Plantations lead to even faster growth, but plantations are restricted to highly productive forestland. New plantations would have to come from prime cropland where new managed forestland would tend to come from marginal farmland. GTM models these choices as forward-looking profit maximizing forest management. But GTM is constrained by an underlying ecological model that determines where forests can grow and how productive the land is. We have revised the manuscript to clarify this point. We also changed all “secondary forest” to “managed forest” to avoid confusion.

In Page 3:

“First, adding the demand for mass timber products *will likely lead to higher wood prices which will reduce the market for IWP*. Second, *higher wood prices will give landowners an incentive to plant more forests and increase forest management intensity*. This likely increases the future supply of wood but incidentally also increases the carbon stored in forests.”

2.4. Comment: L61: What could happen if we just clear-cut all available forests to meet higher CLT demand hypothetically? The authors have presented their model outputs as part of the introduction in multiple places like this.

Response: GTM recognizes that a great deal of forestry involves harvesting with clear cuts although there is some partial harvesting in less productive forests (mixed forests and Boreal forests). But the model also captures the fact that forestry now involves replanting. Europe managed forests renewably starting in the 19th century. The US started planting renewable forests in the 1950's. The entire world has turned to renewable forest management only more recently (since the 1990's). This transition to renewable forestry implies that increases in future global timber demand no longer imply a reduction in fixed forests with fixed forestland. Increases in demand now imply the opposite, there is an increase in managed forestland and there are more trees in anticipation of this increased demand.

2.5. Comment: L68: The authors state that the framework used here can support policymaking but offer no concrete examples of any policy recommendations throughout the manuscript.

Response: We have added policy recommendations to the discussion.

In Page 17:

“The results of this study agree with the literature that adopting CLT will lead to beneficial carbon consequences¹⁰. Using CLT instead of steel and cement will increase wood demand and encourage the market to protect more forestland and intensify forest management which ultimately increases forest carbon stock. Our study suggests there will be a large net increase of carbon stock in the forest and in CLT panels, *although some carbon stock will be lost because of the declining production of IWP*. Encouraging CLT adoption may be a compelling market tool to increase carbon storage in forests and wood products *with an overall* forestland by 30.7–36.5 Mha. On the *other hand, the new demand for CLT will not be beneficial to* natural forestland which will *fall by 10.6–18.9 Mha. This loss of natural forests will likely reduce* natural habitat. These land use effects will vary across the planet with more production and forestland in the United States and Western Europe but large reductions in natural forestland in the tropics. *There may well be a tradeoff between some conservation goals and some carbon goals. The more forestland that is set aside for*

conservation, the more expensive it will be to produce timber products that naturally encourage carbon storage and vice versa. Future policy studies need to look at this tradeoff more closely to guide the best policies for each region. Biodiversity benefits together with carbon storage benefits from forests should be included in future analyses of the effects of CLT production.”

2.6. Comment: L69: Environmental consequences are not concretely discussed anywhere in the manuscript.

Response: The paper does describe what is happening with forests as a result of the increased CLT demand. The paper does highlight that some of the increased managed forestland comes from natural forest. One drawback of reducing natural forestland is a likely reduction in habitat which we now carefully cover in the conclusion as a topic of future research. However, it is important to keep in mind that most of the supply of timber is coming from existing managed forestland, not newly harvested natural forest. The trees in current natural forests are only a small fraction of aggregate supply.

2.7. Comment: L73: Perhaps something to add in the methods section but are NPIs or NDCs included in forest conservation? I could not find it anywhere mentioned in the manuscript.

Response: GTM assumes that land which is formally conservation land is not available for harvesting or conversion. To the extent that increasing CLT demand leads to increased carbon stocks in forest, the CLT demand is contributing to the climate goals of some countries to store more carbon in forests. It can therefore be an assist to countries who want to store more forest carbon but may have limited resources to pay for it. However, the shift from natural forest to managed forests may exacerbate the difficulty of using more forestland for conservation. This is an important topic for both policy and planning and deserves further thought and research. We now address this policy conflict in the conclusion section. NDIs measure plantations which are one of the outcomes predicted by GTM. Higher timber prices make plantations more attractive. However, a country must have productive forestland to support a plantation.

2.8. Comment: L80: Unclear what 30% refers to (in global new construction) – raw materials? Buildings? Floor space?

Response: It refers to the global new construction floor area. We have added that to this sentence.

In Page 5:

“We examine a medium CLT demand scenario from 2020 to 2100 that reaches 30% of global new urban construction floor area, with either fast (Scenario 1, reaching 30% by 2050) or slow (Scenario 2, reaching 30% by 2080) adoption. We also look at fast adoption with a high demand scenario (Scenario 3) reaching 60% by 2050 (see Methods and Supplementary Figure 1).”

2.9. Comment: L83-85: Does GTM have other wood product categories? It could be useful to see what total roundwood demand looks like in GTM. In 2020, global roundwood production was ca. 4000 Mm³ mostly equally divided between industrial roundwood and wood fuel. From Fig. 3, looks like sawtimber and pulpwood demand goes up to 2300+1200 = 3500 Mm³ by 2100 – but these products are (likely) only a portion of total roundwood production.

Response: GTM models only industrial roundwood. GTM modeled total production of industrial roundwood as 2122 Mm³ in 2020 which rises to 3627 by 2100 in the BAU scenario. With the high demand scenario for CLT, the total demand for roundwood rises to 4014 Mm³ by 2100 of which 634 Mm³ is for CLT. The high CLT demand causes aggregate wood harvests to increase by 387 Mm³, an 11% increase by 2100. CLT competition with sawtimber and pulpwood causes these traditional uses to fall by 247 Mm³ in 2100. These changes are shown in Figure 3c and d.

We do not examine the demand for and supply of fuelwood in response to CLT. Fuelwood comes largely from whole trees in areas with trees but not forests (parkland, savannah, and mangroves) and branches in forested areas. These areas rarely contribute to IWP. Fuelwood supply is largely independent of forestland devoted to timber. We have added a footnote making clear we are not talking about fuelwood.

In Page 4:

*“*In this study, GTM does not model the demand and supply of fuelwood which is largely independent of IWP. CLT is likely to have no effect on fuelwood demand or supply.”*

2.10. Comment: L96-97: This is perhaps the main model output from GTM due to the way it is set up, but can the authors elaborate in simpler terms why traditional HWP demand will reduce due to an increase in prices of timber driven by higher demand for CLTs? This dynamic would assume that CLTs are more desirable than traditional HWPs which is not clarified in the manuscript. CLTs are not necessarily a substitute good for sawtimber and pulpwood even if roundwood is used as a raw material for sawtimber (incl. CLTs) and pulpwood. In SI Fig. 1 it looks like sawtimber is converted into CLT. How are the price and demand effects decoupled between traditional sawtimber and CLT?

Response: We assume in the high demand scenario that the demand function for CLT rises enough to get the predicted amounts of CLT. That is, CLT outcompetes other uses of wood leading to this scenario. As discussed in response to the reviewer's previous question, the higher aggregate demand causes the overall wood supply to rise. But the higher overall wood prices reduce the amount of wood going to pulp and sawtimber.

2.11. Comment: L99: Unclear what internally consistent means.

Response: Internally consistent means that the growth of the economy, interest rates, and the initial forest stock are the same in the BAU and in all the CLT scenarios. It also means that the LCA and GTM are based on the same assumptions and so are consistent with each other. The model is capturing the endogenous market responses to all this change.

2.12. Comment: L117-120: What is the reason for ROW not contributing much to CLT production?

Response: ROW largely represents boreal and tropical forest. The model predicts that CLT will largely be produced from the wood from temperate forests. However, as temperate forest supply is used for CLT, there may well be more timber coming from ROW forests to supply IWP. So, ROW forests also respond especially to the high CLT demand scenario as shown in Figure 4.

2.13. Comment: L128: Fig. 3c and 3d show an opposite effect i.e., an increase in sawtimber and pulpwood prices corresponding to lower supply (also mentioned in L137-138). I would urge the authors to stick to consistent naming throughout the manuscript between wood, timber, etc.

Response: We have carefully gone through the manuscript to be more consistent about names. We have revised the discussion in this part.

In Page 7:

“When the global demand for CLT is added, *the prices of sawtimber and pulpwood rise relative to the baseline scenario* as shown in Fig. 3a and 3b. *Compared to the baseline scenario, the faster CLT demand grows, the quicker the prices of sawtimber and pulpwood increase. Higher overall CLT demand leads to higher overall future sawtimber and pulpwood prices. The higher future prices stimulate an increase in the future total wood supply (including sawtimber for IWP, pulpwood, and sawtimber for CLT) (see Supplementary Table 1).*”

2.14. Comment: L129: Where can we see this increase in forest management? Is it implicitly shown as forest area increase presented in Fig. 4?

Response: The increase in forest management leads to an increase in the growth rate of managed forests. High management leads to more biomass in each age class of managed forest. This eventually leads to more yield per hectare at harvest on managed forestland, but it also leads to more carbon in every age class of managed forest. We do not isolate the effect of forest management in any figure.

2.15. Comment: L131: Same as the earlier comment about wood or timber or CLT. Unclear what “wood” means here.

Response: This sentence is referring to aggregate industrial wood. We have revised the sentences there.

In Page 7:

“*Compared to the baseline scenario, the faster CLT demand grows, the quicker the prices of sawtimber and pulpwood increase. Higher overall CLT demand leads to higher overall future sawtimber and pulpwood prices. The higher future prices stimulate an increase in the future total wood supply (including sawtimber for IWP, pulpwood, and sawtimber for CLT) (see Supplementary Table 1).*”

2.16. Comment: L132 – 133: How does this shift of secondary forests into plantations occur in GTM? Does the investment in managed forests (plantations?) increase yields? If yes – how does it impact the prices of sawtimber and pulpwood in GTM?

Response: Global forestry has gradually moved from cut and run practices before 1950 to replanting after harvest. So as natural regenerated forests have been harvested more recently, they have been planted and converted to managed forests. The more intensively managed forests on the most productive land have been turned into plantations by extensive site preparation, overstocking, and then thinning.

2.17. Comment: L134: As authors already mention ecology and related concepts earlier in the manuscript – how can we justify moving low-value farmland and natural forestland into managed forestland? This is likely detrimental to biodiversity. This is also a likely drawback of the model(s) being used here but is not discussed anywhere in the manuscript.

Response: One can justify moving low-valued farmland into forestry both for timber and for carbon. There is very little above ground carbon storage on farmland. Changes in below-ground carbon are relatively small. The movement of natural forest into managed forest does remove above ground carbon. We rely on the average stocking of existing natural forests to measure this carbon (and wood).

The movement of farmland to managed forest is likely to lead to a small improvement in habitat. In contrast, the movement from natural forest into managed forest likely degrades habitat. This becomes a tradeoff between carbon goals and conservation goals. The topic deserves careful attention as we now emphasize. We have added relevant discussions as mentioned in our earlier response in response to comment 2.5.

2.18. Comment: L151 – why increased CLT demand only drive a small increase in plantations? More specifically – how is the demand signal passed on to equation 3 which determines the stock of land in each forest type? Alternatively, how does GTM decide the extent of the new plantation establishment?

Response: We have now made clear that even the high demand CLT scenario increases the aggregate demand for industrial wood only by 11%. This leads to plantations rising slightly. The

reason for not seeing a bigger increase in plantations is that forestland must be highly productive to support a plantation and only a small fraction of forestland fits that description. To get new plantation land, one must rely on prime cropland, highly productive land. There is a high opportunity cost to take prime cropland (it is expensive land). Further, to replace the lost agricultural production, the system will convert forestland into marginal farmland. But it can take many hectares of marginal cropland to replace one hectare of prime cropland. This, in turn, leads to substantial amounts of deforestation and the likely loss of forest carbon.

2.19. Comment: L151 –How exactly does a naturally regenerating forest increase? Does GTM adjust secondary forest area during optimization? Equation 3 has indexes of age classes and time steps. Do all forest classes in GTM have age classes associated with them – certainly looks like it from L395. How was the age class distribution determined for plantations, secondary forests, and natural forests? This is not clear in methods or SI.

Response: Modern forestry replants as it harvests. This causes naturally regenerated forestland to become managed forestland. Planting is a huge part of this transition because the planted forest regenerates immediately and is fully stocked. Managed forest also gets support throughout the life of the trees. Plantations are similar to cropland and get a lot of support. With higher expected future timber prices, GTM assumes that some marginal farmland can be purchased for forestland and some natural forestland can be made accessible through roads.

GTM models all managed forestland in the world by age class. This was done by starting with natural forests and carefully modeling when forests were harvested and replanted using historic harvests from Houghton (2008). Natural forests are assumed to have an average natural stocking given the forest type and region.

2.20. Comment: L155:160 – Authors refer to changes compared to the baseline scenario, but we do not see baseline numbers in most results. This makes it difficult to ascertain what is the magnitude of forest area changes in scenarios 1:3.

Response: This was an oversight that we have corrected. The 2020 total forestland is 3900 Mha of which 27% is managed forestland (of which 9% is plantation). The BAU total forestland in

2100 is 4530 Mha of which 51% is managed. Natural forestland is assumed to fall from 2864 Mha now to 2241 Mha in 2100 in the BAU scenario.

Following the recommendation from the reviewer, we added the source data of baseline forest area to the SI (Supplementary Table 2) and added the reference to the main text.

In Page 7:

“The change in CLT is large enough to shift the aggregate demand for wood but even in the most aggressive scenario, global aggregate wood supply only increases by 11%.”

In Page 9:

“At the global level, CLT demand drives a small increase in plantations, a large increase in managed forest, and a moderate decrease in natural forest and marginal farmland relative to the baseline (see the baseline forest area in Supplementary Table 2).”

2.21. Comment: L155:160 – Demand for sawtimber is almost double in scenario 3 compared to scenarios 1 and 2 –but there is only ca. 3Mha more of net forest change in these scenarios. Can we perhaps see what the timber yields look like in GTM – even in SI? On average this would translate to a yield of about 100m³/ha which seems rather high.

Response: The high CLT scenario has total forestland in 2100 equal to 4567 Mha which is just a 36 Mha increase compared to the BAU. Natural forestland decreases by 19 Mha. Managed forestland increases from 2289 to 2345 Mha by 2100, an increase of just 55 Mha of forestland (of which 15 Mha is plantation).

By 2100 with the BAU scenario, the model predicts the world can harvest 3627 Mm³ of timber per year from 2289 Mha of managed forestland which is about 1.58 m³/yr/ha. In the high CLT demand scenario, the model predicts the world can harvest 4014 Mm³ of timber per year from 2345 Mha of managed forestland which is about 1.71 m³/yr/ha. The increase in harvest from CLT is not just from the increase in managed forestland land but also the increase in forest productivity across all the managed land.

2.22. Comment: L219 – The methodology presented here severely lacks accounting of the whole land-use system. Presenting a simple ratio between CLTs produced (cumulatively) and GHG changes likely provides an incorrect representation of how much mitigation potential is

provided by CLT production. These numbers make sense only in isolation of the modeling boundaries and should be presented in that context.

Response: The model is very careful about accounting for the changes in land use that CLT is likely to cause. This includes the shift from agriculture to forest and the shift from natural forest to managed forest. CLT is not likely to cause other land uses to change. Dividing the cumulative quantity of CLT by the cumulative mitigation provides a policy maker with a simple average effect of CLT that is accessible. We are careful to note that this average is not the same across each scenario.

2.23. Comment: L257: The only ecological constraint seems to be limiting forestland to locations that can support forests. Does GTM account for the following in any capacity?

- changes in biodiversity
- NPIs already in place and promised NDCs
- Land conservation activities across the globe
- COP26 declaration to end deforestation by 2030.

If not, this should probably become part of the discussion especially when the manuscript focuses on land and carbon consequences of mass timber/CLT adoption.

Response: The referee is correct that the predictions of the effect of CLT are independent of possible changes in policy over time. This is worth noting. We have added a section in the conclusion suggesting that future policies could affect the predictions of the model. Policies that would lead to more forestland being devoted to conservation would make it more difficult to supply more timber. Whether conservation goals are in conflict with or support carbon goals is an important topic for more research. Policies that reward landowners for storing carbon in forests would lengthen forest rotations but would not necessarily prevent forestland being used to grow timber. It is not clear that managing forests renewably is necessarily in conflict with deforestation if it in fact leads to more forestland. We now discuss these important points in the conclusion.

2.24. Comment: L263-266: Can the authors elaborate on what exactly is the “net GHG emission effect”? Same comment for the instance of these words in L278.

Response: Net GHG emission effect is the net effect on production and end-of-life GHG emissions from CLT and IWP. It measures the increase change in GHG emissions from increased CLT production and end-of-life and the reduction in GHG emissions from reduced IWP production and end-of-life. We have revised the term in the manuscript to make it clear.

In Page 16:

“For the production and end-of-life stages, the combination of the increased GHG emissions by CLT and the reduced GHG emissions by IWP, leads to a cumulative increase of 1.6–2.4 GtCO_{2e} by 2100. When also considering the substitution of steel and concrete and the use of wood waste to generate electricity, the net production and end-of-life GHG emissions become a reduction of 1.6–9.1 GtCO_{2e}. This is 6.3%–23.6% of the total net cumulative GHG emissions depending on the scenarios.”

2.25. Comment: L284: The 143 Mha additional plantations for 90% of new urban dwellers living in buildings made of mass timber is most likely an infeasible scenario. But to justify why an additional 143Mha of plantations is impractical, authors could compare their CLT demand scenarios to the engineered wood demand from Mishra et al. – the reason is most likely a super high demand for mass timber in Mishra et al. – even higher than the CLT demand in scenario 3 of this manuscript.

Response: It is not clear what the high demand scenario is for mass timber in Mishra et al. It may not be that different from our own assumptions because Mishra et al assumes a huge reduction in the number of new buildings in their high scenario. The big question is whether a high demand scenario leads to the Mishra et al. prediction of 143 Mha of additional forest plantation. Mishra assumes the plantations would come from cropland, secondary forest, and “other land”. However, plantations require highly productive forestland. It is not clear that existing secondary forests or “other land” are highly productive. One could use prime cropland, but this would lead to a serious reduction in aggregate food production. If one used prime cropland, a great deal of remaining forestland would have to be converted into low-productivity farmland to feed the world. But this would have the undesired effect of releasing substantial amounts of forest carbon.

2.26. Comment: L285: “largely ineffective” in which sense? Also, Is the cited reference, correct? The Cited Favero et al. paper nowhere states that plantations are “largely ineffective”. Here is a verbatim quote from the cited paper – “These results show that it is more cost-effective to meet the new demand by investing in plantations than harvesting unmanaged forestland.” Besides this, plantations cover only about 3% of the global forest area but provide more than 33% of global roundwood production (FAO data). Plantations are not an ideal candidate for supporting biodiversity as such, but they are quick in accumulating biomass if managed correctly, even the monocultures.

Response: The phrase “largely ineffective” has been replaced. What we are trying to convey is that one needs very productive land to support a forest plantation. Prime cropland for instance would work. But if one removes 143 Mha of prime cropland, one would need about 450 Mha of low productivity farmland to replace the lost food output. This could not come from “other land”. It would likely come from forestland.

2.27. Comment: L330: Authors use harvested timber output here, and HWPs in other places. Consistent naming across the manuscript would help readability.

Response: We have gone through the manuscript to use Industrial Wood Products (IWP) for sawtimber and pulp, Q^{IWP} for wood inputs to IWP, Q^{CLT} for wood inputs to CLT, and wood to describe harvests from forests throughout.

2.28. Comment: L350-352: Unclear what the authors want to describe here. Sohngen et al. paper cited here states that uncertainty in the parameters dictating forest growth has an impact on carbon storage potential – more so than uncertainty in the land supply elasticity parameters. It does not appear to be a good fit in describing the functioning of GTM in the context of the methods section.

Response: We have done sensitivity analysis of our land use results to different SRES socioeconomic scenarios. Different assumptions about population growth and GDP growth do not have a large effect on the CO₂ consequences of CLT.

2.29. Comment: L367: Is the same income elasticity used for all wood products in GTM? GTM also used to have income elasticity which varied over time applied to GDP per capita.

Alternatively, Morland et al. seem to have some numbers for common wood products which could be useful. Morland, C., Schier, F., Janzen, N., and Weimar, H.: Supply and demand functions for global wood markets: specification and plausibility testing of econometric models within the global forest sector, *Forest Policy Econ.*, 92, 92–105, 2018

Response: Morland et al. income elasticities are also quite close to what we have assumed. Yes, the same income elasticity was used for all wood products in GTM. We add Morland as a reference.

2.30. Comment: L368: These are almost 20-year-old citations for an important parameter of GTM. Have there been no recent updates to this income elasticity parameter?

Response: We substitute Buongiorno 2015 and Morland et al 2018 references:

Buongiorno, J. (2015). Income and time dependence of forest product demand elasticities and implications for forecasting. *Silva Fennica*, 49(5).

Morland, C., Schier, F., Janzen, N. & Weimar, H. Supply and demand functions for global wood markets: Specification and plausibility testing of econometric models within the global forest sector. *For. Policy Econ.* 92, 92–105 (2018).

The most recent GTM paper with elasticity parameters is Tian, X., Sohngen, B., Baker, J., Ohrel, S., & Fawcett, A. A. (2018). Will US forests continue to be a carbon sink?. *Land Economics*, 94(1), 97-113.

2.31. Comment: L376: How is forest management considered in GTM? Same for L379.

Response: Forest management is an endogenous outcome predicted by GTM based on expected forest price and the cost of increasing management intensity. Higher wood prices induce higher investments in forest management increasing its productivity (measured as m³/ha harvested).

2.32. Comment: L384-L387: How does GTM calculate farm prices? And is GTM looking only at price comparison to make land-use decisions? Is there no “minimum” food, feed, and timber demand in the model to provide a minimum production level and land-use levels for non-forest activities and commodities? Hypothetical example – if the world was only 4 pixels large, with 2 pixels for agriculture, and 2 pixels for forests – with some people living in this hypothetical world. Could GTM convert all 4 pixels to forests if farm prices fall below timber prices? How would people feed themselves? It is most likely a non-issue if GTM has no agriculture component but this could be clarified in methods and also stated in discussions if possible.

Response: GTM includes a supply function for farmland which rises with the aggregate size of forestland. The model captures the idea that farmland is price inelastic because the demand for food is price inelastic. That is, the model charges ever higher prices for taking more farmland.

In Page 2:

“Specifically, they did not take account of the indirect effects on *industrial* wood products (*IWP*) (e.g., *lumber, particle boards, pulp and paper* products) which will compete with mass timber demand and the *very high cost* of converting *marginal land* into plantations *as land availability declines*.”

2.33. Comment: L397: Where is the initial stock of land taken from? A citation would help here.

Response: FAO is our primary reference for the stock of forestland. We have added the citation in the main text.

In Page 22:

“The initial stocks of land X_t^i are based on FAOSTAT data²⁸ and all choice variables are constrained to be greater than or equal to zero and the area of timber harvested $H_{a,t}^i$ does not exceed the total timber area.”

28. Food and Agriculture Organization of the United Nations. FAOSTAT.

<https://www.fao.org/faostat/en/#home> (2023).

2.34. Comment: L399: What is the decision-making behind the newly planted forest? i.e., how is N_{ti} determined?

Response: Newly planted farmland is derived from competition for land between managed forestland and marginal farmland and from a supply function to natural forestland. The supply function for farmland has been described above. Forestland will expand into low valued farmland if real wood prices increase. Managed forestland will expand into natural forestland as well if prices increase. The model prohibits expansion into existing conservation land (e.g. national parks, designated conservation land) but allows expansion into inaccessible natural forestland as real wood prices increase. The model assumes expansion into natural forests when the market can pay for access. Previous analyses, for example, show that if the world conserved ever greater fractions of existing forestland, the price of wood would increase because of forestland scarcity.

2.35. Comment: L402: Unclear what “competition for supply equilibrates their prices” means. Eqn. 4 and 5 - Missing symbol descriptions. Same for many other equations. A table for a description of all indices and all symbols would improve the readability of equations.

Response: Competition amongst importers and exporters for international wood products including future CLT, would cause the international price to equilibrate to a single global value. We now add a table (Supplementary Table 6) to include all the variables that appear in the manuscript in SI, since the journal does not allow tables in section Methods.

In Page 21:

“The notations used in this study and corresponding descriptions are available in Supplementary Table 6.”

Variables in equations (4) and (5) are now defined.

2.36. Comment: L20-21: in which period did this expansion happen? (likely by 2100?)

Response: Expansion would occur this century as soon as it is clear that CLT will expand in the future. In each scenario, we assume that the future outcome is suddenly known now. The expansion is spread over several decades depending on the scenario. We have revised this part as below.

In Page 1:

“Results show that higher wood prices reduce the production of traditional wood products but cause productive forestland to expand by around 30.7–36.5 million hectares from 2020 to 2100 and forest management to intensify.”

2.37. Comment: L24: When talking about total net GHG – which sectors are included? Perhaps being specific about sector(s) could be useful.

Response: The land use sectors that will change because of CLT are forestry and farmland. The industrial sectors that will change are forestry, sawtimber, pulp and paper products, steel, concrete, and electricity. We have revised the part in the abstract.

In Page 1:

“Through 2100, producing an average of 3.6–9.6 billion m³ of CLT will drive a total net reduction of life-cycle GHG of 25.7–39.4 GtCO₂e which is largely driven by the increased carbon stock in forests and CLT panels along with GHG reductions by traditional wood products and potential benefits by substituting traditional building materials and market electricity.”

2.38. Comment: L52: Missing citation for previous studies.

Response: This refers to the previous CLT studies discussed in the paragraph above in the last version submitted. We have revised the statement in this version submitted.

In Page 3:

“GTM captures the ability of renewable forestry to anticipate future demand increases through two different channels.”

2.39. Comment: L53: Seems more like a model output rather than a statement for the introduction section.

Response: Revised

In Page 3:

“First, adding the demand for mass timber products will likely lead to higher wood prices which will reduce the market for IWP. Second, higher wood prices will give landowners an incentive to plant more forests and increase forest management intensity. This likely increases the future supply of wood but incidentally also increases the carbon stored in forests.”

2.40. Comment: L55: “Growing wood” is a strange term.

Response: Substitute “plant more forests and increase forest management intensity”.

2.41. Comment: L78: Results sections should ideally not start with stating an assumption.

Response: We have revised the first two paragraphs in Results section.

2.42. Comment: L86-89: Ref to SI Fig. 1 missing to improve readability.

Response: Thanks. We have added the references to SI Fig. 1 in the main text and added the place recording the details.

In Page 5:

“We also look at fast adoption with a high demand scenario (Scenario 3) reaching 60% by 2050 (see Methods and Supplementary Figure 1).”

2.43. Comment: L216: Perhaps cumulative is a better word (instead of accumulative)?

Response: Thank the reviewer for this recommendation. We have changed it throughout the manuscript and SI.

2.44. Comment: Fig. 4 – Can we see global numbers in SI (sum of all regions)?

Response: Thank the reviewer for this suggestion. For the convenience of the readers, we directly put the global numbers in Fig. 4.

Fig. 4. Forest area changes in three scenarios compared to baseline. a, forest area changes by 2050; b, forest area changes by 2100. There are 16 regions: WEU (West Europe), US, CHINA, CANADA, SOUTH ASIA, RUSSIA, EEU (East Europe), E ASIA (East Asia), BRAZIL, JAPAN, CENT AMERICA (Central America), SSAF (Sub-Saharan Africa), OCEANIA, AFME (North Africa and Middle East), RSAM (Rest of South America), and SE ASIA (Southeast Asia). *In GTM, plantations are defined as monoculture forests, even-aged systems that are intensively managed for pulpwood and sawtimber production. Examples of plantation systems in the United States include loblolly, slash, and long-leaf pine, and Douglas fir. Managed forests could originally have been naturally regenerated, but they will be replanted upon harvest. The key distinction between plantations and managed forests is that plantations*

are on highly productive land and involve much more costly and intense management (as in a farm). Natural forests are defined as inaccessible, unmanaged, and naturally regenerating forests.

2.45. Comment: L325: baseline (scenario?)

Response: Yes, we have added that now. Thanks.

2.46. Comment: L328: Are LCA models feeding into GTM or the other way around?

Response: In this study, LCA models are not feeding information to GTM. But GTM is feeding information to the LCA models related to detailed forest carbon stocks and timber output in each region in the period of 2020-2100.

2.47. Comment: L350: Extent possible in which sense?

Response: Thank you. The original sentence refers to calibrating the model to regional forest inventory using FAO database. We have rephrased this part in section Methods.

2.48. Comment: L365: Industrial (roundwood?) demand?

Response: Yes, we have revised the sentence as shown below. Thanks.

In Page 21:

“Total industrial wood demand incorporates separate demand functions for sawtimber and pulpwood.”

2.49. Comment: L375: Missing citation.

Response: We have revised this part. Thanks.

2.50. Comment: L409: Perhaps above ground is a better term?

Response: Yes, we have changed that throughout the manuscript based on the suggestion from the reviewer. Thanks.

In Page 22:

“Aboveground carbon $C_{a,t}^i$ accounts for the carbon in all components of the *aboveground living tree*, as well as carbon in the forest understory and the forest floor, but does not include dead organic matter in slash, which is contained in a separate pool.”

2.51. Comment: L433: Missing citation

Response: We have revised the statement. Thanks.

In Page 23:

“Decomposition rates ϑ^i differ, depending on whether the forest lies in the tropics (3% yr⁻¹), temperate (5% yr⁻¹), or boreal zone.”

13. Favero, A., Baker, J., Sohngen, B. & Daigneault, A. Economic factors influence net carbon emissions of forest bioenergy expansion. *Commun. Earth Environ.* 4, 1–9 (2023)

2.52. Comment: Eqn. 12 – Is all HWP carbon considered sequestered? Eqn 14:16 does contain a decay parameter according to IPCC methodology. Perhaps the LCA model accounts for this already?

Response: Yes, we separately tracked all the biogenic carbon sequestered by trees for IWP ($-\frac{44}{12} \times \Delta Timber C_{t,n}^l$), biogenic GHG emissions by production ($\frac{44}{12} \times \Delta Timber C_{t,n}^l \times (1 - f_m^l)$), fossil GHG emissions by production ($EF^l \times \Delta Timber C_{t,n}^l$), and biogenic GHG emissions due to the decay of produced IWP ($\frac{44}{12} \times ((\Delta IWP_C_{t-1,n}^l + \Delta IWP_In_{t,n}^l) - \Delta IWP_C_{t,n}^l)$).

2.53. Comment: L490: Hypothetically, if a building is made from the same batch of CLT and arrives at the end-of-life at the same time – would whole buildings be dismantled and recycled.

Response: We thank the reviewer for this comment. Combining with the comment from Reviewer 1, we now add two more End-of-Life cases for the CLT panels to test the effects on our results, given there is no data given the young age of existing CLT buildings.

In Page 12:

“To explore the impacts of varied CLT end-of-life cases on the results, two additional conceptual end-of-life cases of CLT panels are conducted: 1) material recycling case with 50% recycling

and 50% landfilling; 2) energy recovery case with 100% CLT panels combusted for power generation. The impacts from the different end-of-life cases of CLT panels are minor (see results in Supplementary Tables 3 and 4) due to the small quantity of CLT that reached their end-of-life (only those adopted in 2020-2040 in Fig.2, given a life span of 60 years).”

Reviewer #3

3.1. Comment: The manuscript presents an interesting study of potential future carbon sequestration in construction by using cross-laminated timber for mid- and high-rise buildings. It investigates the carbon storage in forests and buildings and considers partial-equilibrium timber market effects as well as life-cycle carbon impacts.

The manuscript is well written, and the results are clearly communicated. However, some aspects are not explained.

Response: We thank the reviewer for acknowledging the contribution of our study and providing helpful recommendations and comments. We have carefully addressed the comments and suggestions as shown below on a point-by-point basis.

3.2. Comment: The largest carbon savings are from carbon storage in land converted to forestry given the increased wood prices. However, it appears as if the land is treated as having no carbon before, or in an alternative scenario (Fig.5). This appears problematic. A short description of land-use related carbon storage before and after afforestation would be required.

Response: If the land is coming from farmland, we assume that there is no above ground carbon there before. If the land is coming from natural forestland, we assume it contains the average aboveground carbon for that specific forest type. Fig. 5 shows the changes in carbon stock relative to baseline.

3.3. Comment: The demand for CLT is driven by the demand for floor area. However, it is not explained how this demand is derived. Is it based on a cohort-based stock-flow model for buildings, as they become increasingly common in industrial ecology, or just on a correlation with per-capita GDP?

Response: We thank the reviewer for this question. The demand for floor area in the future, or the newly constructed floor area for urban commercial and residential buildings (mid- and high-rise) is dependent on GDP per capita and urban population (Güneralp et al., 2017). The historical data of floor area data we use to train the regression models are from the material stock-flow model for buildings by Deetman et al. (2020), which is exactly as the reviewer mentioned. To better clarify this point, we have added the description to the main text. At the same time, all the details are available in SI Note 5 Projections of cross-laminated timber demand.

In Page 28:

“The historical data of urban residential and commercial building floor areas from 1970 to 2010 are derived from the study by Deetman et al. for training the regression model³⁸.”

References:

Güneralp, B. et al. Global scenarios of urban density and its impacts on building energy use through 2050. *Proc. Natl. Acad. Sci. U. S. A.* 114, 8945–8950 (2017).

Deetman, S. et al. Modelling global material stocks and flows for residential and service sector buildings towards 2050. *J. Clean. Prod.* 245, 118658 (2020).

3.4. Comment: The supply of timber is somewhat surprising (Fig 2). There is no common pattern across the scenarios. It seems like initially, most timber comes from Europe in Scenario 1 and none in Scenario 2, while the situation for Brazil is vice versa. There is no explanation for why this pattern persists. It appears that the only difference between the scenarios is in the demand for CLT, or did I miss something? The text explains that there is international trade but does not explain whether there are any transport costs assumed (transport would also be source of emissions). It would be nice to know where the demand occurs and these CLT buildings will be constructed.

Response: Note that the country differences in supply generally just expand with size when comparing the high demand and medium demand scenarios. What is noted by the reviewer is the big difference between the fast and slow medium demand scenarios. Brazil and the United States are predicted to plant new forestland to supply CLT. The supply from Brazil and the United States is therefore delayed. With fast adoption, the CLT must come from existing trees so that the Rest of the World and especially Western Europe will be the early suppliers. We now discuss this in the text.

In Page 6:

“Fig. 2 shows the global sawtimber supply for CLT in three scenarios. The annual global wood supply for CLT in Scenario 1 and Scenario 2 reach about 300 million m³ (around 9% of total wood supply for IWP, see Supplementary Table 1) by 2100, whereas the high demand scenario reaches 634 million m³ (19% of total wood supply for IWP, see Supplementary Table 1). Because CLT likely

will come from softwoods, GTM predicts CLT will likely be made from wood that comes from three regions: Western Europe, the United States, and Brazil (Fig. 2). This result is not surprising since these regions together produce more than 60% of the traditional global wood supply in the baseline scenario. It is cost-effective for these regions to switch a portion of their production from IWP into CLT. The United States and Brazil are also predicted to plant trees for CLT. When there is a rapid immediate increase in demand, the early wood supply for CLT comes from existing trees in the Western Europe and Rest of the World. For example, in Scenario 3 with high CLT demand and fast adoption, Western Europe supplies 43.0%–66.2% of wood for CLT from 2020 to 2100. But in Scenario 2, the wood supply for CLT from Western Europe decreases from 54.0% in 2060 to 11.6% in 2100.”

3.5. Comment: With respect to demand, I see it as strange that the per-capita floor area for urban commercial buildings is assumed to remain the same while the GDP rises substantially. Also, the assumption is that few buildings today are with steel and concrete. I have no statistics, but at least for East Africa and Nigeria, almost all urban buildings are of concrete. The forest area coverage changes in Fig.4 do not reflect the patterns of harvest in Fig.2, which is also peculiar.

Response: There will be a small increase in per capita floor area as a result of GDP increasing but larger impacts from continued population growth and urbanization. For example, according to SSP2 (see Supplementary Data), the world urban population increases very slowly from 2060 to 2100 by increasing 614 million, compared to increasing 2435 million from 2020 to 2060. At the same time, the urban population of some countries/regions, like China, decreases after 2050. This makes the annual newly constructed floor areas remain at the same level after reaching the plateau. For steel and concrete, the reviewer is right. Our assumption in BAU is that most buildings (other than single family) would be steel and concrete. The detailed data are available in Supplementary Data (e.g., North Africa and Middle East has more than 75% of the urban apartment and high-rise buildings are made of steel and concrete). The CLT scenarios are discussing replacement of steel and concrete by CLT. To clarify this one, we have revised the manuscript.

In Page 28:

“In equation (18), $NewA_{i,t}$ is the annual newly constructed urban commercial and residential building floor areas *that adopts steel and concrete structures* in year t and region n that can be potentially replaced by CLT.”

3.6. Comment: Fig. 1: End-of-life for HWP considered?

Response: Yes, the production and end-of-life of HWP is considered in this framework. To better exhibit that, we have revised the figure. At the same time, the modeling details for end-of-life for HWP are also shown in the Methods section “GHG emissions from the production and end-of-life of industrial wood products”.

3.7. Comment: Fig. 1: How is the carbon balance of the forests considered? Here, the uptake of the forest is only balanced by wood material flows. How about biomass which degrades in the forest?

Response: We thank the reviewer for the questions. In the forest part, the forest carbon pool has three components, namely aboveground biomass, slash, and soil. The carbon flow input (via biogenic carbon uptake) minus the total carbon flow output (e.g., timber output from the forest, decay emissions) is equal to the net changes of the forest carbon pools. Hence, the net GHG emission balance (total biogenic carbon uptake, decay emissions) of the forest can be evaluated by knowing the forest carbon stock change and timber output. The decay emissions are indeed considered in the soil and slash carbon pools.

3.8. Comment: I recommend setting up a table where variables are defined, as part of the method section. The equations are hard to read as it is.

Response: We thank the reviewer for this recommendation. We now add a table (Supplementary Table 6) to include all the variables that appear in the manuscript in SI, since the journal does not allow tables in section Methods.

In Page 21:

“*The notations used in this study and corresponding descriptions are available in Supplementary Table 6.*”

3.9. Comment: The Excel table should include information on sources, where appropriate. I would also recommend including tables for the information displayed in figures.

Response: We appreciate the reviewer for this recommendation. Following this recommendation, we have now added the data sources to the Excel table and included the source data for the figures we display in the manuscript.

Reviewer #1

1.1. Comment: The paper has undergone significant improvements from its previous version. However, the authors still need to elaborate on and address critical issues regarding the life cycle analysis. These points are discussed as follows.

Response: We appreciate the positive feedback from the reviewer on our last revision. We also appreciate the following helpful comments and recommendations from the reviewer. We have carefully addressed them as shown below and revised our manuscript, SI, and corresponding files.

1.2. Comment: Some underlying assumptions in the paper appear to bias the results in favor of CLT construction and warrant re-examination. Specifically, the recycling rate for steel is assumed to be 35% based on ref.11, with the remaining steel being landfilled. This information contradicts what is reported in the cited study (refer to page 8, paragraph 4). The literature cited actually indicates recycling rates for steel construction materials ranging from 65% to 98%, significantly higher than the 35% used in the present analysis. The 35% recycling rate is notably low for steel, as evidenced by current data from the steel industry. The steel industry's information (e.g. <https://buildsteel.org/why-steel/sustainability/recycles-steel-scrap/>; <https://cssbi.ca/mid-rise-construction/sustainable-steel>; https://www.bir.org/images/BIR-pdf/Ferrous_report_2017-2021_lr.pdf) points to higher recycling rates for steel. It is essential to consider the implications of a higher recycling rate for steel, especially in contexts where materials recovery is expected to become more prevalent in the future.

Response: We thank the reviewer for this helpful recommendation. We checked the references and have now revised the ratio to 65%. This change causes corresponding credits from traditional building materials to decrease by around 10%, leading to a 2% change to the final results of GHG emissions. But all the conclusions and comparative results remain the same. We have revised the related contents throughout the manuscript, SI, and Supplementary Data. Due to the length, we kindly ask the reviewer to refer to these files. The statement related to the recycling rate is revised as shown below and in the SI. To better point this out, we also added discussions to the manuscript.

In Supplementary Note 2:

“The recycling rate for steel is assumed to be 65% based on the literature¹¹.”

In Page 20, the second paragraph:

“Based on this study, future research can further investigate possible improvements in background processes (e.g., electricity generation, *steel recycling rate*, and concrete production) and foreground processes (e.g., CLT production) in each region.”

1.3. Comment: The authors should provide clarification on how they modeled the recycling of CLT in a scenario involving 50% recycling and 50% landfilling. They need to explain what CLT is recycled into or used for and detail the type of data or Ecoinvent unit processes used to assess the benefits and burdens of CLT recycling.

Response: We thank the reviewer for recommending this. Following this recommendation, we have added corresponding details in the manuscript and SI.

In Page 29, the first paragraph:

“Besides the landfill case, two additional end-of-life cases of CLT panels are conducted: 1) material recycling case with 50% closed-loop recycling and 50% landfilling; 2) energy recovery case with 100% CLT panels combusted for power generation. These robustness checks explore how these alternative assumptions change the results. More details are available in Supplementary Note 2.”

In Supplementary Note 2:

“Two additional conceptual end-of-life cases of CLT panels are conducted: 1) material recycling case with 50% recycling and 50% landfilling; 2) energy recovery case with 100% CLT panels combusted for power generation. For the first additional conceptual case, due to the lack of data, this study assumes 50% of recycling rate with the rest for landfilling^{1,13}. The CLT panels are assumed to be recycled back to be shaped into CLT products¹. The GHG emissions and energy consumption of CLT recycling are assumed to be 50% of the normal producing process¹. Future research could update these inventory data if the inventory data of recycling CLT emerge in the future decades. For the second conceptual case, the CLT panels are combusted to generate electricity to replace the market grid electricity. The GHG emissions of the potentially replaced electricity adopts the global average value from the Ecoinvent 3.9 cut-off database (see Supplementary Table 7)¹².”

Reviewer #2

2.1. Comment: I appreciate the authors' efforts in providing additional background details for their manuscript. However, I still have reservations about the suitability of the chosen modeling framework for addressing the research question at hand. While GTM is a robust model, it may not be the most appropriate tool for this specific research question. Despite the authors' detailed responses to my previous review, some of my concerns remain unaddressed.

Regrettably, based on the current state of the manuscript, I am unable to recommend it for publication in Nature Communications.

Attached, I offer further comments based on the authors' responses.

Response: We thank the reviewer for the time invested in reviewing our revisions and we provide a detailed response to each of the comments and background information below. As the reviewer pointed out, GTM is indeed a robust model, which has been applied to different research questions for its ability to project future forest dynamics under different socio-economic, policy and demand scenarios. We thank the comments and recommendations from the reviewer. We have carefully addressed these comments and suggestions as shown below.

2.2. Comment: “• *The more land that the forest wants from farmland, the more expensive the land gets. This recognizes the price inelasticity of farmland (the underlying demand function for food)*”

Price inelasticity of farmland is also influenced by competition for land, which also comes from other land pools which authors do not cover (non forested land, pasture and rangeland etc.). From the description it feels like the modeling world only consists of farmland (Cropland?) and forests (managed or unmanaged) which is not a justifiable assumption.

Response: We assume that growing timber is restricted to land that can support forests. That includes farmland that used to be forest and currently forested land. Farmland includes rangeland, pastureland and cropland (crops and livestock) which we now state explicitly. Forestland includes tropical moist forest, tropical dry forest, temperate deciduous forest, temperate evergreen forest, and boreal forest. We intentionally don't include non-forest lands. Although there are trees on non-forest land (Crowther et al 2015), the biomass on non-forest land is about one-third of what is on forest land (Xu et al 2021). In order to convert an ecosystem type to a new ecosystem type requires expensive inputs. For example, the drylands must be irrigated. Wetlands must be drained. The cost of doing this is prohibitive and in many cases not sustainable (not enough available water).

2.4 billion people rely on wood for fuel largely in Africa and Asia (UNEP 2019). People tend to rely on their local ecosystem for wood fuel. It is likely that most of the trees harvested in nonforest areas is for

wood fuel. GTM captures the tradeoff between farmland and forestland by charging managed forestland and plantations rent that grows with the aggregate size of the land used by managed forestland and plantations. As the price of timber rises (falls) relative to the price of farmland, the market converts more land into forest (farms).

Reference:

Crowther, T. W. et al. Mapping tree density at a global scale. *Nature* 525, 201–205 (2015).

Xu, L. et al. Changes in global terrestrial live biomass over the 21st century. *Sci. Adv.* 7, (2021).

United Nations Environment Program (2019) “Review of Woodfuel Biomass Production and Utilization in Africa” Kenya.

https://na.unep.net/atlas/datlas/sites/default/files/Biomass_Press_FINAL_WEB.pdf

2.3. Comment:

“• We now make clear in the text that CLT causes only a small change in the global demand for wood. Even with the most aggressive CLT scenario, the global demand for wood increases by only 11%”

That was not the reasoning behind requesting baseline results. The idea was to see how the model behaves in the baseline scenario, and perhaps compare it to historical numbers for validation. Also stating that in the most aggressive CLT scenario results in 11% increase in global demand for (industrial roundwood?) seems low. Between 1995 and 2020, industrial roundwood production increased by ca. 30% and between 1962 and 1992 the industrial roundwood demand grew by 54%. Alternatively, it could be argued that CLT demand did not exist in the past so the demand for industrial roundwood increased without pressure from CLT demand. Also, there is case to be made that this production (or demand as proxy) is declining in recent times, and competition for CLT might reduce it even more, but having only an 11% increase in most aggressive CLT demand scenario seems dicey.

(1962 - 1992) <https://www.fao.org/3/w7705e/w7705e0b.htm>

(1995 - 2020) <https://www.fao.org/faostat/en/#data/FO>

Response: The increase of 11% is relative to the future baseline (without CLT), no relative to historical production.

GTM’s results have been validated to historical data (e.g. See Mendelsohn and Sohngen 2019) and calibrated to the FAO data ([How does GTM manage carbon? The role of removals and growth. | Global Forests \(osu.edu\)](https://u.osu.edu/forest/how-does-gtm-manage-carbon-the-role-of-removals-and-growth/) (<https://u.osu.edu/forest/how-does-gtm-manage-carbon-the-role-of-removals-and-growth/>)).

2.4. Comment:

“• Response to “2.2””

Like earlier, this seems to be an incorrect interpretation of existing literature. Modeling framework used in Mishra et al. can indeed account for land conversion possibilities. It is quite likely that land which cannot support plantations is never used for establishing new plantations due to biophysical constraints coming from the DGVM used (LPJmL).

Response: Figure 2 in Mishra et al describes how land changes relative to the present. But the key is to look at how land use changes relative to BAU as one intensifies CLT. Figure 2 suggests that increasing CLT is leading to an increase of plantations. What is changing to make space for the increased plantations is a drop in cropland, a drop in secondary forest, and a drop in land in non-forest. For the land to be suitable for a plantation, it must be highly productive. That implies the model is using high valued cropland, high valued forestland, and converting non-forest into high valued forestland. 1) It is not clear there is that much high productive forest in secondary forest. 2) Converting prime cropland into forests is certainly feasible but it would remove the most productive cropland. This would have a dramatic effect on food production just as the plantations would have a dramatic effect on timber production. 3) Converting non-forest land to highly productive forestland would take vast resources. For example, one would have to irrigate dryland. This would place a huge strain on water supplies. Wetlands would have to be drained. There is no evidence that the study took these conversion costs into account.

2.5. Comment:

“• Response to “2.4””

This is more of a global north perspective. I appreciate the historical context provided by the authors but saying that “entire world has turned to renewable forest management recently” warrants either a use of citation or it should come with more context. This could also simply be a difference of terminology used. Perhaps the authors meant “sustainable forest management”? If yes, then a sweeping remark on global practices could likely not be justified.

Response: We have edited the section describing how the model functions and dropped the offending statement above. What we are trying to get across is that most industrial production of timber involves replanting after clearcuts and harvesting sustainably using partial cuts. As a result, the amount of timber harvested has gradually increased over time and yet there is no net loss of biomass in global forests. According to FAO 2020, the global biomass per ha of forestland has increased from 133 m³/ha in 2000 to 137 m³/ha in 2020. This has been a dramatic change in forestry from historical practice.

2.6. Comment:

“• We have added policy recommendations to the discussion.”

I still do not see a solid policy recommendation in edited text. “Further policy studies” is not a solid policy recommendation. Authors have already included a nice statement in their text “Encouraging CLT adoption may be a compelling market tool to increase carbon storage in forests and wood products (...)”. Maybe something along those line could help frame a policy recommendation?

Response: We now make a policy recommendation in the paper that subsidizing CLT would be an effective mitigation activity. At reasonable cost, the world can get an increase in carbon storage and a small reduction in emissions through a subsidy program for CLT.

In Page 18, the last paragraph:

“The results suggest that it would be reasonable to subsidize the demand for CLT as a GHG mitigation strategy. By increasing the market value of timber, the subsidy would provide an additional market incentive to maintain the world’s existing forest. Given the difficulties of regulating global land use, this subsidy would be a practical tool to increase carbon storage in forests. There are also small gains from reduced GHG emissions and storage of carbon in long-lasting wood products. Encouraging CLT adoption will increase forestland by 30.7–36.5 Mha especially in the long run and reduce carbon in the atmosphere.

However, not every consequence of increasing CLT is beneficial. Despite the large overall increase in forestland, GTM predicts natural forestland will fall by 10.6–18.9 Mha. There is a tradeoff between conservation and carbon goals in these scenarios. Widespread expansion of CLT will likely reduce natural habitat. These land use effects will vary across the planet with substantially increased forestland in the United States and Western Europe, but large reductions in natural forestland in the tropics. Stricter protections of natural lands will reduce this side effect on conservation, but the regulations will also increase the cost of CLT and reduce the carbon stored. There is consequently a limit to how far one should expand CLT and timber production in general. The CLT scenarios we present in this paper are sustainable but increasing subsidies further may not be. Even with the scenarios we present, moving from medium demand to high demand has additional benefits. Future research should look at these tradeoffs carefully and make clear what the consequences of alternative policies will be. There are tradeoffs between conservation and mitigation policies that affect land use. It is critical that policy makers be aware of those consequences.”

2.7. Comment:

“• NDIs measure plantations which are one of the outcomes predicted by GTM”

Requesting the authors to stick to the correct nomenclatures. NDCs (or existing NPIs) likely often refer to “planted” forest category for expanding forest cover, and even more specifically almost point to “other planted forest” category from FAO nomenclature. Additionally, GTM probably does not include these other planted forests.

Response: We have carefully avoided terms such as “NDI”, “NPI”, and NDC in the revision.

2.8. Comment:

“• The reason for not seeing a bigger increase in plantations is that forestland must be highly productive to support a plantation and only a small fraction of forestland fits that description.”

Does that mean that we have already reached a saturation point in establishment of new forest plantations? Given the historical development of plantation area, this does not appear to be the case.

Response: We are not stating that there can be no more plantations. We are simply arguing that the magnitude of plantations suggested by Mishra in the most aggressive case would require using a vast share of the most highly productive land in the world for forest plantations. That would be very expensive because most of this land is already in use as prime cropland. We do not mean to imply that plantations cannot expand at all.

2.9. Comment:

“• GTM models all managed forestland in the world by age class. This was done by starting with natural forests and carefully modeling when forests were harvested and replanted using historic harvests from Houghton (2008). Natural forests are assumed to have an average natural stocking given the forest type and region.”

Houghton et al. paper(s) from 2008 mostly refer to carbon fluxes but not specific to age-classes?

Perhaps adding the exact source would help readers. Also, is that the only source of data available to calibrate historical age-class distribution in GTM?

Response: The European managed forests have been even aged forests since the 19th century. But global forests have been turned into even-aged forests by clear cut harvesting that accelerated in the twentieth century. The Houghton et al 2008 paper measures large scale clearcutting starting in the early twentieth century by decade. Some of this deforested land was converted to farmland, but the land that naturally regenerated became even aged stands of forest afterwards. Natural regeneration takes longer than replanting, but it still leads to an even age stand tied to when it was deforested. The replanting after harvest that began in the US in the 1950’s and became more global by the 1980’s created more precise even-aged stands. There is additional USFS data for the US that calibrates acreage by stand age. This is consistent with the age stand distributions in GTM.

2.10. Comment:

“• (...) 27% is managed forestland (of which 9% is plantation)”

This is way off from FAO reported numbers. Not all planted forest is “plantation”. From FAO – “Ninety three percent of the forest area worldwide is composed of naturally regenerating forests and 7 percent is planted. The area of naturally regenerating forests has decreased since 1990 (at a declining rate of loss), but the area of planted forests has increased by 123 million ha.”

Response: FAO 2020 says there is globally 4,106 million ha of forest, 294 million ha of planted forest, and 131 million ha of plantations in 2020. GTM is consistent with this FAO data. In GTM, 73% of global forest is “Unmanaged/natural” forest that are NOT managed for timber production. Only 27% of total forest area is managed forest for timber. 9% of the managed forest is plantations. The stock of managed forests includes a great deal of naturally regenerated forests.

2.11. Comment:

“• Response to “2.22””

Land cannot be treated in isolation. “CLT is not likely to cause other land uses to change” points that there are not enough complexities in the modeling framework to link all land pools.

The assertion that CLT is unlikely to influence other land uses appears simplistic, considering potential impacts on agricultural production, demand and land-use.

Response: We appreciate the referee requesting the paper be clear. We have revised the paper to be more precise. The paper argues timber can readily be produced on land suitable for forests but not on land that is not suitable for forests. Timber can be grown in boreal forest, temperate deciduous forest, temperate evergreen forest, tropical moist forest, and tropical dry forest. Land in farms (livestock and crops) largely comes from one of these forest ecosystems so that a great deal of farmland is suitable for growing timber as well. GTM looks at the effect of CLT on all of these lands. Of course, previous papers also considered these land substitutions.

What GTM does not do is assume that non-forest lands can also be used to grow timber. This includes land in tropical grassland, tropical savannah, wetlands, temperate grassland, desert, or tundra. Although there are trees on some of these non-forest lands (Crowther et al 2015), the biomass on non-forest land is about one-third of what is on forest land (Xu et al 2021). It is not practical to use these non-forest lands for growing timber, much less plantations. It is extremely expensive to convert non-forest lands into forest lands. For example, one would have to irrigate drylands. One would have to drain wetlands.

2.12. Comment:

“• *Response to “2.26”*”

The model presented in Mishra et al. never claims that “143 Mha of prime cropland” is removed.

Response: This statement was in the response and not in the manuscript. We believe that Mishra et al did not consider the required productivity of land needed to grow plantations. But it is important to understand that the productivity of land needs to be recognized. In order for land to be suitable for a plantation, it must be highly productive. It cannot be marginal farmland. Doubling or tripling the land in plantations implies the reduction in farmland must come from productive cropland. Taking this much land from cropland will have a significant effect on aggregate crop production.

2.13. Comment:

“• *Response to “2.28” and code/data availability*”

SRES drivers are only used for sensitivity analysis or are they also used as model driver in all scenarios (B2 maybe?). SRES B2 is mostly akin with SSP2 numbers but isn't SRES socioeconomic drivers quite old already at this point?

This is also the reason why I would have liked to see how well the baseline scenario behaves in comparison to historically available data (sans CLT signals). Also, it would be nice to see this sensitivity analysis in SI somewhere rather than taking author's word for it. As a side note, SSPv3 was released few weeks ago, but a good justification of using SRES numbers instead of SSP numbers (even SSPv2) could be useful.

The model code shared does appear to be reflecting that some SSP numbers are used but then why use SRES scenarios for sensitivity? Some comments in the GAMS code refer to the IIASA database which seems to point that at least some SSPv2 numbers were plugged in.

Also, with the model code shared on zenodo, some input files are missing

- param2 and param3 (parameters for forestry model)
- param4 (harvesting costs)
- cparam2 (carbon estimations)
- forinv2 and iforin2 (accessible and inaccessible forests)

So, we have no good way to know what is being fed to the model. Without being able to compile the model it becomes difficult to coherently see model dynamics.

For example:

→ Population and GDP growths are hardcoded with only time index. Are the growth rates same for all regions? Or is GTM only doing global numbers for both population and gdp (per capita).

→ Both GDP and population numbers are fed into AF which influences AFP (and AFS) which

influences some more calculations/equations down the line. But these are also with only a time index (no regional information).

Response: SSP2 provides the income and population estimates over time that drive both the future demand for timber and the future demand for CLT. We have a consistent economy driving all elements in the model. We chose SSP2 because it is consistent with the best estimate of future economic projections used in economic modeling. We leave for future research the question of how sensitive the results are to different plausible economic scenarios.

For the files mentioned by the reviewer, we have updated the online data repository on Zenodo (<https://doi.org/10.5281/zenodo.13334682>). The files mentioned by the reviewer can be found under the “Austinetal2020”. At the same time, more information and files related to GTM can be also found at: <https://www.dropbox.com/scl/fo/44h3ro8s2vpk6nt2w6ycb/AO8qXWkHlp13vOSDjtYr8bs?rlkey=au6dkpz4z1ofkjbe4bey318ac&e=2&dl=0>.

Reviewer #3

3.1. Comment: The authors have addressed the review comments well. They have proactively conducted additional analysis, expanded the supporting information, and published the code and data. This is very commendable. I realize that demand scenarios for buildings have not been a focus of this work. Rather, the authors have taken a prominently published scenario as a starting point. While this is a completely legitimate approach, I am not sure about that scenario and would have liked to see other scenarios explored, in particular ones that address the substantial unmet housing demand in developing countries. But their scenario is legitimate, and it is the authors' choice.

I see this paper has a substantial interest. The central message is that the increase of wood-based construction will lead to a substantial increase in the carbon stock of growing trees which surpasses the carbon stored in the timber. It is an insight that is novel, and it is robust to remaining issues raised by me or the other reviewers. It is important for policy, including policy processes that I am involved in, and so I would welcome the speedy publication of this paper.

Response: We thank the reviewer for the very positive opinion and feedback on our work and thinking “this paper has a substantial interest” and “it is important for policy”. We appreciate all the helpful comments and recommendations from the reviewer in the last round to improve the quality of our work. We agree with the referee that examining alternative economic scenarios is an important area for further research.

Reviewer #4

4.1. Comment: This paper was well written overall with deep studies on global CLT supply and demand. There are some comments that require authors' attention.

Response: We appreciate that the reviewer thinks our paper was well written overall with deep studies on global CLT supply and demand. We thank the reviewer for providing the following valuable and helpful comments and recommendations. We have thoroughly addressed these comments and recommendations and revised our work accordingly.

4.2. Comment: Page 8 Line 156, there is grammar error.

Response: We thank the reviewer for this point. We have revised this sentence as shown below and in the manuscript.

In Page 8, the first paragraph:

“Non-forest ecosystems (savannah, parkland, and mangroves) require expensive inputs to make them suitable for growing timber.”

4.3. Comment: The demand increase on CLT would definitely lead to more plantation for saw lumber. However, demand of other wood products (wood studs, composite woods) will still be required. Is this possible that the global market could not provide enough stock to growing CLT demand (assumed in this paper under different scenarios)? Some countries are regulating to limit the harvesting of forest woods.

Response: Yes, it might be a possibility under a very high CLT demand scenario that we are not considering in this study since it is not supported by current projections on CTL consumption. For instance, under a combination of specific assumptions where short-term demand is very high (E.g. as high as current global demand of sawtimber) and some/all regions do not allow forests conversion to plantations and land rents are very high (it is very expensive to convert cropland to forests), it is likely that the stock of biomass is not enough to meet the demand for CLT.

4.4. Comment: In the building industry, the AEC professionals are working to improve the circularity of buildings by reusing the materials. As per the assumption period in this paper (80 years from 2020-2100), the building designs might push to allow all the CLT materials to be reused, which would lead to less CLT requirements in the future scenarios. This aspect should definitely be considered in the “Discussion” section.

Response: We thank the reviewer for this very helpful recommendation. We agree that this aspect needs to be discussed in the discussion section. The improvement of CLT circularity could affect the future CLT demand. But within 2020-2100, given the long period of CLT life span, this effect on our GHG balance results could be minor. This has been demonstrated by the end-of-life cases explored in this study. The impacts from the different end-of-life cases of CLT panels are minor (see results in Supplementary Tables 3 and 4) due to the small quantity of CLT that reached their end-of-life (only those adopted in 2020-2040 in Fig.2, given a life span of 60 years). To better clarify this point following the reviewer’s recommendation, we now have added the discussion related to this aspect in the discussion section.

In Page 19, the second paragraph:

“From now through 2100, alternative assumptions about the end-of-life cases of CLT panels have only minor impacts on the results. The long life span of CLT panels implies few panels will reach end-of-life this century. With future improved circularity of mass timber products beyond 2100 (ref.²⁰), this could potentially reduce the timber demand for CLT, and further impact the forest carbon stock and timber supply.”

4.5. Comment: Right now, applying mass timber (including CLT) into building design would count the carbon sequestration as biogenic carbon. However, if the mass timber materials could be designed to be reused in a few life cycles, then the biogenic carbon might be treated as embodied carbon reductions. Is this a potential scenario to be considered and discussed?

Response: We thank the reviewer for this question. In this study, we accounted for the life-cycle carbon flows (both biogenic and fossil-based) across the forest, industrial systems and the entire life cycles of CLT materials at the global level, including the biogenic carbon uptake throughout to end-of-life. At the same time, we also explored two additional end-of-life cases of the CLT: 1) material recycling case with 50% recycling and 50% landfilling; 2) energy recovery case with 100% CLT panels combusted for power generation. The impacts of different end-of-life cases of CLT panels on our global results within the timeframe 2020-2100 are minor (see results in Supplementary Tables 3 and 4) due to the small quantity of CLT that reached their end-of-life (only those adopted in 2020-2040 in Fig.2, given a life span of 60 years). The varied circularities of CLT could potentially affect the product-level life-cycle GHG emissions, given the different allocation methods and methods of treating upstream GHG emissions and biogenic carbon, as suggested by the reviewer. To clarify this point as suggested by the reviewer, we have added more discussion in the manuscript.

In Page 19, the second paragraph:

“From now through 2100, alternative assumptions about the end-of-life cases of CLT panels have only minor impacts on the results. The long life span of CLT panels implies few panels will reach end-of-life this century. With future improved circularity of mass timber products beyond 2100 (ref.²⁰), this could potentially reduce the timber demand for CLT, and further impact the forest carbon stock and timber supply. Future research could investigate how alternative circularities of CLT affect the global carbon consequences of adopting CLT in the long-term. At the same time, alternative recycling rates of CLT could also impact the product-level life-cycle GHG emissions of CLT panels by affecting the upstream GHG emissions. But this depends highly on the allocation methods.”

4.6. Comment: The CLT environmental impacts might vary in different countries. Does the author ensure the accuracy of the impact values? Are the harvesting techniques variances and impacts to the soil considered?

Response: We thank the reviewer for this question. In this study, we focus on quantifying the global change in life-cycle GHG emissions and carbon stock of adopting CLT compared to the baseline scenario. To ensure the accuracy of our analysis, we combined the economic-ecological model with LCA model. For the LCA model, we evaluated the life-cycle GHG emissions based on the process models. The parameters we used in the LCA model were globally averaged since we were focusing on the global impact. This is due to the reason that the lumber (for CLT production) and CLT production locations largely depend on international trade. Since it remains unknown where the lumber for CLT and CLT will be produced, it would be not feasible to use region- or country-based data for the GHG emissions of energy and materials for CLT production. Hence, we had to use the global average data for the upstream GHG emissions of the energy and materials. All the details of the LCA modeling can be found in the section “Life cycle assessment” in the manuscript, Supplementary Notes 2-4, and Supplementary Tables 8-11. The economic-ecological model, the GTM, quantified the carbon stocks of standings, slash, and soil, and the harvested timber for CLT and industrial wood products. The GTM considered the detailed variations in 16 regions related to forest area change, forest type change, timber supply and demand. The details of GTM can be found in the section “Global timber model” in the manuscript. This method has been validated in the previous publications¹¹⁻¹⁴.

In GTM, estimates of soil carbon in forests include carbon stored in mineral and organic soils (including peat) and they are unique for each region and forest type. GTM models changes in soil carbon storage from land use change (land converted to forests) but it does not capture nuanced soil carbon dynamics associated with forest operations (see Favero et al. 2023). Our analysis does consider the effects of CLT demand on forest carbon across different pools (including soil carbon) but the change relative to the

scenario without CLT demand is very small (e.g. the largest increase is equal to 0.05% in 2100 under the high CLT demand scenario).

Reference:

Favero, A., Baker, J., Sohngen, B., & Daigneault, A. (2023). Economic factors influence net carbon emissions of forest bioenergy expansion. *Communications Earth & Environment*, 4(1), 41.

4.7. Comment: In Figure 3, the global prices increase in the next 40 years, how are these trends would help mass timber competing with concrete and steel from a developer's perspective, assuming concrete and steel price stay stable? The policy support factors and implementation strategies should be emphasized here for the decision makers.

Response: This is a great point, and we have included it in the discussion.

In Page 19, the last paragraph:

“Some market dynamics are not included in this study that might affect the competitiveness of CLT relative to other materials. For instance, we show that as CLT compete with other timber products, it will drive up average global prices of timber. If CLT becomes too expensive, it may face more difficulty replacing steel and cement (assuming no carbon prices will be applied to them). It would be attractive to explore this more carefully using an economy-wide economic model to assess these dynamic tradeoffs.”

4.8. Comment: The title should be CLT not mass timber products as it does not cover all the other mass timbers.

Response: We appreciate the reviewer for this recommendation. We have changed the title to: “Global Land and Carbon Consequences of Mass Timber Products: *the Case of Cross-Laminated Timber*”.

Summary of Last Round Review Response (Reviewers 1–4) and Actions Adopted

In the last round of revision, we received comments and recommendations from 4 reviewers. In this round, we received comments and recommendations from 3 reviewers. We thank all the reviewers for their helpful comments and recommendations to improve our work and thank the editor for handling this manuscript. Here, we provide a one-page summary of the review response and actions adopted for the **last round** with the detailed responses shown in **Appendix A**. The detailed review response of **this round begins after the one-page summary**.

Reviewer 1. We thank reviewer 1 for the two comments and the comment that our paper has “**undergone significant improvements** from its previous version”. First, we changed the recycling rates for steel from 35% to 65% and revised all the results accordingly. Second, for the comment related to exploring more end-of-life cases, we added two more conceptual end-of-life cases of CLT panels: 1) a material recycling case with 50% recycling and 50% landfilling; 2) an energy recovery case with 100% CLT panels combusted for power generation.

Reviewer 2. We thank reviewer 2 for the 13 helpful comments and recommendations. Based on these comments, we explicitly defined land categories, clarified land competition assumptions, and provided further justification for the suitability of GTM in modeling future forest dynamics and CLT expansion. The revisions emphasize that GTM is well-calibrated to historical FAO data and has been validated in prior studies. We clarified that the 11% increase in wood demand is relative to a future baseline, not historical trends, and reaffirmed that GTM accounts for productive land availability and conversion costs. Revisions also emphasize the biophysical constraints that limit plantation expansion and the trade-offs between cropland conversion and timber production. We refined our policy discussion by explicitly recommending subsidies for CLT as a viable GHG mitigation strategy while acknowledging trade-offs between carbon storage and conservation goals. We updated the online data repository with missing files to facilitate model replication and provided additional clarifications on the use of SSP2 economic projections as consistent drivers for future timber and CLT demand. We think these revisions significantly improve the manuscript thanks to the help from the reviewer.

Reviewer 3. We thank the reviewer for the very positive opinion and feedback on our work, and thinking “**this paper has a substantial interest**” and “**it is important for policy**” and welcoming “**the speedy publication of this paper**”.

Reviewer 4. We thank the reviewer for thinking our paper “was well written overall with deep studies on global CLT supply and demand” in the last round, along with the **recommendation for a publication in this round**. In the last round, we corrected grammatical issues and revised the title. We expanded the discussion to consider future advancements in CLT reuse and circularity. While the impacts on GHG balances within 2020-2100 are minor due to CLT’s long lifespan, we acknowledged that future research should assess the long-term implications of improved CLT circularity. We clarified how life-cycle carbon flows are accounted for and added a discussion on how alternative CLT recycling scenarios could affect product-level GHG emissions, depending on allocation methods. We explained why global average data were used for LCA due to the uncertainty of future CLT trade and production locations. We also clarified that while GTM accounts for soil carbon storage and land-use changes, it does not capture nuanced soil carbon dynamics from forest operations. We emphasized the importance of policy support and implementation strategies for decision-makers, reinforcing the relevance of our findings for sustainable construction and carbon mitigation strategies.

More details of the responses and actions in the last round are available in **Appendix A** at the end of this file.

Responses to the Reviewers' Comments in This Round

Reviewer #4

4.1. Comment: I reviewed this article last time. The reviewer revised the article appropriately, and I am happy to recommend it for publication.

Response: We thank the reviewer for recommending the article to be published and for all the valuable comments and recommendations from the last round.

Reviewer #5

5.1. Comment: While I find this manuscript useful, I see a need for further improvement in three main areas.

Response: We appreciate the positive feedback and all the valuable comments and recommendations from the reviewer. For the following comments and recommendations, we have carefully addressed them on a point-by-point basis. All the revised places are highlighted in the manuscript and SI, and italic in this response document.

5.2. Comment: I am disappointed to see the authors' lack of comprehensive review on the related recent literature. One of the research gaps highlighted in the paper is that prior study didn't evaluate the market impacts of increased mass timber demand. This is not true. Nepal et al. (2021) evaluated global wood products market impacts of increased mass timber demand. You might want to emphasize how your evaluated impacts support or contradict with prior studies instead of claiming that no prior studies have evaluated the market impacts of increased mass timber demand. I believe that your evaluation of mass timber-induced increases on planted forests and intensified management activities (via timber price increases) can be considered a newer contribution, which you can emphasize in explaining how your studies complement prior studies in this topic area.

Response: We appreciate the recommendation from the reviewer on this point. We have searched the literature again, added more articles to our literature review part, and revised this part as recommended by the reviewer. Following the reviewer's suggestion, we discussed our contribution to evaluating how different forest types, management activities, and the timber market will change due to the new demand for mass timber. We also discussed that our work coupled a cradle-to-grave LCA with the economic model to investigate the carbon consequences throughout the life cycle stages of CLT, pulp, and sawtimber products, including forest carbon stock change, CLT production, CLT end-of-life, the potential benefit from byproducts of CLT production, the avoided GHG emissions from traditional construction materials, and the GHG emissions caused by changes in producing pulp and sawtimber products.

In Page 2, the first paragraph:

“Previous studies have shown the potential carbon benefits of adopting mass timber products over traditional construction materials⁶⁻⁸. Life Cycle Assessment (LCA) has been employed at the product level (e.g., 1 m³ wood product)^{9,10} and the stand level (e.g., 1-ha forest land)¹¹ to quantify the environmental performance of mass timber products. Although these studies advanced the knowledge of CLT, most of them only include the direct impacts associated with wood product life cycles (e.g., forest growth,

harvesting, wood production, product use, and end of life). They do not consider the effects of increasing wood demand on the life-cycle GHG emissions of both traditional wood products (e.g., lumber, particle boards, pulp and paper products) and mass timber products, and the long-run implications to land use and forest management¹⁰.”

In Page 2, the last paragraph:

“Several studies have investigated the global forest impact of mass timber adoption^{6,12,13}, recognizing the value of using CLT as part of a potential global climate mitigation program. However, these studies have not considered how various forest types, forest management activities, and the timber market will evolve in response to the new demand for mass timber. Moreover, they did not consider the carbon consequences throughout the life cycles of both CLT and traditional wood products, from forest to the end-of-life.”

References:

6. Churkina, G. et al. Buildings as a global carbon sink. *Nat. Sustain.* 3, 269–276 (2020).
7. Nepal, P. et al. The potential use of mass timber in mid-to high-rise construction and the associated carbon benefits in the United States. *PLoS One* 19, 1–18 (2024).
8. Taylor, A., Gu, H., Nepal, P. & Bergman, R. Carbon Credits for Mass Timber Construction. *Bioprod. Bus.* 8, 1–12 (2023).
9. Chen, C. X., Pierobon, F. & Ganguly, I. Life Cycle Assessment (LCA) of Cross-Laminated Timber (CLT) produced in Western Washington: The role of logistics and wood species mix. *Sustainability* 11, (2019).
10. Athena Sustainable Materials Institute. A Life Cycle Assessment of Cross-Laminated Timber Produced in Canada. <http://www.athenasmi.org/resources/publications/> (2013).
11. Lan, K., Kelley, S. S., Nepal, P. & Yao, Y. Dynamic life cycle carbon and energy analysis for cross-laminated timber in the Southeastern United States. *Environ. Res. Lett.* 15, 124036 (2020).
12. Mishra, A. et al. Land use change and carbon emissions of a transformation to timber cities. *Nat. Commun.* 13, (2022).
13. Nepal, P., Johnston, C. M. T. & Ganguly, I. Effects on global forests and wood product markets of increased demand for mass timber. *Sustain.* 13, (2021).

5.3. Comment: However, the effects of on added planted forests and increased forest management activities, given a modest projected increase in timber price (up to 59% for sawtimber and up to 22.8% for pulpwood by 2100) seems substantial. It is important that the authors review prior studies evaluating the magnitude of increases in planted forests and management intensities due to price increases and clarify

whether the magnitude of the large projected effects in this paper is supported by the prior studies. These papers do not necessarily need to be mass timber-induced price increases.

Response:

We appreciate the reviewer's comments. There have been several past studies that have explored how various policy or exogenous changes would affect timber prices and thus planted forest and forest management. For example, Sohngen et al. (2001) examined the effects of climate change upon forests. Sohngen and Mendelsohn (2003) examined the effects of a global program that sequestered carbon in the forest. Favero et al. (2107) examined how both sequestration and woody biomass for energy would affect forests. All these studies used GTM, the same model used in this study, to understand how the timber market would change, how timber prices would change, and how the size and management of forests would change. Perhaps the most relevant is the study by Mendelsohn and Sohngen (2019) that used GTM to understand how past deforestation would affect historic forest size, timber prices, and forest management. The model was able to closely predict how timber prices have changed over time and how forests have managed to store slightly more carbon today than in 1900.

References:

Favero, A., R. Mendelsohn and B. Sohngen. 2017. "Using Forests for Climate Mitigation: Sequester Carbon or Produce Woody Biomass?" *Climatic Change* 144 (2): 195-206.

Mendelsohn, R. and B. Sohngen. 2019. "The Net Carbon Emissions from Historic Land Use and Land Use Change" *Journal of Forest Economics* 34(3-4): 263-283.

Sohngen, B. and R. Mendelsohn. 2003. "An Optimal Control Model of Forest Carbon Sequestration" *American Journal of Agricultural Economics* **85** 448-457.

Sohngen, B., R. Mendelsohn and R. Sedjo. 2001. "A Global Model of Climate Change Impacts on Timber Markets" *Journal of Agricultural and Resource Economics* **26(2)**: 326-343.

5.4. Comment: Based on my understanding of the several past studies on the related topic (which authors neglected to review), I feel that your estimated total carbon benefit of increased mass timber demand (4 tCO₂e to 7.2 CO₂e per m³ of CLT) is much higher. Apparently, this is because of your large projected forest carbon contribution (45% to 65%) to total estimated carbon benefits that you showed due to increases in planted forests and managed forests. To give you a perspective, for example, Taylor et al. (2023) estimated carbon benefits (carbon stored in CLT plus avoided GHG emissions) of 1.5 tCO₂/m³ of CLT, using data of actual U.S. specific mass timber building projects. Similarly, in another U.S. specific study, without considering such land use effect, Nepal et al. (2024) finds a carbon benefits of 1.1 t CO₂e per m³ of mass timer adopted in new mid to high-rise buildings in the United States. These numbers are

comparable to yours without the land use and forest management effects and provide reference points to compare/contrast your findings.

Response: We appreciate this comment from the reviewer and the observation that the previous literature numbers are comparable to our estimates when excluding the land use and forest management effects. We also noticed that previous studies did not consider the GHG emission changes caused by the change in pulp and sawtimber products, nor did they include detailed GHG emissions from the end-of-life cases of wood products, which are considered in this study. To isolate the effects of the forest stock changes, we have calculated the CLT carbon benefit of three scenarios to be 2.20-2.25 tCO₂e/m³ if the forest stock (i.e., standing, soil, slash) changes are not included. These benefit ranges align more closely with the previous estimates mentioned by the reviewer. To better clarify this point, we have revised our results and discussion sections in the manuscript to directly compare our results to previous findings.

On page 15, the third paragraph:

“Previous studies estimated the carbon benefits (including avoided GHG emissions of traditional building materials and carbon storage in CLT panels) of 1 m³ CLT to be 1.1–1.5 tCO₂e in the U.S.^{7,8}. The results in this study are higher because we account for two factors: one is the increased carbon storage in forests, and the other is the decrease of life-cycle GHG emissions due to the reduction in traditional wood products. Without these factors, our estimated global average GHG benefit per 1 m³ CLT by 2100 is 2.1 tCO₂e, which is comparable to the previous studies. The remaining differences likely arise from the different time frames, locations, building structures, and end-of-life assumptions.”

References:

7. Nepal, P. et al. The potential use of mass timber in mid-to high-rise construction and the associated carbon benefits in the United States. PLoS One 19, 1–18 (2024).
8. Taylor, A., Gu, H., Nepal, P. & Bergman, R. Carbon Credits for Mass Timber Construction. Bioprod. Bus. 8, 1–12 (2023).

5.5. Comment: Because your substantial estimated carbon benefits of mass timber adoption stem from large increases in planted forests and forest management activities, it is important that authors review and compare what prior studies have found regarding price effects on forest area/forest management intensities (these do not necessarily need to be mass timber specific studies). Your results that forest carbon shares 45% to 69% of total carbon benefits of mass timber adoption implies almost one to one relationship of price increase with planted forest and forest management activities (and hence forest stock/forest carbon), given your projection of up to 59% increases in timber price by 2100. Please

corroborate your findings with prior studies on the effects of timber price increases on forest land use/management.

Response: We thank the reviewer for the comments. The Sohngen et al. (2001) paper found that climate change would lead to a 20% increase in timber supply, which lowered timber prices by 17% and caused the size of managed forests to decline by 10-16%. Sohngen and Mendelsohn (2003) predict that a global carbon sequestration program that would encourage 100 GtCO₂ to be stored in forests by 2100 would increase carbon prices by 34% and timber flow by 34% and lead to about a 27% increase in forestland. Favero et al. 2017 found that a combination of forest sequestration and woody BECCS production that would remove 908 Gt CO₂ through the forest by 2100 would drive timber prices more than four times higher, increase forestland by 57% (+2/3.5), and almost doubling timber output. Further, Favero et al. 2023 showed how future price expectations will drive the size of investments in forests in the short and long term, affecting carbon stored in forests. Finally, in a recent multi-model comparison, Daigneault et al. 2022 showed how expected increases in timber prices drive forest expansion and management intensification to boost forest productivity with corresponding effects on forest carbon stock.

References:

- Sohngen, B. and R. Mendelsohn. 2003. "An Optimal Control Model of Forest Carbon Sequestration" *American Journal of Agricultural Economics* **85** 448-457.
- Sohngen, B., R. Mendelsohn and R. Sedjo. 2001. "A Global Model of Climate Change Impacts on Timber Markets" *Journal of Agricultural and Resource Economics* **26(2)**: 326-343.
- A Favero, J Baker, B Sohngen, A Daigneault. 2023. Economic factors influence net carbon emissions of forest bioenergy expansion, *Communications Earth & Environment* 4 (1), 41
- Favero, A., Mendelsohn, R. & Sohngen, B. Using forests for climate mitigation: sequester carbon or produce woody biomass? *Clim. Change* 144, 195–206 (2017).
- A Daigneault, JS Baker, J Guo, P Lauri, A Favero, N Forsell, C Johnston, ...2022. How the future of the global forest sink depends on timber demand, forest management, and carbon policies. *Global Environmental Change* 76, 102582

5.6. Comment: A whole building LCA (WBLCA) is an appropriate approach to compare GHG emissions of wood vs alternate buildings. It is not clear if the process-based LCA that you are using, combined with prior literature (e.g. D'Amico et al. 2021) uses WBLCA data. If you are not using WBLCA, then please provide a justification for not using WBLCA approach and how your results would have changed had you chosen to use WBLCA approach. If you use WBLCA, clarify it in the methods section.

Response: We thank the reviewer for this question. We agree that a whole building LCA (WBLCA) is an appropriate approach to compare the life-cycle GHG emissions of wood in specific buildings or alternative buildings. According to the literature (Feng et al., 2022), a “*whole-building life cycle assessment (WBLCA) has pervaded the analysis of the overall building performance by monitoring and assessing buildings’ life-cycle environmental impacts (e.g., production, construction, operation and maintenance, and decommission phases)*”. In this study, we are focusing on quantifying the carbon impacts of adopting mass timber products on the global forest and forest product sector by using consequential LCA. We do not intend to analyze individual buildings worldwide, as this falls outside the scope of our study and is not feasible from both data and modeling perspectives. To conduct the consequential LCA, the LCI data majorly contains the mass and energy balances from the process models for wood products and end-of-life. We also utilized the building material usage data for 1 m² floor area of traditional buildings and CLT buildings from the WBLCA study (D’Amico et al. 2021) as the reviewer mentioned. The building material usage data for 1 m² floor area from this study was adopted to evaluate the potential substitution benefits of CLT over the traditional building materials. To better clarify this point, we have revised our manuscript accordingly.

In Page 22, the last paragraph:

“We develop an integrated framework that combines the GTM (an economic forest model) and LCA models to assess the carbon consequences of adopting emerging mass timber products from 2020 to 2100. ... The LCA also estimates both the fossil-based and the biogenic GHG emissions of production of these *wood products based on the process-based models, as well as the potential substitution benefits of CLT replacing the steel and cement based on the average data from the whole building LCA literature*²⁷.”

Reference:

Feng, H., Zhao, J., Zhang, H., Zhu, S., Li, D., & Thurairajah, N. (2022). Uncertainties in whole-building life cycle assessment: A systematic review. *Journal of Building Engineering*, 50, 104191.

D’Amico, B., Pomponi, F., & Hart, J. (2021). Global potential for material substitution in building construction: The case of cross laminated timber. *Journal of Cleaner Production*, 279, 123487.

5.7. Comment: Specific Comments: Scenario development: Many countries are already producing and consuming CLTs today and they will continue to do so in future. Provide an estimate of the current global level CLT/mass timber production for a context, and also provide a justification of why baseline scenario does not consider any current demand in CLT.

Response: We appreciate the question from the reviewer. Currently, several estimations of the CLT

market are available, but with large uncertainty. According to a survey by Larasatie et al. (2020), the estimated global CLT output in 2019 is 1.4 million m³, based on responses from 12 out of 66 CLT companies. The U.S. Maine State Department of Economic and Community Development (2021) estimated the 2020 global CLT production capacity to be 2.8 million m³. These numbers are substantially different, but overall they are less than 0.3% of the annual global wood supply. Hence, in this study, we did not assume a number for the current CLT demand. This practice is consistent with the methodology employed by the study (D’Amico et al., 2021). To better clarify this point, we have added discussions accordingly.

In Supplementary Note 5:

“Note that the baseline of this study excludes current CLT supply in 2020 due to data uncertainty and its minimal contribution to the total global wood supply (less than 0.3%) based on the literature estimation^{10,23}.”

References:

D’Amico, B., Pomponi, F., & Hart, J. (2021). Global potential for material substitution in building construction: The case of cross laminated timber. *Journal of Cleaner Production*, 279, 123487.

Larasatie, P., Albee, R., Muszynski, L., Guerrero, J. M., & Hansen, E. (2020). Global CLT industry survey: The 2020 updates. In *World Conference on Timber Engineering; WCTE: Santiago, Chile* (pp. 1-8).

State of Maine DECD. (2020). Cross Laminated Timber Market Profile. chrome-extension://efaidnbmnnnibpcajpcglclefindmkaj/https://www.maine.gov/decd/sites/maine.gov.decd/files/in-line-files/Market%20Profile%20%20-%20Cross%20Laminated%20Timber.pdf

5.8. Comment: ‘Introduction’ heading is missing.

Response: We thank the reviewer for pointing it out. We have added the Introduction heading to the manuscript.

5.9. Comment: Line 44. This statement is not true. Nepal et al. (2021) considers the effects of increasing wood demand on markets.

Response: We thank the reviewer for this comment. Nepal et al. (2021) indeed consider the effects of increasing wood demand on markets but did not consider the life-cycle GHG emissions change associated with producing various industrial wood products, including pulp and paper products and sawn timber products. Additionally, as the reviewer mentioned in Comment 5.2, another innovation of this study is the

assessment of how various forest types, forest management activities, and timber markets will change in response to the new demand for mass timber. To clarify this point, we have revised the manuscript and cited the paper here.

In Page 2, the first paragraph:

“Previous studies have shown the potential carbon benefits of adopting mass timber products over traditional construction materials^{6–8}. Life Cycle Assessment (LCA) has been employed at the product level (e.g., 1 m³ wood product)^{9,10} and the stand level (e.g., 1-ha forest land)¹¹ to quantify the environmental performance of mass timber products. Although these studies advanced the knowledge of CLT, most of them only include the direct impacts associated with wood product life cycles (e.g., forest growth, harvesting, wood production, product use, and end of life). They do not consider the effects of increasing wood demand on the life-cycle GHG emissions of both traditional wood products (e.g., lumber, particle boards, pulp and paper products) and mass timber products, and the long-run implications to land use and forest management¹⁰.”

References:

6. Churkina, G. et al. Buildings as a global carbon sink. *Nat. Sustain.* 3, 269–276 (2020).
7. Nepal, P. et al. The potential use of mass timber in mid-to high-rise construction and the associated carbon benefits in the United States. *PLoS One* 19, 1–18 (2024).
8. Taylor, A., Gu, H., Nepal, P. & Bergman, R. Carbon Credits for Mass Timber Construction. *Bioprod. Bus.* 8, 1–12 (2023).
9. Chen, C. X., Pierobon, F. & Ganguly, I. Life Cycle Assessment (LCA) of Cross-Laminated Timber (CLT) produced in Western Washington: The role of logistics and wood species mix. *Sustainability* 11, (2019).
10. Athena Sustainable Materials Institute. A Life Cycle Assessment of Cross-Laminated Timber Produced in Canada. <http://www.athenasmi.org/resources/publications/> (2013).
11. Lan, K., Kelley, S. S., Nepal, P. & Yao, Y. Dynamic life cycle carbon and energy analysis for cross-laminated timber in the Southeastern United States. *Environ. Res. Lett.* 15, 124036 (2020).

5.10. Comment: Line 46. More than these two studies have investigated the global forest impacts of mass timber adoption. See for example, Nepal et al. (2021).

Lines 48 to 51. This statement is not true. Nepal et al. (2021) reports that global increases in mass timber demand would not only increase wood prices and therefore reduce quantities of traditional industrial wood products (e.g., lumber, particle boards, pulp and paper products).

Response: We appreciate the reviewer for the point. As mentioned above, we have revised this part based on the reviewer's recommendation in Comment 5.2.

In Page 2, the last paragraph:

“Several studies have investigated the global forest impact of mass timber adoption^{6,12,13}, recognizing the value of using CLT as part of a potential global climate mitigation program. However, these studies have not considered how various forest types, forest management activities, and the timber market will evolve in response to the new demand for mass timber. Moreover, they did not consider the carbon consequences throughout the life cycles of both CLT and traditional wood products, from forest to the end-of-life.”

References:

6. Churkina, G. et al. Buildings as a global carbon sink. *Nat. Sustain.* 3, 269–276 (2020).

12. Mishra, A. et al. Land use change and carbon emissions of a transformation to timber cities. *Nat. Commun.* 13, (2022).

13. Nepal, P., Johnston, C. M. T. & Ganguly, I. Effects on global forests and wood product markets of increased demand for mass timber. *Sustain.* 13, (2021).

5.11. Comment: Line 50 and throughout the manuscript. CLT is also an industrial wood product (IWP). Consider replacing the term industrial wood products (IWP) with traditional wood products.

Response: We thank the reviewer for this point. We have changed the **industrial wood products** to **traditional wood products** throughout the manuscript and supporting information. Due to the substantial changes, we kindly ask the reviewer to refer to the manuscript and supporting information.

5.12. Comment: Line 66. What do you mean by reduce market? Market is a broad term; be specific and clarify whether you mean reduced supply, reduced demand, or reduced trade of these traditional products. Also specify whether this result is true at the global or regional level? Some countries may reduce production while other could increase production depending on their comparative advantage.

Response: We thank the questions from the reviewer. We are referring to further reducing the global supply here. To clarify this point, we have revised the manuscript accordingly.

In Page 3, the third paragraph:

“We utilize GTM to capture how adding the demand for mass timber products will likely lead to higher wood prices which will further potentially reduce the global supply for traditional wood products.”

5.13. Comment: Line 68. Clarify what is meant by increased forest management intensity? What kind and level of silvicultural activity? (e.g., is it thinning only or thinning and fertilization etc.).

Response: We thank the reviewer for these questions. To clarify this, we have added the following description.

On page 3, the third paragraph:

“The higher wood prices give landowners an incentive to plant more forests and increase forest management intensity *by, for instance, increasing fertilization or changing forest rotation.*”

5.14. Comment: Line 76. What do you mean by wood going market to IWP? Correct/clarify.

Response: We appreciate the question from the reviewer. We have revised this sentence to clarify it.

On page 4, the first paragraph:

“The higher prices, however, will *reduce* the quantities of future wood *that is produced into traditional wood products.*”

5.15. Comment: Line 77. Future changes in supply does not necessarily increase carbon. Future changes in supply may also decrease carbon, for example, when we supply more wood than we grow from forest. Correct/clarify the sentence.

Response: We thank the reviewer for this point. We have revised the sentence to clarify it.

On page 4, the first paragraph:

“The changes in supply, in turn, *impact* carbon storage.”

5.16. Comment: Line 96 to 114. The text and figure here all describe methods. There is nothing about results. These should go to the methods section.

Response: We thank the reviewer for this comment. Due to the journal’s requirements, the result section appears before the methods section. Given the modeling details and complexity of our study, we believe it is important to provide a brief overview of the modeling methods before diving into the results. This approach can help readers better understand the results by providing necessary contexts. This practice is common in this journal, as demonstrated by other LCA studies as shown below. To better clarify this point, we have added the subsection heading “**Overview of Methods and Scenarios**” to this part. We hope this can be acceptable to the reviewer.

Example LCA studies published in *Nature Communications* that present an overview of methodology and scenarios at the beginning of their result sections.

Machala, M. L., Chen, X., Bunke, S. P., Forbes, G., Yegizbay, A., de Chalendar, J. A., ... & Tarpeh, W. A. (2025). Life cycle comparison of industrial-scale lithium-ion battery recycling and mining supply chains. *Nature Communications*, *16*(1), 988.

Bhattacharya, A., Papakonstantinou, K. G., Warn, G. P., McPhillips, L., Bilec, M. M., Forest, C. E., ... & Chavda, D. (2025). Optimal life-cycle adaptation of coastal infrastructure under climate change. *Nature Communications*, *16*(1), 1076.

Zhang, X., Schwarze, M., Schomäcker, R., van de Krol, R., & Abdi, F. F. (2023). Life cycle net energy assessment of sustainable H₂ production and hydrogenation of chemicals in a coupled photoelectrochemical device. *Nature Communications*, *14*(1), 991.

5.17. Comment: Line 124. Correct typo (two periods).

Response: We thank the reviewer for this point. We have corrected it.

5.18. Comment: Line 131. Correct typo. A space needed before the number “estiamte34.2%”.

Response: We thank the reviewer for this point. We have corrected it.

5.19. Comment: Line 131. Add missing word between the words ‘new’ and ‘will’.

Response: We appreciate the comment from the reviewer. We have revised the sentence as suggested.

On page 7, the first paragraph:

“We estimate 34.2%–40.0% of the *new demand will* come from developed countries, while the rest is from developing regions (see Supplementary Figure 1).”

5.20. Comment: Lines 150 to 151. Figure 2. Where are fig a and b? How is Fig 2 title different from its fig 2b title?

Response: We thank the reviewer for this question. We have revised the title of Figure 2.

In Page 8:

“Fig. 2. *Supply of sawtimber for CLT production from 2020 to 2100.*”

5.21. Comment: Lines 171 to 172. This is contrary to what you said above in lines 157-159: “The higher future prices stimulate an increase in the future total wood supply (including sawtimber for IWP, pulpwood, and sawtimber for CLT) (see Supplementary Table 1)”.

Response: We thank the reviewer for the question. This statement “Finally, it is important to recognize that higher future wood prices will reduce the quantity of wood going to sawtimber and pulpwood, so there is more wood available for CLT.” is not contrary to “The higher future prices stimulate an increase in the future total wood supply (including sawtimber for IWP, pulpwood, and sawtimber for CLT) (see Supplementary Table 1)”. As shown in Supplementary Table 1, the total wood supply increases as the wood prices increase. For example, in 2100, the total wood supply increases from 3627 Mm³ in the baseline to 3913-4014 Mm³ in Scenarios 1-3. But the supply of sawtimber for traditional IWP and pulpwood decreases. For example, in 2100, the supply of sawtimber for traditional IWP decreases from 2358 in the baseline to 2195-2374 Mm³ in Scenarios 1-3. To better clarify this, we have revised this part.

On page 8, the first paragraph:

“Finally, it is important to recognize that higher future wood prices will reduce the quantity of wood going to *sawtimber for traditional wood products* and pulpwood, so there is more wood available for CLT.”

5.22. Comment: Lines 216-217. The distinction between plantations and managed forests is not clear. It reads that managed forests are also plantations but planted on less productive land. Is it correct? If so clarify.

Response: Correct. We have added more clarification here.

On Page 11, caption of Fig. 4:

“Managed forests, *referring to managed naturally generated forests*, originally were *naturally regenerated*, but most of them are now replanted forests. The key distinction between plantations and managed forests is that plantations are on highly productive land and involve much more costly and intense management (as in a farm). Natural forests are defined as inaccessible, unmanaged, and naturally regenerating forests.”

5.23. Comment: Line 282. How does CLT reduce/halt the extent of future deforestation? By adding new plantation? Clarify. What kind of forest management investments? Specify.

Response: We thank the reviewer for asking these questions. We now have revised the discussion here to make it clear.

On page 14, the last paragraph:

“Relative to the baseline without CLT, future CLT encourages less future low-valued cropland and more managed forestland. Higher wood prices also increase forest management investments (*increasing plantations and managed forests, as shown in Fig. 4*), raising forest growth rates.”

5.24. Comment: Line 326-328. Nepal et al. (2021) also reported market competition for wood between CLT and traditional wood products. They showed how increased demand for wood for CLT can divert use of wood away from traditional wood products.

Response: We thank the reviewer for this recommendation. We have added the discussion as the reviewer suggested.

On page 17, the first paragraph:

“Another insight of the economic framework is that CLT will actively compete with *traditional wood products for wood*¹³.”

5.25. Comment: Line 346. Clarify that carbon storage means the sum of carbon sequestered in forests, carbon stored in wood products carbon in use, and carbon stored in wood products in landfills.

Response: We appreciate the suggestion by the reviewer. As suggested, we have clarified this point in the manuscript.

On page 18, the first paragraph:

“The net increase in carbon storage, *including in forests (standing trees, soil, slash), in CLT, in traditional wood products, and in landfills*, is 75%, 82%, and 65% of the total net GHG emission reduction for Scenario 1, 2, and 3, respectively.”

5.26. Comment: Line 358. Is comparing your results with only one study enough to assert that your finding agrees with the literature? How does your results compare with other studies (e.g., Churkina et al. 2020, Taylor et al. 2023, Nepal et al. 2024?). I know that these studies do not use the same scenario or same methods as yours but they provide you some reference points to compare and contrast your results.

Response: We thank the reviewer for this recommendation. We have added two additional references to this sentence.

In Page 18, the third paragraph:

“The results of this study agree with the literature that adopting CLT will lead to net beneficial carbon consequences^{7,8,12}.”

5.27. Comment: Lines 433. See general comment #3 (Comment 5.6).

Response: We thank the reviewer for this question. Indeed, a whole building LCA (WBLCA) is an appropriate approach to compare the life-cycle GHG emissions of wood in specific buildings or alternative buildings. According to the literature (Feng et al., 2022), a “*whole-building life cycle assessment (WBLCA) has pervaded the analysis of the overall building performance by monitoring and assessing buildings’ life-cycle environmental impacts (e.g., production, construction, operation and maintenance, and decommission phases)*”. In this study, we are focusing on quantifying the carbon consequences of adopting mass timber products on the global forest and forest product sector by using consequential LCA. We do not intend to analyze individual buildings worldwide, as this falls outside the scope of our study and is not feasible from both data and modeling perspectives. To conduct the consequential LCA, the LCI data majorly contains the mass and energy balances from the process models for wood products and end-of-life. We also utilized the building material usage data for 1 m² floor area of traditional buildings and CLT buildings from the WBLCA study (D’Amico et al. 2021) as the reviewer mentioned. The building material usage data for 1 m² floor area from this study was adopted to evaluate the potential substitution benefits of CLT over the traditional building materials. To better clarify this point, we have revised our manuscript accordingly.

On page 22, the last paragraph:

“We develop an integrated framework that combines the GTM (an economic forest model) and LCA models to assess the carbon consequences of adopting emerging mass timber products from 2020 to 2100. ... The LCA also estimates both the fossil-based and the biogenic GHG emissions of production of these *wood products based on the process-based models, as well as the potential substitution benefits of CLT replacing the steel and cement based on the average data from the whole building LCA literature*²⁷.”

Reference:

Feng, H., Zhao, J., Zhang, H., Zhu, S., Li, D., & Thurairajah, N. (2022). Uncertainties in whole-building life cycle assessment: A systematic review. *Journal of Building Engineering*, 50, 104191.

D’Amico, B., Pomponi, F., & Hart, J. (2021). Global potential for material substitution in building construction: The case of cross laminated timber. *Journal of Cleaner Production*, 279, 123487.

5.28. Comment: Line 435. “future baseline scenario”. How is a future baseline scenario different from a current baseline scenario? Rephrase for better clarity.

Response: We thank the reviewer for this question. We have revised this to “a baseline scenario” to better clarify this point.

On page 22, the last paragraph:

“We use this framework to examine three CLT adoption scenarios and compare the results with *a baseline scenario* without CLT.”

5.29. Comment: Line 436. “input to the modeling”. I believe that the ssp2 forecasts of future economic, population, and urbanization outcomes are input to the baseline scenario modeling and the estimated CLT demand are input to the CLT scenario modeling. Rephrase/Clarify.

Response: We appreciate the comment from the reviewer. Yes, the reviewer is correct. We have revised this statement to clarify this point.

On page 22, the last paragraph:

“The middle of the road socioeconomic projections of future economic, population, and urbanization outcomes, SSP2 (ref.¹⁹), *are the inputs for projecting CLT demand and GTM.*”

5.30. Comment: Line 438. Why not consider a low adoption scenario. Why only medium slow, medium fast and high scenarios? See specific comment 1.

Response: We appreciate this question from the reviewer. In this study, we used medium and high demanding levels to study the impacts of CLT adoption levels, we used the fast and slow adoption rates to investigate the impacts of adoption speed. For a low adoption scenario, their values will just be lower than existing results with no significant difference in trends. Adding it will not provide any additional insights. Hence, in this study, we focus on medium fast, medium slow, and high fast as the three scenarios. For the specific comment 1 from the reviewer, we have thoroughly addressed it as shown above.

5.31. Comment: Line 440. CLT is also IWP. Consider labeling CLT and traditional wood products instead of CLT and IWP. See specific comment 6 (Comment 5.11).

Response: We thank the reviewer for this point. We have changed the **industrial wood products** to **traditional wood products** throughout the manuscript and supporting information. Due to the substantial changes, we kindly ask the reviewer to refer to the manuscript and supporting information.

5.32. Comment: Line 444. GHG's or GHGs. Correct.

Response: We appreciate the reviewer for this point. We have revised this sentence.

On page 23, the first paragraph:

“The integrated analysis consequently captures the relevant changes in *carbon flows* from carbon storage in forests, products, and end of use pools as well as the direct emissions and avoided emissions caused by CLT.”

5.33. Comment: Line 449, 472 and throughout the manuscript. Clarify what is management intensity? What kinds and levels of silvicultural activities you are referring to?

Response: We thank the reviewer for the questions. Here are some examples of management activities:

- Thinning: Selectively removing some trees to reduce competition, allowing the remaining trees to grow better.
- Clear-Cutting with Regeneration: A method where a larger section of the forest is harvested, but efforts are made to regenerate the area, either through planting or natural regeneration.
- Planting: Often used in highly managed forests where species are selected and planted to achieve specific objectives (e.g., high timber yield, uniform structure).
- Intensive Thinning: More frequent and aggressive thinning to create space for the remaining trees to grow faster.
- Clear-Cutting: The entire area is harvested in one go, often followed by planting or another form of regeneration.

As the price of timber increases under CLT demand, these activities increase because the value of timber is higher, and it justifies more investments per hectare of forests.

We have revised the sentence to make it clear.

On page 23, the second paragraph:

“GTM determines the level of management intensity (e.g., thinning, planting, intensive thinning, clear cutting, natural regeneration, fertilization) together with harvesting rates for each period to maximize the present value of timber market surplus over the next 100 years given future expected demand for timber products including CLT.”

5.34. Comment: Line 476. Add the phrase ‘in the past’ at the end of the sentence for better clarity. although the model has been used to examine the impact of climate change in the past.

Response: We appreciate this recommendation from the reviewer. We have revised the sentence exactly as instructed by the reviewer.

On page 24, the second paragraph:

“In this paper, we do not project the impact of climate change on future forests although the model has been used to examine the impact of climate change *in the past*.”

5.35. Comment: Line 485-486. This phrase to describe CG ‘the cost planting to management intensity’ is not clear. How it is different from the cost of planting CPL?

Response: We appreciate the questions from the reviewer. The first term refers to forest that is already forest, the second to the cost of planting new forests. We have now clarified this in the manuscript.

In Page 25, the second paragraph:

“... C_{PL}^i is the cost of planting *new forests*...”

5.36. Comment: Line 499. The term “stock of land” is unclear I believe you are referring to forest growing stock or forest inventory. If the land is the farmland, then what does the ‘stock of land’ mean?

Response: Thanks. We have replaced “stock” with “amount”.

On Page 25:

“The *amount* of land in each forest type adjusts over time according to:”

5.37. Comment: Line 543. What is consequential LCA and why use it? Why can’t the regular (attributional) LCA be used?

Response: We thank the questions from the reviewer. Consequential LCA (CLCA) describes how physical flows or impacts can change as a consequence of an increase or decrease in demand for the product system; attributional LCA (ALCA) typically describes what portion of the flows or impacts can be attributed to a product (Earles and Halog, 2011). The goal of this study is to quantify the global carbon consequences of future growth of CLT, CLCA should be used given its focus on system changes, in contrast to ALCA that focuses on the environmental burden attributed to an individual product.

Our modeling approach is also aligned with common CLCA considerations. According to Earles and Halog, “*unlike ALCA, CLCA includes unit processes inside and outside of the product's immediate system boundaries. It utilizes economic data to measure physical flows of indirectly affected processes. Moreover, allocation is avoided in CLCA by expanding the system boundary.*” In this study, we have integrated the global forest economic-ecological model with LCA to quantify the carbon consequences of increasing CLT demand. These carbon consequences will not only come from the life cycle stages of CLT (direct processes), but also come from changes in forest types, forest management activities, and timber markets impacted by this increasing demand (indirect processes). This approach is well aligned with CLCA considerations of indirect impacts using economic data.

Reference:

Earles, J. M., & Halog, A. (2011). Consequential life cycle assessment: a review. *The international journal of life cycle assessment*, 16, 445-453.

5.38. Comment: A brief and clear description of how substitution benefits of replacing traditional building materials is needed in the main text. Details can go to the Supplementary materials. The current supplementary material indicates substitution benefits of replacing traditional building materials were calculated based on D’Amato. Need a bit more details (e.g., boundary of LCA used, the estimated substitution factor (e.g., xx kg/m³ etc.), whether WBLCA was used to derive the substitution benefits etc.).

Response: We appreciate the comment from the reviewer. The detailed data were recorded in Supplementary Table 10. To add more details as the reviewer suggested, we have revised the main text and put the details into the supporting information accordingly.

On page 30, the last paragraph:

“Since CLT as a mass timber product can potentially substitute the traditional reinforced concrete and steel in buildings (*life span assumed as 60 years^{45,46}*), this study calculates the potential substitution benefits of CLT to replace the conventional structural materials by providing the same floor area²⁷. *The details related to the material usage of the CLT and traditional building structures are available in Supplementary Note 2 and Supplementary Table 10.*”

In Supplementary Note 2:

“In this study, the potential benefits of substituting CLT buildings for traditional reinforced concrete and steel buildings are also considered¹⁰. The structural material consumption for CLT buildings and steel &

concrete buildings are estimated on 1 m² floor area basis, *based on the whole building LCA study by D'Amico et al.¹⁰, as shown in Supplementary Table 10 (ref.^{10,11}). For the upstream production of the building materials, this study uses the globally average processes from the Ecoinvent 3.9 cut-off database (see Supplementary Table 10) to determine the GHG emission reduction per m² floor area. The recycling rate for steel is assumed to be 65% based on the literature¹¹. The rest of the steel is landfilled. The upstream production burdens of steel and concrete products and landfill of steel are derived from Ecoinvent 3.9 cut-off database (see Supplementary Table 7)¹².”*

Supplementary Table 10:

Supplementary Table 10. Average structural material usage for 1 m² floor area^{10,11}.

Material	Unit	Steel & concrete building	CLT building
Steel frame	kg	30.8	27.7
Concrete	kg	230	
Steel deck	kg	14.3	
Reinforcing bar	kg	8.9	
CLT (also referred as f_{CLT} in equation (8) in main text)	m ³		0.12

5.39. Comment: Lines 592. Clarify that the assumed life span of CLT buildings are 60 years, after which buildings are demolished and CLT panels from demolished buildings are sent to landfills or recycled or both. Also clarify the life span of steel and concrete buildings. These are important pieces of information and should be included in the main text (currently missing from supplementary material or the methods section in the main text).

Response: We thank the reviewer for the recommendations. The life span of CLT building was assumed to be 60 years based on the literature data (Sharma et al., 2011; Andersen and Nengendahl, 2023) and recorded in Supplementary Table 8. To better clarify this in the main text, we have revised the contents as the reviewer recommended.

In Page 30, the first paragraph:

“After the life span of CLT, *assumed 60 years*⁴³, the CLT panels are demolished and the discarded CLT waste is sent to the landfill site as the end-of-life. The GHG emissions from landfill are estimated based on the IPCC First Order Decay method for landfill^{11,44}. Since landfill gas contains a high-volume fraction of CH₄, the energy recovery from landfill gas is considered. The details of landfilling are shown in Supplementary Note 3 and Supplementary Table 9. Besides the landfill case, two additional end-of-life cases of CLT panels are conducted: 1) material recycling case with 50% closed-loop recycling and 50% landfilling; 2) energy recovery case with 100% CLT panels combusted for power generation. These

robustness checks explore how these alternative assumptions change the results. More details are available in Supplementary Note 2.”

In Page 30, the last paragraph:

“Since CLT as a mass timber product can potentially substitute the traditional reinforced concrete and steel in buildings (*life span assumed as 60 years^{45,46}*), this study calculates the potential substitution benefits of CLT to replace the conventional structural materials by providing the same floor area²⁷. *The details related to the material usage of the CLT and traditional building structures are available in Supplementary Note 2 and Supplementary Table 10.*”

References:

- Sharma, A., Saxena, A., Sethi, M., & Shree, V. (2011). Life cycle assessment of buildings: a review. *Renewable and Sustainable Energy Reviews*, 15(1), 871-875.
- Andersen, R., & Negendahl, K. (2023). Lifespan prediction of existing building typologies. *Journal of Building Engineering*, 65, 105696.

References provided by the reviewer:

- D’Amico, B., Pomponi, F. & Hart, J. Global potential for material substitution in building construction: The case of cross laminated timber. *J. Clean. Prod.* 279, 123487 (2021).
- Churkina, G. et al. Buildings as a global carbon sink. *Nat. Sustain.* 3, 269–276 (2020).
- Nepal, P., Johnston, C. M. T. & Ganguly, I. Effects on Global Forests and Wood Product Markets of Increased Demand for Mass Timber. *Sustainability*. 13(24): 13943 (2021). <https://doi.org/10.3390/su132413943>.
- Nepal, P. et al. The potential use of mass timber in mid-to high-rise construction and the associated carbon benefits in the United States. *PLOS ONE*. 19(3): e0298379. (2024) <https://doi.org/10.1371/journal.pone.0298379>
- Taylor, A.; Gu, H., Nepal, P., Bergman, R.. Carbon Credits for Mass Timber Construction. *BioProducts Business*. 8, 12 (2023). <https://doi.org/10.22382/bpb-2023-001>.

Reviewer #6

6.1. Comment: The main novelty of the paper is that despite additional substantial industrial wood harvest due to wood construction scenarios, forest growing stock is not depleted, but due to intensified forest management forest carbon stock is expected to increase substantially. The latter is caused by increased sawtimber prices due to rising lumber demand, and forest owners are intensifying management in anticipation of the future high lumber demand. From one side, GTM model is unique due to its dynamic optimization structure, which is able to capture forest management change in reaction to future lumber demand growth. On the other hand, no one has perfect foresight of the future. Nevertheless, dynamic optimization model is a valid approach to study possible future scenarios and optimal transitions towards the future. In addition, GTM model captures global wood markets interactions, including interactions between new mass timber products such as CLT and old type of wood-based products (IWP). Higher demand for CLT (driven by assumed CLT adoption rate in the new urban construction) drives additional sawtimber price increase, which in turn reduce demand for IWP products.

However, according to provided GTM method description, long run demand function for wood products is determined by GDP per capita development, price, income and price elasticity. Price elasticity is assumed as -1 based on provided references. At least two of the provided references do not have any information regarding industrial roundwood demand price elasticity. Morland et al., 2018 does provide industrial roundwood supply price elasticity, which isn't the same. Both Morland et al. 2018 and Buongiorno 2015 provide data on traditional wood-based products such as sawnwood, wood-based panels and various paper and paperboards grades. Based on these references price elasticity for the demand of sawnwood and wood-based panels are in the range of -0.1 to -0.5 (-0.3 on average). Price elasticity of -0.3 for traditional wood products imply relatively inelastic demand, while assuming -1 price elasticity for IWP imply rather elastic demand. With the assumed IWP price elasticity of -1, each percent of price increase reduces the demand by the same percentage. While each 1% of price increase with the price elasticity of -0.3 is going reduce demand by 0.3% only. Therefore, assuming elastic demand for IWP products leads to a greater IWP demand reduction due to sawtimber price increases.

One of the main findings of the study is that despite of the high demand CLT scenario the resulting additional sawtimber demand is more moderate due to substantial reductions of IWP products demand. However, based on the more conservative price elasticity estimates (taken from the same references), IWP products demand reductions are likely to be 2-3 times lower, which would result in the higher additional sawtimber demand (in the range of additional of 150 – 200 million m³ on top of high CLT demand scenario).

I am not asking the authors to make new model sensitivity runs with alternative price elasticity for IWP, since the paper has gone through several revisions already. However, it would be good to acknowledge in the discussion part the uncertainty regarding the extent of possible IWP products demand reduction, which will lead to higher sawtimber demands. Consequently, based on higher sawtimber demand, what is the likely change for the overall carbon balance in the forest and forest products plus substitution of other non-wood construction materials. Is even higher demand for CLT going to improve the final carbon balance or other way around?

Besides the problem related to overestimation of IWP demand reduction, the paper in the current state is well written and present very interesting and novel results worth publishing.

Response: We greatly thank the reviewer for the very positive feedback on our work and thinking our paper is “**well written and present very interesting and novel results worth publishing**”. We appreciate the reviewer’s comments on price elasticity and acknowledge that, as GTM is an economic model, the assumptions on demand, income, and supply elasticities do drive the key findings of the manuscript. The choice of using a unitary demand elasticity assumption in GTM is based on the long-term (i.e., decadal) and aggregate nature of the demand function used in the model. There have been extensive reviews of the literature on the demand elasticity for wood (Daigneault et al., 2016). The range in the literature is wide, and it is accurate that the -1.0 used in GTM is at the top end of the range of demand elasticities. The Daigneault et al (2016) paper explored the effect of a long-run price elasticity of -0.5. A price inelastic demand function for wood leads to larger wood price increases over time and encourages larger investments in forest resources (a larger supply response). A more price inelastic demand leads to larger increases in in forestland and higher forest management intensity. As a result, there is even more carbon storage in forests. These changes apply to the modeling in our paper on CLT as well.

There is another effect in our paper if traditional wood products are more priced inelastic. As CLT grows, the substitution between CLT and traditional wood products will be smaller. With a unitary price elasticity, the model predicts that traditional wood products will fall substantially as CLT grows. However, with a -0.5 price elasticity for IWP, there will be much less substitution. The changes in traditional wood products will be much smaller. The increase in CLT will be met by even larger increases in wood supply rather than reductions in traditional wood products. The overall effect will be an even larger increase in carbon absorption and an even larger fraction of this carbon benefit will come from the increase of carbon in forests.

In this study, medium CLT demand of 30% accounts for 7.8% of the total global wood supply in 2100;

high CLT demand of 60% accounts for 15%. We have shown that the high CLT demand scenario led to higher forest carbon benefits and net carbon benefits than two medium CLT demand scenarios. Consequently, since the CLT demand is relatively small compared to other wood products, even with higher CLT demand, it is less likely to adverse the trend of the forest carbon stock benefit. At the same time, an even higher CLT demand will lead to lower wood supply for traditional wood products (reducing the production GHG emissions), increase the CLT carbon stocks, increase the substitution GHG benefits to replace the traditional construction materials. Hence, combining these aspects, if the high CLT demand keeps increasing at the same fast adoption rate, the net carbon benefits are likely to be larger.

To better reflect the aspects that the reviewer recommends, we have added the contents to the discussion section related to this.

On page 20, the second paragraph:

“GTM focuses on capturing the long-term forest sector adjustments that evolve over time in response to long-term demand or supply stimulus. This study adopted a unitary demand price elasticity of -1.0. The literature reports a range of demand price elasticities for IWP with the unitary elasticity on the high end²⁴⁻²⁶. A more price inelastic demand for traditional wood products would lead to larger wood price increases over time and therefore a larger supply response, including more forestland and higher management intensity. This would lead to more carbon storage in forests. However, if traditional wood products have a lower price elasticity, increases in CLT would cause less substitution with traditional wood products. Traditional wood products would shrink much less than we estimated in the high CLT scenarios. The overall result is that CLT would still lead to substantial carbon benefits, but an even larger fraction of those savings would come from the forest itself. Additionally, if the CLT demand is even higher than the high demand scenario with the same fast adoption and elasticity, it can be anticipated that the net carbon benefits are likely to grow with higher forest and CLT carbon stocks, lower production and EOL GHG emissions of traditional wood products, and higher potential substitution benefits.”

Reference:

24. Daigneault, A. J., Sohngen, B. & Kim, S. J. Estimating welfare effects from supply shocks with dynamic factor demand models. For. Policy Econ. 73, 41–51 (2016).
25. Buongiorno, J. Income and time dependence of forest product demand elasticities and implications for forecasting. Silva Fenn. 49, (2015).

26. Morland, C., Schier, F., Janzen, N. & Weimar, H. Supply and demand functions for global wood markets: Specification and plausibility testing of econometric models within the global forest sector. *For. Policy Econ.* 92, 92–105 (2018).

Summary:

Authors have used a combined forest economics model (Global Timber Model i.e., GTM) and an LCA model to explore the potential environmental impacts of using CLT in future building construction scenarios. Their findings suggest that widespread CLT adoption could lead to a significant increase in carbon storage (up to 7 GtC) in standing forests and harvested wood products (including CLTs). Additionally, they estimate a reduction in greenhouse gas emissions of approximately 40 GtCO_{2e} due to improved biogenic carbon uptake and lower emissions associated with HWP production and disposal.

This paper was an excellent read, offering a comprehensive analysis of the environmental consequences of leveraging future timber buildings using CLTs for GHG mitigation. The authors innovatively combined a well-known forest economics model (GTM) with a comprehensive LCA analysis, a unique approach I have not encountered in any other existing study. This undertaking deserves high praise, especially considering how it highlights critical research gaps in current literature.

While I acknowledge the expertise of the co-authors and my own limited experience with an older version of GTM, I believe that open-sourcing both the GTM code and the employed LCA model would significantly enhance the transparency and replicability of this study. This would allow other researchers to verify the results, build upon them, and address potential limitations in future studies.

The research presented here is innovative, but I have significant concerns regarding both the chosen tools and the authors justification for the study. Notably, the authors provide minimal justification for selecting GTM as a suitable model for addressing their research question. GTM is inherently not a land-use model and only simulates specific forest land pools. Additionally, GTM lacks the ability to model competition for land between agriculture, forestry, and other land uses. This significantly weakens the land-use arguments presented in the paper. Furthermore, the authors rely heavily on citing papers written among themselves, necessitating constant referencing back and forth between multiple publications. While they may have extensive background knowledge informing this study, the manuscript lacks sufficient context for neutral readers to fully grasp the results.

Most of the results are presented in relation to the baseline but the manuscript lacks explicit baseline results (excluding Figure 2, where it appears unnecessary, and Figure 3 does have baseline results). This omission hinders our ability to assess the magnitude of differences across the three scenarios compared to the baseline scenario itself. Consequently, we cannot evaluate the performance of the presented modeling framework on the baseline scenario.

This manuscript's writing style primarily reflects a modeling study approach, potentially limiting its accessibility to the broader audience of Nature Communications. Given the lack of broader appeal and the presence of methodological concerns, I would not be able to recommend this manuscript for publication in Nature Communications.

I have attached my detailed and minor comments below which hopefully are of use to the authors.

Detailed Comments:

Introduction:

L44-46: This is an incorrect representation of Citation 10. The land conversion dynamics in the model used by Mishra et al. already take account of the cost of converting land – especially marginal land. Additionally, the assumption made by Mishra et al. is that the mass timber demand is counted as an additional demand on top of business as usually roundwood demand.

L55: How exactly can secondary forest increase? These are naturally regenerating forests. The statement would make sense if secondary forests were newly established/reclassified. Unclear how this dynamic works in the model presented here.

L61: What could happen if we just clear-cut all available forests to meet higher CLT demand hypothetically? The authors have presented their model outputs as part of the introduction in multiple places like this.

L68: The authors state that the framework used here can support policymaking but offer no concrete examples of any policy recommendations throughout the manuscript.

L69: Environmental consequences are not concretely discussed anywhere in the manuscript.

L73: Perhaps something to add in the methods section but are NPIs or NDCs included in forest conservation? I could not find it anywhere mentioned in the manuscript.

Results:

L80: Unclear what 30% refers to (in global new construction) – raw materials? Buildings? Floor space?

L83-85: Does GTM have other wood product categories? It could be useful to see what total roundwood demand looks like in GTM. In 2020, global roundwood production was ca. 4000 Mm³ mostly equally divided between industrial roundwood and wood fuel. From Fig. 3, looks like sawtimber and pulpwood demand goes up to 2300+1200 = 3500 Mm³ by 2100 – but these products are (likely) only a portion of total roundwood production.

L96-97: This is perhaps the main model output from GTM due to the way it is set up, but can the authors elaborate in simpler terms why traditional HWP demand will reduce due to an increase in prices of timber driven by higher demand for CLTs? This dynamic would assume that CLTs are more desirable than traditional HWPs which is not clarified in the manuscript. CLTs are not necessarily a substitute good for sawtimber and pulpwood even if roundwood is used as a raw material for sawtimber (incl. CLTs) and pulpwood. In SI Fig. 1 it looks like sawtimber is converted into CLT. How are the price and demand effects decoupled between traditional sawtimber and CLT?

L99: Unclear what internally consistent means.

L117-120: What is the reason for ROW not contributing much to CLT production?

L128: Fig. 3c and 3d show an opposite effect i.e., an increase in sawtimber and pulpwood prices corresponding to lower supply (also mentioned in L137-138). I would urge the authors to stick to consistent naming throughout the manuscript between wood, timber, etc.

L129: Where can we see this increase in forest management? Is it implicitly shown as forest area increase presented in Fig. 4?

L131: Same as the earlier comment about wood or timber or CLT. Unclear what “wood” means here.

L132 – 133: How does this shift of secondary forests into plantations occur in GTM? Does the investment in managed forests (plantations?) increase yields? If yes – how does it impact the prices of sawtimber and pulpwood in GTM?

L134: As authors already mention ecology and related concepts earlier in the manuscript – how can we justify moving low-value farmland and natural forestland into managed forestland? This is likely detrimental to biodiversity. This is also a likely drawback of the model(s) being used here but is not discussed anywhere in the manuscript.

L151 – why increased CLT demand only drive a small increase in plantations? More specifically – how is the demand signal passed on to equation 3 which determines the stock of land in each forest type? Alternatively, how does GTM decide the extent of the new plantation establishment?

L151 – How exactly does a naturally regenerating forest increase? Does GTM adjust secondary forest area during optimization? Equation 3 has indexes of age classes and time steps. Do all forest classes in GTM have age classes associated with them – certainly looks like it from L395. How was the age-class distribution determined for plantations, secondary forests, and natural forests? This is not clear in methods or SI.

L155:160 – Authors refer to changes compared to the baseline scenario, but we do not see baseline numbers in most results. This makes it difficult to ascertain what is the magnitude of forest area changes in scenarios 1:3.

L155:160 – Demand for sawtimber is almost double in scenario 3 compared to scenarios 1 and 2 – but there is only ca. 3Mha more of net forest change in these scenarios. Can we perhaps see what the timber yields look like in GTM – even in SI? On average this would translate to a yield of about 100m³/ha which seems rather high.

L219 – The methodology presented here severely lacks accounting of the whole land-use system. Presenting a simple ratio between CLTs produced (cumulatively) and GHG changes likely provides an incorrect representation of how much mitigation potential is provided by CLT production. These numbers make sense only in isolation of the modeling boundaries and should be presented in that context.

Discussion:

L257: The only ecological constraint seems to be limiting forestland to locations that can support forests. Does GTM account for the following in any capacity?

- changes in biodiversity
- NPIs already in place and promised NDCs

- Land conservation activities across the globe
- COP26 declaration to end deforestation by 2030.

If not, this should probably become part of the discussion especially when the manuscript focuses on land and carbon consequences of mass timber/CLT adoption.

L263-266: Can the authors elaborate on what exactly is the “net GHG emission effect”? Same comment for the instance of these words in L278.

L284: The 143 Mha additional plantations for 90% of new urban dwellers living in buildings made of mass timber is most likely an infeasible scenario. But to justify why an additional 143Mha of plantations is impractical, authors could compare their CLT demand scenarios to the engineered wood demand from Mishra et al. – the reason is most likely a super high demand for mass timber in Mishra et al. – even higher than the CLT demand in scenario 3 of this manuscript.

L285: “largely ineffective” in which sense? Also, Is the cited reference, correct? The Cited Favero et al. paper nowhere states that plantations are “largely ineffective”. Here is a verbatim quote from the cited paper – “These results show that it is more cost-effective to meet the new demand by investing in plantations than harvesting unmanaged forestland.”

Besides this, plantations cover only about 3% of the global forest area but provide more than 33% of global roundwood production (FAO data). Plantations are not an ideal candidate for supporting biodiversity as such, but they are quick in accumulating biomass if managed correctly, even the monocultures.

Methods:

L330: Authors use harvested timber output here, and HWPs in other places. Consistent naming across the manuscript would help readability.

L350-352: Unclear what the authors want to describe here. Sohngen et al. paper cited here states that uncertainty in the parameters dictating forest growth has an impact on carbon storage potential – more so than uncertainty in the land supply elasticity parameters. It does not appear to be a good fit in describing the functioning of GTM in the context of the methods section.

L367: Is the same income elasticity used for all wood products in GTM? GTM also used to have income elasticity which varied over time applied to GDP per capita. Alternatively, Morland et al. seem to have some numbers for common wood products which could be useful.

Morland, C., Schier, F., Janzen, N., and Weimar, H.: Supply and demand functions for global wood markets: specification and plausibility testing of econometric models within the global forest sector, Forest Policy Econ., 92, 92–105, 2018

L368: These are almost 20-year-old citations for an important parameter of GTM. Have there been no recent updates to this income elasticity parameter?

L376: How is forest management considered in GTM? Same for L379.

L384-L387: How does GTM calculate farm prices? And is GTM looking only at price comparison to make land-use decisions? Is there no “minimum” food, feed, and timber demand in the model to provide a minimum production level and land-use levels for non-forest activities and commodities?

Hypothetical example – if the world was only 4 pixels large, with 2 pixels for agriculture, and 2 pixels for forests – with some people living in this hypothetical world. Could GTM convert all 4 pixels to forests if farm prices fall below timber prices? How would people feed themselves? It is most likely a non-issue if GTM has no agriculture component but this could be clarified in methods and also stated in discussions if possible.

L397: Where is the initial stock of land taken from? A citation would help here.

L399: What is the decision-making behind the newly planted forest? i.e., how is N_t^i determined?

L402: Unclear what “competition for supply equilibrates their prices” means.

Eqn. 4 and 5 - Missing symbol descriptions. Same for many other equations. A table for a description of all indices and all symbols would improve the readability of equations.

Minor comments:

L20-21: in which period did this expansion happen? (likely by 2100?)

L24: When talking about total net GHG – which sectors are included? Perhaps being specific about sector(s) could be useful.

L52: Missing citation for previous studies.

L53: Seems more like a model output rather than a statement for the introduction section.

L55: “Growing wood” is a strange term.

L78: Results sections should ideally not start with stating an assumption.

L86-89: Ref to SI Fig. 1 missing to improve readability.

L216: Perhaps cumulative is a better word (instead of accumulative)?

Fig. 4 – Can we see global numbers in SI (sum of all regions)?

L325: baseline (scenario?)

L328: Are LCA models feeding into GTM or the other way around?

L350: Extent possible in which sense?

L365: Industrial (roundwood?) demand?

L375: Missing citation.

L409: Perhaps above ground is a better term?

L433: Missing citation

Eqn. 12 – Is all HWP carbon considered sequestered? Eqn 14:16 does contain a decay parameter according to IPCC methodology. Perhaps the LCA model accounts for this already?

L490: Hypothetically, if a building is made from the same batch of CLT and arrives at the end-of-life at the same time – would whole buildings be dismantled and recycled?

I appreciate the authors' efforts in providing additional background details for their manuscript. However, I still have reservations about the suitability of the chosen modeling framework for addressing the research question at hand. While GTM is a robust model, it might not be the most appropriate tool to answer the research question presented. The manuscript has attempted to fill an important research gap but falls short of justifying using the modeling framework presented to fill this research gap. Despite the authors' detailed responses to my previous review, some of my concerns remain unaddressed.

Regrettably, based on the current state of the manuscript, I am unable to recommend it for publication in Nature Communications.

Below, I offer further comments based on the authors' responses, which could aid them in refining their work for submission to another journal. It is worth noting that while this is not a *forestry* or *forest economics* paper, providing additional background clarification could strengthen the authors' case with future reviewers.

Authors comments are in bullet points, and my comments follow the authors response.

-
- The more land that the forest wants from farmland, the more expensive the land gets. This recognizes the price inelasticity of farmland (the underlying demand function for food)

Price inelasticity of farmland is also influenced by competition for land, which also comes from other land pools which authors do not cover (non forested land, pasture and rangeland etc.). From the description it feels like the modeling world only consists of farmland (Cropland?) and forests (managed or unmanaged) which is not a justifiable assumption.

- We now make clear in the text that CLT causes only a small change in the global demand for wood. Even with the most aggressive CLT scenario, the global demand for wood increases by only 11%

That was not the reasoning behind requesting baseline results. The idea was to see how the model behaves in the baseline scenario, and perhaps compare it to historical numbers for validation.

Also stating that in the most aggressive CLT scenario results in 11% increase in global demand for (industrial roundwood?) seems low. Between 1995 and 2020, industrial roundwood production increased by ca. 30% and between 1962 and 1992 the industrial roundwood demand grew by 54%. Alternatively, it could be argued that CLT demand did not exist in the past so the demand for industrial roundwood increased without pressure from CLT demand. Also, there is case to be made that this production (or demand as proxy) is declining in recent times, and competition for CLT might reduce it even more, but having only an 11% increase in most aggressive CLT demand scenario seems dicey.

(1962 - 1992) <https://www.fao.org/3/w7705e/w7705e0b.htm>

(1995 - 2020) <https://www.fao.org/faostat/en/#data/FO>

- Response to “2.2”

Like earlier, this seems to be an incorrect interpretation of existing literature. Modeling framework used in Mishra et al. can indeed account for land conversion possibilities. It is quite likely that land which cannot support plantations is never used for establishing new plantations due to biophysical constraints coming from the DGVM used (LPJmL).

- Response to “2.4”

This is more of a global north perspective. I appreciate the historical context provided by the authors but saying that “entire world has turned to renewable forest management recently” warrants either a use of citation or it should come with more context. This could also simply be a difference of terminology used. Perhaps the authors meant “sustainable forest management”? If yes, then a sweeping remark on global practices could likely not be justified.

- We have added policy recommendations to the discussion.

I still do not see a solid policy recommendation in edited text. “Further policy studies” is not a solid policy recommendation. Authors have already included a nice statement in their text “Encouraging CLT adoption may be a compelling market tool to increase carbon storage in forests and wood products (...)”. Maybe something along those line could help frame a policy recommendation?

- NDIs measure plantations which are one of the outcomes predicted by GTM

Requesting the authors to stick to the correct nomenclatures. NDCs (or existing NPIs) likely often refer to “planted” forest category for expanding forest cover, and even more specifically almost point to “other planted forest” category from FAO nomenclature. Additionally, GTM probably does not include these other planted forests.

- The reason for not seeing a bigger increase in plantations is that forestland must be highly productive to support a plantation and only a small fraction of forestland fits that description.

Does that mean that we have already reached a saturation point in establishment of new forest plantations? Given the historical development of plantation area, this does not appear to be the case.

- GTM models all managed forestland in the world by age class. This was done by starting with natural forests and carefully modeling when forests were harvested and replanted using historic harvests from Houghton (2008). Natural forests are assumed to have an average natural stocking given the forest type and region.

Houghton et al. paper(s) from 2008 mostly refer to carbon fluxes but not specific to age-classes? Perhaps adding the exact source would help readers. Also, is that the only source of data available to calibrate historical age-class distribution in GTM?

- (...) 27% is managed forestland (of which 9% is plantation)

This is way off from FAO reported numbers. Not all planted forest is “plantation”. From FAO – “Ninety-three percent of the forest area worldwide is composed of naturally regenerating forests and 7

percent is planted. The area of naturally regenerating forests has decreased since 1990 (at a declining rate of loss), but the area of planted forests has increased by 123 million ha.”

- Response to “2.22”

Land cannot be treated in isolation. “CLT is not likely to cause other land uses to change” points that there are not enough complexities in the modeling framework to link all land pools.

The assertion that CLT is unlikely to influence other land uses appears simplistic, considering potential impacts on agricultural production, demand and land-use.

- Response to “2.26”

The model presented in Mishra et al. never claims that “143 Mha of prime cropland” is removed.

- Response to “2.28” and code/data availability

SRES drivers are only used for sensitivity analysis or are they also used as model driver in all scenarios (B2 maybe?). SRES B2 is mostly akin with SSP2 numbers but isn't SRES socioeconomic drivers quite old already at this point?

This is also the reason why I would have liked to see how well the baseline scenario behaves in comparison to historically available data (sans CLT signals). Also, it would be nice to see this sensitivity analysis in SI somewhere rather than taking author's word for it. As a side note, SSPv3 was released few weeks ago, but a good justification of using SRES numbers instead of SSP numbers (even SSPv2) could be useful.

The model code shared does appear to be reflecting that some SSP numbers are used but then why use SRES scenarios for sensitivity? Some comments in the GAMS code refer to the IIASA database which seems to point that at least some SSPv2 numbers were plugged in.

Also, with the model code shared on zenodo, some input files are missing

- param2 and param3 (parameters for forestry model)
- param4 (harvesting costs)
- cparam2 (carbon estimations)
- forinv2 and iforin2 (accessible and inaccessible forests)

So, we have no good way to know what is being fed to the model. Without being able to compile the model it becomes difficult to coherently see model dynamics.

For example:

- Population and GDP growths are hardcoded with only time index. Are the growth rates same for all regions? Or is GTM only doing global numbers for both population and gdp (per capita).
- Both GDP and population numbers are fed into AF which influences AFP (and AFS) which influences some more calculations/equations down the line. But these are also with only a time index (no regional information).